# Projection Pursuit Density Ratio Estimation

**Meilin Wang** [1]  **Wei Huang** [2]  **Mingming Gong** [2]  **Zheng Zhang** [1]

## Abstract

*Density ratio estimation* (DRE) is a paramount task in machine learning, for its broad applications across multiple domains, such as covariate shift adaptation, causal inference, independence tests and beyond. Parametric methods for estimating the density ratio possibly lead to biased results if models are misspecified, while conventional non-parametric methods suffer from the curse of dimensionality when the dimension of data is large. To address these challenges, in this paper, we propose a novel approach for DRE based on the projection pursuit (PP) approximation. The proposed method leverages PP to mitigate the impact of high dimensionality while retaining the model flexibility needed for the accuracy of DRE. We establish the consistency and the convergence rate for the proposed estimator. Experimental results demonstrate that our proposed method outperforms existing alternatives in various applications.

## 1. Introduction

Density ratio estimation (DRE) is a fundamental concept in machine learning that focuses on directly estimating the ratio between two probability density functions (PDFs), avoiding the need to estimate each density function individually. DRE has widespread applications, such as covariate shift adaptation (Shimodaira, 2000; Sugiyama et al., 2007), outlier detection (Hido et al., 2011), independence test (Ai et al., 2024), mutual information estimation (Suzuki et al., 2009; Ai et al., 2024), importance sampling (Meng & Wong, 1996; Sinha et al., 2020), and treatment effect estimation in causal inference (Ai et al., 2021; Matsushita et al., 2023). See Sugiyama et al. (2012) for a comprehensive review of

DRE and its applications in machine learning.

There are a variety of methods concerning the DRE problem in the existing literature, which can be divided into three paradigms: the parametric methods (Liu et al., 2013; 2017; Nagumo & Fujisawa, 2024), the nonparametric linear sieve methods (Kanamori et al., 2009; Sugiyama et al., 2008; 2010) and the nonparametric neural network methods (Nam & Sugiyama, 2015; Fang et al., 2020; Rhodes et al., 2020). The validity of the parametric modeling builds on the correct model specification for the density ratio. If the model is misspecified, the results may be severely biased. Nonparametric neural network methods (Nam & Sugiyama, 2015; Fang et al., 2020; Rhodes et al., 2020) can achieve superior performance without imposing parametric model assumptions; however, they require a substantial number of training samples to learn the extensive parameters involved. Furthermore, theoretical guarantees for neural network estimations remain insufficient.

Nonparametric linear sieve estimation is a statistical technique that combines the flexibility of nonparametric methods with the simplicity of linear models (Chen, 2007b). The term "linear sieve" refers to a sequence of increasingly complex linear models that are used to approximate the unknown function of interest. With different choices of criteria, e.g. squared-loss (Kanamori et al., 2009; Sugiyama et al., 2011; Yamada et al., 2013), Kullback–Leibler divergence (Sugiyama et al., 2008; Tsuboi et al., 2009), the method of sieves is highly flexible in estimating complicated models. Unlike nonparametric neural network methods that can become excessively complex and computationally intensive, the sieve approach seeks to balance model complexity with interpretability and computational feasibility.

Although the linear sieve methods enjoy computational convenience and theoretical support (Kanamori et al., 2010), they suffer from a significant challenge known as the curse of dimensionality. Specifically, as the dimension of the data increases, there is a notable and substantial deterioration in the performance of these linear sieve methods. This phenomenon is clearly evidenced by both ratio pattern visualization and estimation error in our experimental results. Conventional linear sieve methods, such as KLIEP (Sugiyama et al., 2007) and uLSIF (Kanamori et al., 2009) can correctly capture the pattern of density ratio functions when

[1]Center for Applied Statistics, Institute of Statistics and Big Data, Renmin University of China, Beijing, China [2]School of Mathematics and Statistics, University of Melbourne, Melbourne, Australia. Correspondence to: Zheng Zhang <zhengzhang@ruc.edu.cn>.

*Proceedings of the 42$^{nd}$ International Conference on Machine Learning*, Vancouver, Canada. PMLR 267, 2025. Copyright 2025 by the author(s).

the dimension of variables is 2 (see in Figure 1). However, they experience a significant deterioration in performance when the dimension of variables increases to 10 (see Figure 2). The root mean squared logarithmic error (RMSLE) for the DRE based on conventional linear sieve methods further quantifies this performance decline, exhibiting a dramatic escalation as the dimension of variables increases (see in Figure 3). This deterioration in performance is attributed to the sparsity of the data in high-dimensional spaces, which hinders the ability of the linear sieve methods to effectively learn from the data and generalize to new, unseen instances.

To address the curse of dimensionality, a variety of $D^3$ (Direct DRE with Dimension reduction) methods have been proposed (Sugiyama et al., 2010; 2011; Yamada & Sugiyama, 2011) by incorporating a dimension reduction step prior to applying a linear sieve estimation. These dimension reduction techniques assume that the density ratio function is located in a low-dimensional intrinsic space characterized by a linear transform of the high-dimensional variables. However, this fundamental assumption can be restrictive in many practical scenarios.

To overcome these limitations, in this paper, we propose a novel method for DRE based on the projection pursuit (PP) (Friedman & Tukey, 1974). The core idea is to approximate the density ratio function by a product of PP functions and estimate them iteratively. Each of the PP functions is a projection of the target function to a low-dimensional space. As a result, at each iteration, we only need to estimate a semiparametric single-index function based on a *univariate* linear sieve basis. It is known that the approximation error vanishes as the number of iterations approaches infinity, mitigating the curse of dimensionality when the target function can be expressed as many (or infinitely many) low-dimensional projections along different directions (Diaconis & Shahshahani, 1984). We further provide a theoretical justification for our method by establishing its consistency and convergence rate for the proposed estimator. In application, we apply the proposed method in causal inference, mutual information estimation and covariate shift adaptation, and find consistent improvements in performance.

This paper is organized as follows. Section 2 discusses recent advancements in the field through analysis of related work. Section 3 introduces the conventional methods for DRE from the perspective of model specification. In Section 4, we propose the projection pursuit DRE method that can achieve efficient estimation even in high-dimensional data settings, and we establish the theoretical properties. Section 5 demonstrates the superiority of our proposed method in various applications.

## 2. Related Work

While this work primarily addresses the curse of dimensionality in DRE, the field has also made significant progress on two related challenges: stabilizing estimators through effective regularization strategies, and addressing geometric disparities between distributions known as density-chasm effects.

**Regularized Kernel Learning Methods**. The regularization scheme within reproducing Kernel Hilbert space (RKHS) has been developed for estimating the DRE problem. Que & Belkin (2013) reformulated the DRE problem as an inverse problem in terms of an integral operator corresponding to a kernel, then proposed a regularized estimation method with an RKHS norm penalty. Gizewski et al. (2022) applied the regularized kernel methods in the context of unsupervised domain adaptation under covariate shift and developed the convergence rates. Gruber et al. (2024) proposed iterated regularization and developed an improved error bounds faster than the non-iterated error bound under the Bregman distance and certain regular conditions (e.g. source condition and capacity condition). Nguyen et al. (2024) established the pointwise convergence rate of the regularized estimator taking into account both the smoothness of the density ratio and the capacity of the space in which it is estimated.

**Density-Chasm Problem** Density ratio estimation faces an additional challenge known as the density-chasm problem, which occurs when the distributions differ substantially (Rhodes et al., 2020; Choi et al., 2021; 2022). This phenomenon arises because samples are less likely to be observed in the low-density regions between the two distributions. To overcome this challenge, Choi et al. (2021) proposed an invertible parametric transform mapping the data onto a shared feature space, thereby bringing the transformed densities become closer. Then they estimate the density ratio in the feature space based on the key property that, the ratio remains invariant under such invertible transformation. Notably, this invertible transformation preserves the original data dimensionality. Therefore, it is necessary to clarify that they neither map the data to a low-dimensional latent space nor address the curse of dimensionality as focused by our paper.

## 3. Density Ratio Estimation

Let $p(\boldsymbol{x})$ and $q(\boldsymbol{x})$ be two probability density functions of the target and reference datasets respectively, where $\boldsymbol{x} \in \mathbb{R}^d$ is a $d$-dimensional variable. The density ratio estimation (DRE) problem is to estimate

$$r^*(\boldsymbol{x}) := \frac{p(\boldsymbol{x})}{q(\boldsymbol{x})}$$

based on two independently and identically distributed (*i.i.d.*) samples from the two referred distributions, i.e. $\{x_i^p\}_{i=1}^{n_p} \overset{\text{i.i.d.}}{\sim} p(\boldsymbol{x})$ and $\{x_i^q\}_{i=1}^{n_q} \overset{\text{i.i.d.}}{\sim} q(\boldsymbol{x})$. To make the density ratio function $r^*(\boldsymbol{x})$ well-defined, we assume $q(\boldsymbol{x})$ dominates $p(\boldsymbol{x})$, i.e. $p(\boldsymbol{x}) > 0$ implies that $q(\boldsymbol{x}) > 0$, and $\mathcal{X} \subset \mathbb{R}^d$ denotes the support of $r^*(\boldsymbol{x})$.

Below, we briefly summarize the existing methods for the DRE problem and highlight their limitations.

**Parametric Methods**   The density ratio function is assumed to satisfy the following parametric model (Liu et al., 2013; 2017; Nagumo & Fujisawa, 2024):

$$r^*(\boldsymbol{x}) = C \exp\{\boldsymbol{\theta}^\top h(\boldsymbol{x})\},$$

where $C \in \mathbb{R}$ is a normalizing constant, $h(\boldsymbol{x}) : \mathbb{R}^d \mapsto \mathbb{R}^p$ is the feature transformation function, $p$ is a known *fixed* positive integer, and $\boldsymbol{\theta} \in \mathbb{R}^p$ is the parameter of interest to be estimated. See Nagumo & Fujisawa (2024, Appendix A) for more detailed discussion on this parametric formulation. The parametric formulation is particularly useful for sparse estimation in high-dimensional problems. However, it relies heavily on the correct model specification and may fail to capture non-linear relationships or interactions within the data as effectively as non-parametric methods. This limitation diminishes their adaptability and flexibility in handling complex or varying datasets.

**Nonparametric Linear Sieve Methods**   The density ratio function is approximated by a sequence of increasingly complex linear models (Kanamori et al., 2009; Sugiyama et al., 2008; 2012):

$$r^*(\boldsymbol{x}) \approx r_p^{lm}(\boldsymbol{x}) := \boldsymbol{\theta}^\top \boldsymbol{\psi}(\boldsymbol{x}),$$

or log-linear models (Kanamori et al., 2010; Tsuboi et al., 2009):

$$r^*(\boldsymbol{x}) \approx r_p^{llm}(\boldsymbol{x}) := C \exp\{\boldsymbol{\theta}^\top \boldsymbol{\psi}(\boldsymbol{x})\},$$

where $\boldsymbol{\psi}(\boldsymbol{x}) : \mathbb{R}^d \mapsto \mathbb{R}^p$ is a user-specified basis function (also called "sieve" in statistical literature), $\boldsymbol{\theta} \in \mathbb{R}^p$ is the $p$-dimensional coefficient parameter, and $p \in \mathbb{N}$ is unknown and will go to infinity at an appropriate rate as the sample size increases. The linear sieve methods adaptively learn the model parameters $(\boldsymbol{\theta}, p)$ from the data by minimizing the empirical discrepancy between the true density ratio function and its linear sieve estimators under some distance measure (e.g. the Kullback-Leibler divergence, the squared distance, and the general Bregman distance). The linear sieve methods seek to balance model complexity with interpretability and computational feasibility.

However, the performance of linear sieve methods will significantly deteriorate as the dimension of $\boldsymbol{x}$ becomes large. Such a curse of dimensionality can be explained as follows. The linear sieve approximation error for the density

ratio function is of order $p^{-s/d}$ (Lorentz, 1986, Theorem 8), where $s$ is the Hölder-smoothness of $r^*(\boldsymbol{x})$, thus it requires $p$ larger than $[1/\epsilon]^{d/s}$ to achieve an $\epsilon$-error approximation accuracy. On the other hand, a large $p$ will significantly enlarge the variance of these estimators; indeed, the variance of the linear sieve estimator is of order $\sqrt{p/n}$ (Li & Racine, 2023, Theorem 15.1).

**Deep Neural Network Methods**   Several deep neural network-based methods have been proposed for the DRE problem (Nam & Sugiyama, 2015; Fang et al., 2020; Rhodes et al., 2020; Kato & Teshima, 2021; Choi et al., 2022), with the aim of improving performance in high-dimensional data. Despite their superior performance, these methods tend to lack good interpretability and theoretical guarantees. Moreover, the effectiveness of deep neural network estimation is intrinsically dependent on the availability of substantial training datasets, and its computational burden is much heavier than the parametric and linear sieve methods.

## 4. Projection Pursuit DRE

Projection pursuit (PP) is a statistical technique used for multidimensional data analysis. The key idea is to approximate a high-dimensional function by progressively projecting it onto efficient low-dimensional spaces that capture the most significant features of the data structure. It was first developed by Friedman & Tukey (1974) for exploratory data analysis and has been applied to various problems such as PP-classification (Lee et al., 2005; da Silva et al., 2021), PP-regression (Friedman & Stuetzle, 1981; Zhan et al., 2025), and PP-density estimation (Friedman et al., 1984; Aladjem, 2005). Being inspired by the strengths of PP in mitigating the curse of dimensionality (Huber, 1985), we propose a PP-based method for estimating the density ratio $r^*(\boldsymbol{x})$ and develop valid asymptotic theories.

### 4.1. Projection Pursuit Approximation

We propose to approximate the density ratio function $r^*(\boldsymbol{x})$ by the multiplicative projection pursuit:

$$r^*(\boldsymbol{x}) \approx r_K(\boldsymbol{x}) = \prod_{k=1}^{K} f_k(\boldsymbol{a}_k^\top \boldsymbol{x}), \tag{1}$$

where $K \in \mathbb{N}$ denotes the number of projections, $\{\boldsymbol{a}_k\}_{k=1}^K$ are unit $d$-dimensional vectors indicating the projection directions, and $\{f_k\}_{k=1}^K$ are unknown univariate pursuit functions. That is, the PP approximation (1) converts the estimation of $r^*(\boldsymbol{x})$, whose direct estimation is challenging in the case of large-dimensional $\boldsymbol{x}$, to simpler tasks of estimating projected univariate functions $\{f_k(\boldsymbol{a}_k^\top \boldsymbol{x})\}_{k=1}^K$.

Estimation of $\{f_k(\boldsymbol{a}_k^\top \boldsymbol{x})\}_{k=1}^K$ can be carried out iteratively based on the relation $r_k(\boldsymbol{x}) = r_{k-1}(\boldsymbol{x}) f_k(\boldsymbol{a}_k^\top \boldsymbol{x})$ for $k \in$

$\{1, ..., K\}$, where $r_0(\boldsymbol{x}) \equiv 1$. Specifically, let $\mathbb{E}_q[\cdot]$ and $\mathbb{E}_p[\cdot]$ denote the expectation taken with respect to (w.r.t.) the probability densities $q(\cdot)$ and $p(\cdot)$, respectively. Suppose that the first $k-1$ terms are given, that is $r_{k-1}(\boldsymbol{x}) = \prod_{m=1}^{k-1} f_m(\boldsymbol{a}_m^\top \boldsymbol{x})$ is given, the $k$th projection direction $\boldsymbol{a}_k$ and the $k$th pursuit function $f_k$ can be determined by minimizing the $L^2$-distance, $\mathbb{E}_q\{[r^*(\boldsymbol{x}) - r_{k-1}(\boldsymbol{x})f(\boldsymbol{a}^\top \boldsymbol{x})]^2\}$. We show in Appendix E that minimizing the $L^2$-distance w.r.t. $f$ or $\boldsymbol{a}$ is equivalent to minimizing

$$
\begin{aligned}
H(f, \boldsymbol{a}) :=& \mathbb{E}_q[r_{k-1}^2(\boldsymbol{x})f^2(\boldsymbol{a}^\top \boldsymbol{x})] \\
& - 2\mathbb{E}_p[r_{k-1}(\boldsymbol{x})f(\boldsymbol{a}^\top \boldsymbol{x})].
\end{aligned} \tag{2}
$$

For model identification, we assume $\boldsymbol{a}_k \in \mathbb{S}_d^+$ as in Wang & Yang (2009), where $\mathbb{S}_d^+$ is the $d$ dimensional upper unit hemisphere, i.e. $\mathbb{S}_d^+ = \{\boldsymbol{a} = (a_1, \ldots, a_d) \in \mathbb{R}^d, \|\boldsymbol{a}\| = 1, a_1 > 0\}$, and $\|\cdot\|$ denotes the Euclidean norm, i.e. for a vector $\boldsymbol{v}$, $\|\boldsymbol{v}\| := \sqrt{\boldsymbol{v}^\top \boldsymbol{v}}$.

### 4.2. Estimation Procedure

Since the function space for searching $f_k$ is infinitely dimensional, direct optimization based on the sample analogue of (2) is impossible. We consider seeking the estimator of the pursuit function $f_k(z)$, for $z \in \{\boldsymbol{a}^\top \boldsymbol{x} : \boldsymbol{a} \in \mathbb{S}_d^+, \boldsymbol{x} \in \mathcal{X}\}$, from the linear sieve class:

$$
\mathcal{F}_{J_k} := \left\{ \boldsymbol{\beta}^\top \boldsymbol{\Phi}_k(z) = \sum_{j=1}^{J_k} \beta_j \phi_j(z), \ J_k \in \mathbb{N} \right\},
$$

where $\{\phi_j\}_{j=1}^{J_k}$ is a sequence of univariate basis, $\boldsymbol{\Phi}_k(z) = (\phi_1(z), \ldots, \phi_{J_k}(z))^\top$, and $\boldsymbol{\beta} = (\beta_1, ..., \beta_{J_k})^\top$ is the approximation coefficients. The rationale is that any continuous function can be approximated arbitrarily well by a linear sieve in $\mathcal{F}_{J_k}$ as $J_k$ goes to infinity (see e.g., Chen, 2007b).

For $k = 1, \ldots, K$, based on (2) and the linear sieve class, we define the estimator in the $k$th iteration of $(f_k, \boldsymbol{a}_k)$ by

$$
\hat{f}_k(z) := \hat{\boldsymbol{\beta}}_k^\top \boldsymbol{\Phi}_k(z), \ (\hat{\boldsymbol{a}}_k, \hat{\boldsymbol{\beta}}_k) := \arg\min_{\boldsymbol{a}, \boldsymbol{\beta}} \hat{\mathcal{L}}_k(\boldsymbol{a}, \boldsymbol{\beta}; \lambda) \tag{3}
$$

where

$$
\begin{aligned}
\hat{\mathcal{L}}_k(\boldsymbol{a}, \boldsymbol{\beta}; \lambda) :=& \frac{1}{n_q} \sum_{i=1}^{n_q} \hat{r}_{k-1}^2(\boldsymbol{x}_i^q) \cdot [\boldsymbol{\beta}^\top \boldsymbol{\Phi}_k(\boldsymbol{a}^\top \boldsymbol{x}_i^q)]^2 \\
& - \frac{2}{n_p} \sum_{i=1}^{n_p} \hat{r}_{k-1}(\boldsymbol{x}_i^p) \cdot \boldsymbol{\beta}^\top \boldsymbol{\Phi}_k(\boldsymbol{a}^\top \boldsymbol{x}_i^p) + \lambda \|\boldsymbol{\beta}\|^2
\end{aligned} \tag{4}
$$

is the regularized empirical loss, where $\hat{r}_{k-1}(\boldsymbol{x}) := \prod_{m=0}^{k-1} \hat{f}_m(\hat{\boldsymbol{a}}_m^\top \boldsymbol{x})$, $\|\boldsymbol{\beta}\|^2 := \boldsymbol{\beta}^\top \boldsymbol{\beta}$ is the $\ell_2$-penalty on the model complexity, and $\lambda > 0$ is a tuning parameter. Here, we use the $\ell_2$-penalty to circumvent the over-fitting problem, while maintaining a closed-form solution in $\boldsymbol{\beta}$ for the problem (4) when keeping $\boldsymbol{a}$ fixed.

**Proposition 4.1.** *For every fixed $\boldsymbol{a} \in \mathbb{R}^d$, we have*

$$
\begin{aligned}
\hat{\boldsymbol{\beta}}_k(\boldsymbol{a}) :=& \arg\min_{\boldsymbol{\beta}} \hat{\mathcal{L}}_k(\boldsymbol{a}, \boldsymbol{\beta}; \lambda) \\
=& \left[ \frac{\boldsymbol{Z}_k(\boldsymbol{a})^\top \boldsymbol{Z}_k(\boldsymbol{a})}{n_q} + \lambda \boldsymbol{I}_{J_k} \right]^{-1} \left[ \frac{\boldsymbol{W}_k(\boldsymbol{a})}{n_p} \right],
\end{aligned}
$$

*where*

$$
\boldsymbol{Z}_k(\boldsymbol{a}) := \{\hat{r}_{k-1}(\boldsymbol{x}_i^q) \boldsymbol{\Phi}_k(\boldsymbol{a}^\top \boldsymbol{x}_i^q)\}_{i=1}^{n_q} \in \mathbb{R}^{n_q \times J_k},
$$

$$
\boldsymbol{W}_k(\boldsymbol{a}) := \sum_{i=1}^{n_p} \hat{r}_{k-1}(\boldsymbol{x}_i^p) \boldsymbol{\Phi}_k(\boldsymbol{a}_k^\top \boldsymbol{x}_i^p) \in \mathbb{R}^{J_k},
$$

*and $\boldsymbol{I}_{J_k}$ is an identity matrix of size $J_k \times J_k$.*

*Proof.* Using above notation, $\hat{\mathcal{L}}_k(\boldsymbol{a}, \boldsymbol{\beta}; \lambda)$ can be written as a quadratic function of $\boldsymbol{\beta}$:

$$
\begin{aligned}
\hat{\mathcal{L}}_k(\boldsymbol{a}, \boldsymbol{\beta}; \lambda) =& \frac{1}{n_q} \sum_{i=1}^{n_q} [\boldsymbol{\beta}^\top \hat{r}_{k-1}(\boldsymbol{x}_i^q) \boldsymbol{\Phi}_k(\boldsymbol{a}^\top \boldsymbol{x}_i^q)]^2 \\
& - \frac{2}{n_p} \boldsymbol{\beta}^\top \left[ \sum_{i=1}^{n_p} \hat{r}_{k-1}(\boldsymbol{x}_i^p) \cdot \boldsymbol{\Phi}_k(\boldsymbol{a}^\top \boldsymbol{x}_i^p) \right] + \lambda \boldsymbol{\beta}^\top \boldsymbol{\beta} \\
=& \boldsymbol{\beta}^\top \left[ \frac{\boldsymbol{Z}_k^\top(\boldsymbol{a}) \boldsymbol{Z}_k(\boldsymbol{a})}{n_q} + \lambda \boldsymbol{I}_{J_k} \right] \boldsymbol{\beta} - \frac{2}{n_p} \boldsymbol{\beta}^\top \boldsymbol{W}_k(\boldsymbol{a}).
\end{aligned}
$$

Differentiating it with respect to $\boldsymbol{\beta}$ and setting the derivative to zero give the desired result. $\square$

By Proposition 4.1 and (3), the estimators of $f_{\boldsymbol{a}}(z)$ and $\boldsymbol{a}_k$ can be represented respectively as

$$
\hat{f}_{\boldsymbol{a}, k}(z) := \hat{\boldsymbol{\beta}}_k(\boldsymbol{a})^\top \boldsymbol{\Phi}_k(z), \tag{5}
$$

and

$$
\begin{aligned}
\hat{\boldsymbol{a}}_k =& \arg\min_{\boldsymbol{a} \in \mathbb{R}^d} \hat{\mathcal{L}}_k(\boldsymbol{a}, \hat{\boldsymbol{\beta}}_k(\boldsymbol{a}); \lambda) \\
=& \arg\min_{\boldsymbol{a} \in \mathbb{R}^d} \frac{1}{2n_q} \|\boldsymbol{Z}_k(\boldsymbol{a}) \hat{\boldsymbol{\beta}}_k(\boldsymbol{a})\|^2 \\
& - \frac{1}{n_p} \hat{\boldsymbol{\beta}}_k(\boldsymbol{a})^\top \boldsymbol{W}_k(\boldsymbol{a}) + \lambda \hat{\boldsymbol{\beta}}_k(\boldsymbol{a})^\top \hat{\boldsymbol{\beta}}_k(\boldsymbol{a}).
\end{aligned}
$$

Then, we have $\hat{f}_k(z) = \hat{f}_{\hat{\boldsymbol{a}}_k, k}(z)$, the estimators of $\{r_k(\boldsymbol{x})\}_{k=1}^K$ are defined by

$$
\hat{r}_k(\boldsymbol{x}) = \hat{r}_{k-1}(\boldsymbol{x}) \hat{f}_k(\hat{\boldsymbol{a}}_k^\top \boldsymbol{x}), \ k \in \{1, ..., K\},
$$

and the estimator of the density ratio function $r^*(\boldsymbol{x})$ is given by $\hat{r}_K(\boldsymbol{x})$. In practice, to ensure non-negativity of the estimated density ratio function, we truncate the negative values of $\hat{f}_k$ to the minimum positive estimated values for every iteration. The complete process of our approach given tuned hyper-parameters is summarized in Algorithm 1.

In the following theorem, we focus on each iteration $k$ and show that, given the estimate $\hat{r}_{k-1}(\boldsymbol{x})$ of $r_{k-1}(\boldsymbol{x})$, as the sample sizes $n_p, n_q \to \infty$, $\hat{f}_{\boldsymbol{a},k}$ and $\hat{\boldsymbol{a}}_k$ converge respectively to

$$f_{\boldsymbol{a},k} := \underset{f}{\arg\min}\, H(f,\boldsymbol{a}), \qquad (6)$$

and

$$\boldsymbol{a}_k := \underset{\boldsymbol{a}}{\arg\min}\, H(f_{\boldsymbol{a},k},\boldsymbol{a}). \qquad (7)$$

**Theorem 4.2.** *Suppose that Assumptions F.1 to F.4 in Appendix F hold. Then, for each $k = 1,\ldots,K$, if $n_p^{-1} \sum_{i=1}^{n_p} \{\hat{r}_{k-1}(\boldsymbol{x}_i^p) - r_{k-1}(\boldsymbol{x}_i^p)\}^2 = O_p(\xi_{n,k-1})$, we have*

$$\sup_{\boldsymbol{a}\in\mathcal{A}}\sup_{z\in\mathcal{Z}} \left|\hat{f}_{\boldsymbol{a},k}(z) - f_{\boldsymbol{a},k}(z)\right|$$

$$= O_p\Bigg(J_k^{-s}\zeta_0(J_k) + \sqrt{\xi_{n,k-1}}\,\zeta_0(J_k)^2$$

$$+ \frac{\sqrt{J_k}\zeta_0(J_k)^2}{\sqrt{n_q \wedge n_p}}\Bigg), \qquad (8)$$

*and*

$$\|\hat{\boldsymbol{a}}_k - \boldsymbol{a}_k\| = O_p\Bigg(\Bigg\{J_k^{-(s-1)} + \sqrt{\xi_{n,k-1}} \qquad (9)$$

$$+ \frac{\sqrt{J_k}}{\sqrt{n_q \wedge n_p}}\Bigg\} \cdot \sqrt{\tilde{\zeta}_1(J_k)}\Bigg),$$

*where $\mathcal{A} \subset \mathbb{S}_d^+$ is a compact set containing $\boldsymbol{a}_k$, $\mathcal{Z} := \{\boldsymbol{a}^\top\boldsymbol{x} : \boldsymbol{a} \in \mathcal{A} \text{ and } \boldsymbol{x} \in \mathcal{X}\}$, $s$ denotes the number of continuous derivatives that $f_{\boldsymbol{a},k}(z)$ possesses w.r.t. $z \in \mathcal{Z}$ for any $\boldsymbol{a} \in \mathcal{A}$, $\zeta_0(J_k)$ is a sequence of constants such that $\sup_{z\in\mathcal{Z}}\|\boldsymbol{\Phi}_k(z)\| \le \zeta_0(J_k)$, $\tilde{\zeta}_1(J_k)$ is a sequence of constants such that the maximum eigenvalue of $\mathbb{E}[\boldsymbol{\Phi}_k^{(1)}(\boldsymbol{a}^\top\boldsymbol{x})\boldsymbol{\Phi}_k^{(1)}(\boldsymbol{a}^\top\boldsymbol{x})^\top]$ is bounded by $\tilde{\zeta}_1(J_k)$ uniformly in $\boldsymbol{a} \in \mathcal{A}$, and $n_q \wedge n_p = \min(n_q,n_p)$.*

*Finally, we have*

$$\sup_{\boldsymbol{x}\in\mathcal{X}} |\hat{r}_K(\boldsymbol{x}) - r_K(\boldsymbol{x})| = O_p\Bigg(\sum_{\ell=1}^{K}\Bigg[\Bigg\{J_\ell^{-(s-1)} + \sqrt{\frac{J_\ell}{n_q \wedge n_p}}\Bigg\}$$

$$\times \prod_{i=\ell}^{K}\Big\{\sqrt{\tilde{\zeta}_1(J_i)} \vee \zeta_0^2(J_i)\Big\}\Bigg]\Bigg),$$

*where $\tilde{\zeta}_1(J_k) \vee \zeta_0(J_k) = \max\{\tilde{\zeta}_1(J_k),\zeta_0(J_k)\}$.*

The proof is available in Appendix F. Assumption F.1 imposes compactness conditions on $\mathcal{X}$ and parameter spaces $\mathcal{A}$. Assumption F.2 requires $f_{\boldsymbol{a},k}(z)$ to be bounded and bounded away from 0, $s$-times continuously differentiable w.r.t $z \in \mathcal{Z}$ and has a bounded derivative w.r.t. $\boldsymbol{a} \in \mathcal{A}$, which are some common regularity conditions in the literature. Assumption F.3 excludes near multicollinearity among the basis functions, $\boldsymbol{\Phi}_k(z)$, and regulates the rate of $J_k$ relative

to $n_p$ and $n_q$ to guarantee the consistency of our estimators. Such conditions are standard in sieve regression. Assumption F.4 requires the Hessian matrix of the sieve approximation of $H(f,\boldsymbol{a})$ to be positive definite at its minimum w.r.t. $\boldsymbol{a}$, which is met when the minimum is in the interior of $\mathcal{A}$.

---

**Algorithm 1** ppDRE

---

1: **Input**: samples $\{\boldsymbol{x}_i^p\}_{i=1}^{n_p}$ and $\{\boldsymbol{x}_i^q\}_{i=1}^{n_q}$; number of basis functions $J_k$, learning rate $\delta$, ridge penalty parameter $\lambda$, maximum steps $K$.
2: **Initialize**: $\hat{r}_0(\boldsymbol{x}) \equiv 1$
3: **for** $k = 1$ to $K$ **do**
4:     Initialize randomly $\boldsymbol{a}^{(0)},\boldsymbol{\gamma}^{(0)}$, and set $t = 1$.
5:     **while** not converge **do**
6:         Calculate

$$\boldsymbol{z}_k^{(t)} \leftarrow \{\hat{r}_{k-1}(\boldsymbol{x}_i^q)\boldsymbol{\Phi}_k(\boldsymbol{a}^{(t-1)\top}\boldsymbol{x}_i^q;\boldsymbol{\gamma}^{(t-1)})\}_{i=1}^{n_q},$$

$$\boldsymbol{w}_k^{(t)} \leftarrow \sum_{i=1}^{n_p} \hat{r}_{k-1}(\boldsymbol{x}_i^p)\boldsymbol{\Phi}_k(\boldsymbol{a}_k^{(t-1)\top}\boldsymbol{x}_i^p;\boldsymbol{\gamma}^{(t-1)}).$$

7:         Update

$$\boldsymbol{\beta}_k^{(t)} \leftarrow \left[\frac{1}{n_q}\boldsymbol{z}_k^{(t)\top}\boldsymbol{z}_k^{(t)} + \lambda\boldsymbol{I}_{J_k}\right]^{-1}\left[\frac{1}{n_p}\boldsymbol{w}_k^{(t)}\right].$$

8:         Evaluate the loss function

$$\hat{\mathcal{L}}_k^{(t)} \leftarrow \frac{1}{n_q}\|\boldsymbol{z}_k^{(t)}\boldsymbol{\beta}_k^{(t)}\|^2 - \frac{2}{n_p}\boldsymbol{\beta}_k^{(t)\top}\boldsymbol{w}_k^{(t)} + \lambda\boldsymbol{\beta}_k^{(t)\top}\boldsymbol{\beta}_k^{(t)}.$$

9:         Update $(\boldsymbol{a}_k^{(t)},\boldsymbol{\gamma}_k^{(t)})$ with stochastic gradient descent algorithm, such as Adam, with learning rate $\delta$.
10:         Update $t \leftarrow t+1$
11:     **end while**
12:     Update $\hat{r}_k(\boldsymbol{x}) \leftarrow \hat{r}_{k-1}(\boldsymbol{x}) \cdot \boldsymbol{\beta}_k^{(t)\top}\boldsymbol{\Phi}_k(\boldsymbol{a}_k^{(t)\top}\boldsymbol{x};\boldsymbol{\gamma}_k^{(t)})$
13: **end for**
14: **return** $\hat{r}_K(\boldsymbol{x})$.

---

*Remark* 4.3 (Selection of Tuning Parameters). In practice, we suggest using cross-validation (CV) to determine the number of projections $K$ as well as other tuning parameters. Specifically, in experiments and applications, we adopt the Gaussian basis $\phi_j(z) = \phi(z;\gamma_j) = \exp\{-(z-\gamma_j)^2/2\}$, where $\gamma_j$ is the location parameter of the Gaussian basis. For each $k \in \{1,\ldots,K\}$, we determine $\{\hat{\gamma}_j\}_{j=1}^{J_k}$, together with $\hat{\boldsymbol{a}}_k$ and $\hat{\boldsymbol{\beta}}_k$ jointly by minimizing the loss function $\hat{\mathcal{L}}_k$. We monitor the minimal value of the loss function $\hat{\mathcal{L}}_K$ in a validation set for a gradually growing $K$, and determine the optimal $K$ until no further improvement is observed.

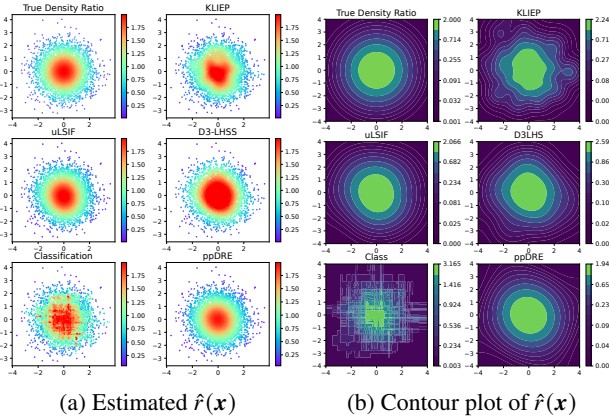

(a) Estimated $\hat{r}(x)$      (b) Contour plot of $\hat{r}(x)$

*Figure 1.* 2-D DRE Experiment. The warmer color represents a higher estimated density ratio value. The top-left plot in each panel shows the true density ratio, while the remaining plots illustrate the estimates using various methods.

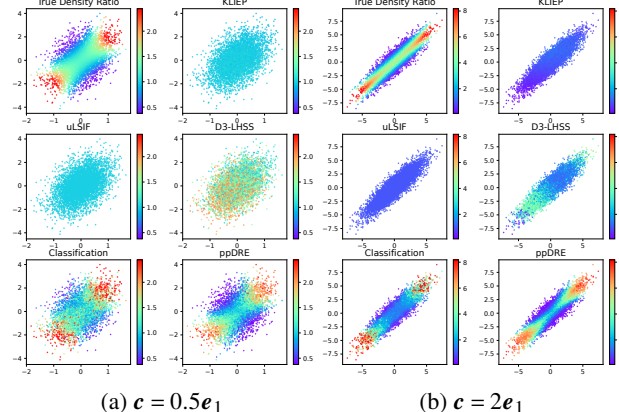

(a) $c = 0.5e_1$      (b) $c = 2e_1$

*Figure 2.* Stabilized weights estimation ($d_X = 10$). The y-axis represents the treatment $t$ and the x-axis represents $c^\top x$. Each point represents a sample data point, and the color indicates the magnitude of the estimated density ratio. The top-left plot in each panel shows the true density ratio, while other plots illustrate estimates using various methods.

## 5. Experiments and Applications

In this section, we compare our proposed projection pursuit density ratio estimation (ppDRE) method with existing alternatives using experimental and real-world data. The compared methods include two classical linear sieve methods, KLIEP (Sugiyama et al., 2007) and uLSIF (Kanamori et al., 2009), a linear sieve method with dimension reduction, D³-LHSS (Sugiyama et al., 2011), a probabilistic classification approach (Qin, 1998; Bickel et al., 2007) based on machine learning classifiers, two latest baselines, fDRE (Choi et al., 2021) and RRND (Nguyen et al., 2024), and a neural network-based density ratio estimation (nnDRE) method, which is a variant of our framework in the sense that it replaces the projection pursuit with a feedforward neural network to model the density ratio. Detailed descriptions of these baselines and implementation specifics are provided in Appendix A and Appendix B.

### 5.1. 2-D DRE Experiment

We first consider a toy example where $p(x) = \mathcal{N}(\mathbf{0}_d, I_d)$ and $q(x) = \mathcal{N}(\mathbf{0}_d, 2I_d)$, where $\mathcal{N}(\mu, \Sigma)$ denotes a multivariate Gaussian probability density with mean $\mu$ and covariance matrix $\Sigma$, and $I_d$ denotes an identity matrix of size $d$. We consider a low-dimensional case with $d = 2$, which facilitates visualization of the estimated results. The sample sizes are $n_p = n_q = 5000$. Estimates of this density ratio are shown in Figure 1a, and the corresponding contour plot is presented in Figure 1b. These figures demonstrate that the proposed ppDRE method and the uLSIF method significantly outperform the other baseline approaches, yielding estimates that closely align with the true density ratio.

### 5.2. Application in Causal Inference

We apply our proposed ppDRE method to estimate continuous treatment effects in the framework of Ai et al. (2021). Let $T \in \mathbb{R}$ denote the observed continuous treatment status variable, with a PDF $f_T(t)$. Let $Y^*(t)$ denote the potential response when treatment $t$ is assigned, and let $Y = Y^*(T)$ denote the observed response. Let $X \in \mathbb{R}^{d_X}$ denote a vector of observable covariates. To identify the causal effect, we assume the unconfoundedness condition that $Y^*(t)$ and $T$ are conditionally independent given $X$. We assume a parametric model $g(t; \theta^*)$, called the *general dose-response function*, for the potential outcome $Y^*(t)$:

$$\theta^* := \operatorname*{argmin}_{\theta \in \mathbb{R}^p} \int \mathbb{E}[L(Y^*(t) - g(t; \theta))] f_T(t) dt$$
$$= \operatorname*{argmin}_{\theta \in \mathbb{R}^p} \mathbb{E}[\pi_0(T, X) L(Y - g(T; \theta))], \quad (10)$$

where $L(\cdot)$ is a user-specified loss function, the second equality holds by the unconfoundedness condition, $\pi_0(t, x)$ is called the *stabilized weights*, defined by

$$\pi_0(t, x) := \frac{f_T(t) f_X(x)}{f_{T,X}(t, x)} = \frac{f_T(t)}{f_{T|X}(t|x)},$$

where $f_{T,X}$ is the joint PDF of $T$ and $X$, $f_{T|X}$ is the conditional PDF of $T$ given $X$, $f_T$ and $f_X$ are marginal PDFs of $T$ and $X$, respectively.

The causal model (10) encompasses a variety of continuous treatment effect parameters of interest. For example, with $\mathcal{L}(v) = v^2$, model (10) gives $g(t; \theta^*) = \mathbb{E}\{Y^*(t)\}$, the average dose-response function (ADRF). With $\mathcal{L}(v) = v\{\tau - I(v \le 0)\}$ for some $\tau \in (0, 1)$, the model (10) gives

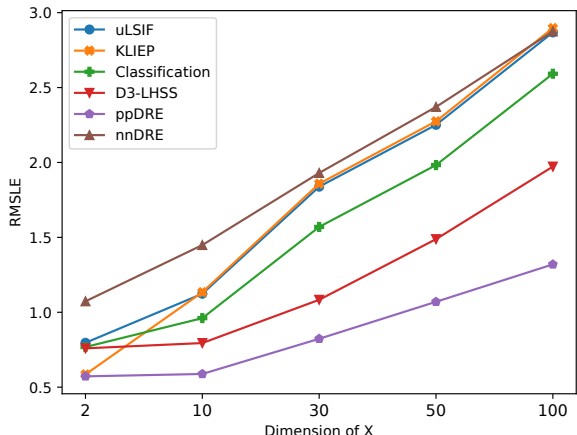

*Figure 3.* Average RMSLE over 50 replicates for stabilized weights estimation with varying dimensions of covariates.

the $\tau$th quantile dose-response function (QDRF) $g(t;\theta^*) = F_{Y^*(t)}^{-1}(\tau) = \inf\{q : \mathbb{P}\{Y^*(t) \geq q\} \leq \tau\}$.

Note that, by definition, the stabilized weights $\pi_0(t, x)$ can be viewed as a density ratio. We compare our method with different estimators of $\pi_0(t, x)$ based on simulated data sets in Section 5.2.1. In Section 5.2.2, we investigate the application of DRE in both ADRF and QDRF analysis based on a semi-synthetic data set.

### 5.2.1. STABILIZED WEIGHTS ESTIMATION

In this simulation study, we design the treatment assignment model as $T_i = c^\top X_i + \epsilon_i$, where $c \in \mathbb{R}^{d_X}$ is specified below, $X_i \overset{\text{i.i.d.}}{\sim} \mathcal{N}(0_{d_X}, I_{d_X})$ and $\epsilon_i \overset{\text{i.i.d.}}{\sim} N(0, 1)$. In all simulation scenarios, the sample size is fixed to be $n = 5000$.

We first investigate the visual performance of the estimators for a fixed dimension of covariates, $d_X = 10$. To facilitate visualization, we consider two choices of the coefficient: $c = 0.5e_1$ and $c = 2e_1$, respectively, where $e_1$ is a vector with 1 in the first component while others are zeros. In both scenarios, only the first component of $x$ affects the treatment assignment. The estimated density ratios are visualized in Figure 2, where the y-axis represents the treatment $t$ and the x-axis is $c^\top x$. In both scenarios, the density ratio estimates produced by our ppDRE method show the closest alignment with the true density ratios.

We further investigate the estimators' performances in terms of the root mean squared logarithmic error (RMSLE) defined in Appendix C.1, for varying dimensions of covariates $d_X \in \{2, 10, 30, 50, 100\}$. We set $c = 0.5 \cdot 1_{d_X}$ to be a $d_X$-dimensional vector whose components are all 0.5. Figure 3 presents the average RMSLE values of various estimators over 50 replications. The RMSLE of all estimators increases as the dimension $d_X$ grows, but our proposed

ppDRE method consistently outperforms its competitors. Moreover, the advantage of ppDRE is more pronounced for a larger dimension of covariates. Additional experimental results can be found in Appendix C.1.

### 5.2.2. DOSE RESPONSE FUNCTION ESTIMATION

This section investigates the performance of estimating both ADRF and QDRF using the semi-synthetic variant of the Infant Health and Development Program (IHDP) dataset. The original IHDP dataset (Hill, 2011) consists of 747 observations, each characterized by $d_X = 25$ covariates. Following Nie et al. (2020) and Gao et al. (2023), we generate the semi-synthetic IHDP-continuous dataset by leveraging the real-world covariates from the original IHDP dataset to simulate continuous dosages and responses. A detailed description of the data generation process can be found in Appendix D.1. Specifically, the randomly assigned treatment $T$ is generated from $X$ by (12), and the potential outcome $Y^*(t) = h(t, X) + 0.5\epsilon$, where $h(t, X)$ is defined in (13) and $\epsilon \sim \mathcal{N}(0, 1)$. Then the true ADRF is $g^*(t) = \mathbb{E}_X[h(t, X)]$, and the true $\tau$-th QDRF is $g^*(t) = \inf[q : \mathbb{P}\{h(t, X) + 0.5\epsilon \geq q\} \leq \tau]$.

We estimate ADRF and QDRFs at various quantile levels $\tau = \{0.1, 0.25, 0.5, 0.75, 0.9\}$ using a parametric model $g(t; \theta) := \theta_0 + \theta_1 t + \theta_2 t^2 + \theta_3 t^3 + \theta_4 t^4$, and obtain $\hat{\theta}$ by solving the empirical version of the optimization problem (10). To measure the accuracy of the estimated dose-response functions across the full dosage range, we compute the average squared error (ASE):

$$\text{ASE} = \frac{1}{n} \sum_{i=1}^{n} \left( g(T_i; \hat{\theta}) - g^*(T_i) \right)^2.$$

The mean and standard error of the ASE computed based on 100 replicates are reported in Table 1. The numerical results indicate that our ppDRE method attains the lowest mean ASE in most cases, demonstrating its superior performance compared to the other alternatives.

### 5.3. Application in Mutual Information Estimation

The mutual information (MI) is a measure of the mutual dependence between two continuous random vectors $U$ and $V$, which is defined by

$$\text{MI}_{U,V} = \iint f_{U,V}(u, v) \cdot \log \frac{f_{U,V}(u, v)}{f_U(u) f_V(v)} \, du \, dv.$$

Given a sample from $f_{U,V}(u, v)$, we can create another sample following the distribution $f_U(u) f_V(v)$ by permuting the $v$ vectors across the dataset. This enables the application of DRE methods to estimate the MI by

$$\widehat{\text{MI}}_{U,V} = \frac{1}{n_p} \sum_{i=1}^{n_p} \log \hat{r}(u_i, v_i),$$

Table 1. ASE in dose response function estimation with IHDP-continuous dataset.

| Method | ARDF | QDRF | | | | |
|---|---|---|---|---|---|---|
| | | $\tau = 0.1$ | $\tau = 0.25$ | $\tau = 0.5$ | $\tau = 0.75$ | $\tau = 0.9$ |
| uLSIF | 0.112(0.022) | 0.338(0.058) | 0.130(0.027) | 0.045(0.016) | 0.090(0.022) | 0.278(0.064) |
| KLIEP | 0.112(0.022) | 0.339(0.059) | 0.130(0.027) | 0.045(0.016) | 0.089(0.022) | 0.278(0.064) |
| Classification | 0.107(0.028) | 0.341(0.065) | 0.130(0.032) | 0.045(0.018) | 0.088(0.025) | 0.276(0.066) |
| D³-LHSS | 0.110(0.024) | 0.336(0.058) | 0.128(0.030) | 0.045(0.016) | 0.088(0.022) | 0.279(0.068) |
| fDRE | 0.106(0.026) | 0.331(0.056) | 0.127(0.029) | 0.045(0.016) | 0.088(0.021) | 0.267(0.061) |
| RRND | 0.117(0.022) | **0.307**(0.053) | 0.125(0.026) | 0.043(0.015) | 0.090(0.021) | 0.408(0.082) |
| nnDRE | 0.107(0.022) | 0.333(0.059) | 0.126(0.027) | 0.043(0.015) | 0.089(0.021) | 0.297(0.066) |
| ppDRE | **0.091**(0.025) | 0.338(0.066) | **0.124**(0.029) | **0.041**(0.016) | **0.079**(0.022) | **0.266**(0.060) |

Table 2. MAE in mutual information estimation for varying dimension $p$ and correlation coefficient $\rho$.

| Method | $p = 2$ | | $p = 10$ | | $p = 20$ | |
|---|---|---|---|---|---|---|
| | $\rho = 0.2$ | $\rho = 0.8$ | $\rho = 0.2$ | $\rho = 0.8$ | $\rho = 0.2$ | $\rho = 0.8$ |
| uLSIF | 0.044(0.001) | 0.573(0.015) | 0.200(0.008) | 5.093(0.031) | 0.415(0.023) | 10.242(0.088) |
| KLIEP | 0.044(0.000) | 1.099(0.011) | 0.204(0.007) | 4.885(0.019) | 0.516(0.032) | 9.991(0.091) |
| Classification | 0.270(0.006) | 0.169(0.019) | 0.381(0.015) | 2.579(0.062) | 0.272(0.021) | 7.185(0.044) |
| D³-LHSS | 0.339(0.077) | 0.248(0.162) | 0.282(0.084) | 4.080(0.201) | 0.164(0.132) | 9.241(0.198) |
| fDRE | 0.043(0.000) | 0.907(0.048) | 0.199(0.007) | 4.283(0.014) | 0.487(0.036) | 8.710(0.059) |
| RRND | 0.061(0.001) | 0.926(0.006) | 0.200(0.008) | 5.092(0.031) | 0.414(0.022) | 10.241(0.089) |
| nnDRE | **0.038**(0.037) | 1.064(0.275) | 1.183(1.725) | 3.838(1.912) | 1.671(2.202) | 8.629(2.577) |
| ppDRE | 0.051(0.044) | **0.069**(0.015) | **0.164**(0.038) | **0.272**(0.183) | **0.074**(0.098) | **2.966**(1.089) |

with $p(\boldsymbol{u}, \boldsymbol{v}) = f_{\boldsymbol{U},\boldsymbol{V}}(\boldsymbol{u}, \boldsymbol{v})$ and $q(\boldsymbol{u}, \boldsymbol{v}) = f_{\boldsymbol{U}}(\boldsymbol{u}) f_{\boldsymbol{V}}(\boldsymbol{v})$ for $r^*(\boldsymbol{u}, \boldsymbol{v}) = p(\boldsymbol{u}, \boldsymbol{v})/q(\boldsymbol{u}, \boldsymbol{v})$.

We adopt the experimental setting in Belghazi et al. (2018); Rhodes et al. (2020); Choi et al. (2022). Specifically, we consider two standard multivariate Gaussian random vectors, $\boldsymbol{U} = (U_1, ..., U_p)^\top \in \mathbb{R}^p$ and $\boldsymbol{V} = (V_1, ..., V_p)^\top \in \mathbb{R}^p$, with component-wise correlation, $\mathrm{corr}(U_i, V_j) = \delta_{ij}\rho$, where $\rho \in (-1, 1)$ and $\delta_{ij}$ is Kronecker's delta. Performance is measured by the mean absolute error, $\mathrm{MAE} := |\widehat{\mathrm{MI}}_{\boldsymbol{U},\boldsymbol{V}} - \mathrm{MI}_{\boldsymbol{U},\boldsymbol{V}}|$. We consider three replicates with sample size $n = 5000$. The MAE averaged over three runs for various values of $p$ and $\rho$ are reported in Table 2. Overall, our proposed ppDRE method demonstrates superior or comparable performance in all circumstances, indicating its robustness and effectiveness in mutual information estimation.

## 5.4. Application in Covariate Shift Adaptation

Covariate shift refers to the change in the distribution of the input variables in the training and the test data sets, i.e. $p_{\mathrm{te}}^*(\boldsymbol{x}) \neq p_{\mathrm{tr}}^*(\boldsymbol{x})$. In the case of covariate shift, learning a

parameter, $\boldsymbol{\theta}$ in a model $f(\boldsymbol{x}; \boldsymbol{\theta})$ regarding the probability distribution of $y$ given $\boldsymbol{x}$, using standard learning techniques such as empirical risk minimization (ERM) can become biased (Sugiyama et al., 2012). To mitigate this issue, the importance-weighted ERM is widely used. The core idea is to re-weight the training samples in order to learn a model that minimizes the loss on the test dataset:

$$\mathbb{E}_{(\boldsymbol{x},y) \sim p_{\mathrm{te}}^*(\boldsymbol{x},y)}[L(f(\boldsymbol{x}; \boldsymbol{\theta}), y)]$$
$$= \mathbb{E}_{(\boldsymbol{x},y) \sim p_{\mathrm{tr}}^*(\boldsymbol{x},y)} \left[ \frac{p_{\mathrm{te}}^*(\boldsymbol{x})}{p_{\mathrm{tr}}^*(\boldsymbol{x})} L(f(\boldsymbol{x}; \boldsymbol{\theta}), y) \right],$$

where $L(\cdot)$ is a user-specified loss function, and the density ratio $r^*(\boldsymbol{x}) = p_{\mathrm{te}}^*(\boldsymbol{x})/p_{\mathrm{tr}}^*(\boldsymbol{x})$ is referred to as the *importance*, acting as an adjusting weight during the training process.

To simulate the covariate shift setting, we adopt a biased sampling scheme as described by Stojanov et al. (2019), on various classical benchmark regression datasets. In this setup, we introduce a sample selection variable $s \in \{0, 1\}$, and the probability of a sample being selected for the train-

*Table 3.* NMSE under covariate shift adaptation on various benchmark datasets.

| Method | Abalone | Billboard Spotify | Cancer Mortality | Computer Activity | Diamond Prices |
|---|---|---|---|---|---|
| Unweighted | 0.544(0.057) | 0.582(0.044) | 0.704(0.069) | 0.278(0.050) | 0.231(0.082) |
| uLSIF | 0.525(0.057) | 0.582(0.044) | 0.704(0.069) | 0.668(0.360) | 0.239(0.068) |
| KLIEP | 0.531(0.055) | 0.565(0.028) | 0.696(0.058) | 0.302(0.049) | 0.228(0.080) |
| Classification | 0.535(0.049) | 0.579(0.042) | 0.691(0.067) | 0.273(0.051) | 0.183(0.056) |
| $D^3$-LHSS | 0.520(0.054) | 0.564(0.031) | 0.672(0.066) | 0.318(0.052) | 0.220(0.070) |
| fDRE | 0.586(0.136) | 0.615(0.084) | 0.664(0.064) | 0.294(0.053) | 0.233(0.078) |
| RRND | 0.574(0.144) | 0.616(0.087) | **0.655**(0.069) | 0.279(0.051) | 0.232(0.082) |
| nnDRE | 0.806(0.829) | 0.822(0.861) | 0.660(0.063) | 0.301(0.033) | 0.360(0.429) |
| ppDRE | **0.514**(0.049) | **0.559**(0.026) | 0.661(0.048) | **0.199**(0.019) | **0.133**(0.024) |

ing set is given by:

$$\mathbb{P}(s = 1 | \boldsymbol{x}) = \frac{e^v}{(1 + e^v)}, \quad \text{with } v = 4 \cdot \frac{\omega^\top (\boldsymbol{x} - \bar{\boldsymbol{x}})}{\sigma_{\omega^\top (\boldsymbol{x} - \bar{\boldsymbol{x}})}},$$

where $\bar{\boldsymbol{x}}$ is the sample mean of the covariates, $\omega$ is a random projection vector uniformly selected from the interval $[-1, 1]^d$ and $\sigma_{\omega^\top (\boldsymbol{x} - \bar{\boldsymbol{x}})}$ is the standard deviation. Subsequently, we learn a kernel ridge regression model by minimizing the importance-weighted empirical risk:

$$\hat{\boldsymbol{\theta}} = \operatorname*{argmin}_{\boldsymbol{\theta}} \frac{1}{n_{\mathrm{tr}}} \sum_{i=1}^{n_{\mathrm{tr}}} \hat{r}(\boldsymbol{x}_i^{\mathrm{tr}}) \left\{ \left[ f(\boldsymbol{x}_i^{\mathrm{tr}}; \boldsymbol{\theta}) - y_i^{\mathrm{tr}} \right]^2 + \lambda \|\boldsymbol{\theta}\|^2 \right\},$$

where $f(\boldsymbol{x}; \boldsymbol{\theta}) = \sum_{i=1}^{n_{\mathrm{tr}}} \theta_i K(\boldsymbol{x}, \boldsymbol{x}_i^{\mathrm{tr}})$, $K(\boldsymbol{x}, \boldsymbol{x}') = \exp\{-\|\boldsymbol{x} - \boldsymbol{x}'\|^2/d\}$ is the kernel basis, and $\hat{r}$ is an estimator of $r^*$ based on DRE methods.

To evaluate the performance, we compute the normalized mean squared error (NMSE) on the test dataset:

$$\mathrm{NMSE} = \frac{1}{n_{\mathrm{te}}} \sum_{i=1}^{n_{\mathrm{te}}} \frac{(y_i^{\mathrm{te}} - \hat{y}_i^{\mathrm{te}})^2}{\sigma_y^2},$$

where $n_{\mathrm{te}}$ is the number of test samples, $y_i^{\mathrm{te}}$ is the true value for the $i$th sample, $\hat{y}_i^{\mathrm{te}}$ is the predicted value, and $\sigma_y^2$ is the variance of the response variable on the test dataset. Average NMSE values across 15 random replicates are reported in Table 3. More details on the datasets can be found in Appendix D.2.

As shown in Table 3, the unweighted regression model performs poorly across all datasets, underscoring the necessity for adjustments for the covariate shift problem. In contrast, our proposed ppDRE method consistently outperforms or matches the performance of the baseline methods across various datasets. This highlights the effectiveness of ppDRE in mitigating the impact of covariate shift, positioning it as a promising approach for regression tasks affected by this issue.

## 6. Conclusion

We propose a novel projection pursuit-based method for estimating the density ratio function, which does not require parametric assumptions, enjoys computational convenience, and can alleviate the curse of dimensionality. The asymptotic consistency and the convergence rates are established to guarantee the validity of the proposed method. Numerical experiments demonstrate that our method outperforms existing alternatives in a variety of applications.

Density ratio estimation based on projection pursuit admits many exciting directions for future work. One particularly promising direction is the extension of ppDRE to independence testing. However, this extension requires rigorous theoretical development, particularly in establishing the estimator's limiting distributions under the null hypothesis of independence. Furthermore, the method's iterative optimization process may still face challenges with computational complexity as the number of projections $K$ increases. Developing adaptive stopping criteria could alleviate the computational burden by avoiding tuning $K$ with cross validation.

## Acknowledgements

The research of Zheng Zhang is supported by the funds from the National Key R&D Program of China [grant number 2022YFA1008300], the Fundamental Research Funds for the Central Universities, and the Research Funds of Renmin University of China [project number 23XNA025]. Mingming Gong is supported by ARC DE210101624, ARC DP240102088, and WIS-MBZUAI 142571.

## Impact Statement

This paper presents work whose goal is to advance the field of Machine Learning in high-dimensional data. There are many potential societal consequences of our work, none

which we feel must be specifically highlighted here.

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

# A. Baseline Methods

This section presents the baseline methods used in our numerical studies.

**uLSIF**   The unconstrained Least-Squares Importance Fitting (uLSIF) method, proposed by Kanamori et al. (2009), models the density ratio $r^*(\boldsymbol{x})$ as follows:

$$r^*(\boldsymbol{x}) \approx \boldsymbol{\theta}^\top \psi(\boldsymbol{x}) = \sum_{\ell=1}^{n_p} \theta_\ell K(\boldsymbol{x}, \boldsymbol{x}_\ell^p), \tag{11}$$

where $K(\boldsymbol{x}, \boldsymbol{x}')$ is the Gaussian kernel basis, and the definitions of $\psi(\boldsymbol{x})$ and $\boldsymbol{\theta}$ are straightforward. The model parameter $\boldsymbol{\theta}$ is learned by minimizing the squared loss with a ridge penalty. For implementation, we employ the functions provided by the Python package *densratio*[1].

**KLIEP**   The Kullback-Leibler importance estimation procedure (KLIEP) method, proposed by Sugiyama et al. (2008), uses the same Gaussian kernel model for approximating $r^*(\boldsymbol{x})$ as that in the uLSIF method (11), but the model parameter $\boldsymbol{\theta}$ is identified by minimizing the unnormalized Kullback-Leibler (UKL) divergence:

$$\text{UKL}^*(r) = \mathbb{E}_q[r(\boldsymbol{x})] - \mathbb{E}_p[\log r(\boldsymbol{x})].$$

With the fact that $\mathbb{E}_q[r(\boldsymbol{x})] = 1$, the estimation is implemented as follows:

$$\max_{\boldsymbol{\theta}} \quad \frac{1}{n_p} \sum_{i=1}^{n_p} \log(\boldsymbol{\theta}^\top \psi(\boldsymbol{x}_i^p)) \qquad \text{s.t.} \quad \frac{1}{n_p} \sum_{i=1}^{n_q} \boldsymbol{\theta}^\top \psi(\boldsymbol{x}_i^q) = 1 \quad \text{and} \quad \boldsymbol{\theta} \geq \boldsymbol{0},$$

where the inequality for vectors is applied in the element-wise manner. The matlab code of the KLIEP method is available online[2].

**Probabilistic Classification**   By assigning the label $y = +1$ to the sample from $p(\boldsymbol{x})$ and $y = -1$ to the sample from $q(\boldsymbol{x})$ respectively (Qin, 1998; Bickel et al., 2007), the density ratio function $r^*(\boldsymbol{x}) = p(\boldsymbol{x})/q(\boldsymbol{x})$ can be expressed by

$$r^*(\boldsymbol{x}) = \frac{\mathbb{P}(\boldsymbol{x}|y=+1)}{\mathbb{P}(\boldsymbol{x}|y=-1)} = \frac{\mathbb{P}(y=-1)\mathbb{P}(y=+1|\boldsymbol{x})}{\mathbb{P}(y=+1)\mathbb{P}(y=-1|\boldsymbol{x})}.$$

Given an estimator of the posterior probability, $\hat{p}(y|\boldsymbol{x})$, the density ratio estimator $\hat{r}(\boldsymbol{x})$ can be constructed as

$$\hat{r}(\boldsymbol{x}) = \frac{n_q}{n_p} \cdot \frac{\hat{p}(y=+1|\boldsymbol{x})}{\hat{p}(y=-1|\boldsymbol{x})}.$$

In our experiments, we use the LightGBM model as the classifier, which is an ensemble learning algorithm based on gradient boosting decision trees.

**D³-LHSS**   The Direct Density-ratio estimation with Dimensionality reduction via Least-squares Heterodistributional Subspace Search (D³-LHSS) method is proposed by Sugiyama et al. (2011). This method assumes that the density ratio can be identified in a low-dimensional space specified by a projection matrix $\boldsymbol{U} \in \mathbb{R}^{m \times d}$. Given the projection matrix $\hat{\boldsymbol{U}}$ obtained by the LHSS algorithm, the estimator of the density ratio is given by

$$\hat{r}(\boldsymbol{x}) = \sum_{\ell=1}^{b} \hat{\theta}_\ell \psi_\ell(\hat{\boldsymbol{U}}\boldsymbol{x}),$$

where $\{\hat{\theta}_\ell\}_{\ell=1}^{b}$ are the learned using the uLSIF method on the heterodistributional subspace corresponding to $\hat{\boldsymbol{U}}$. For more details, we refer to Sugiyama et al. (2011), whose matlab implementation is available online[2].

---

[1]https://github.com/hoxo-m/densratio_py
[2]https://www.ms.k.u-tokyo.ac.jp/sugi/software.html

**fDRE**  The featurized density ratio estimation (fDRE) framework, proposed by Choi et al. (2021), addresses distributional discrepancy challenges through feature learning. By employing a normalizing flow model $f_\theta : \mathcal{X} \to \mathcal{Z}$ with invertible transformation properties, this method embeds heterogeneous data distributions into a unified feature space where density ratios remain preserved. The crucial invariance property is formally expressed as:

$$r^*(\boldsymbol{x}) = \frac{p(\boldsymbol{x})}{q(\boldsymbol{x})} = \frac{p'(f_\theta(\boldsymbol{x}))}{q'(f_\theta(\boldsymbol{x}))},$$

where $p', q'$ are the densities of $f_\theta(\boldsymbol{x}_p)$ and $f_\theta(\boldsymbol{x}_q)$ respectively, $\boldsymbol{x}_p \sim p(\boldsymbol{x})$, and $\boldsymbol{x}_q \sim q(\boldsymbol{x})$. This preservation enables direct application of classical density ratio estimators, such as KLIEP, in the transformed space $\mathcal{Z}$, often achieving superior numerical stability compared to native space estimation.

**RRND**  The regularized Radon-Nikodym differentiation estimation (RRND) method, proposed by Nguyen et al. (2024), introduces a kernel-based regularization framework within reproducing kernel Hilbert spaces (RKHS) for accurate pointwise estimation of Radon-Nikodym derivatives, which are equivalent to density ratio functions. This approach can extend the conventional kernel uLSIF through an iterative Lavrentiev regularization scheme, in which the core algorithm operates through successive approximations as follows:

$$\beta_{\boldsymbol{X}}^{\lambda,0} = 0, \qquad \beta_{\boldsymbol{X}}^{\lambda,l} = (\lambda \boldsymbol{I} + S_{X_p}^* S_{X_p})(S_{X_q}^* S_{X_q} \boldsymbol{1} + \lambda \beta_{\boldsymbol{X}}^{\lambda,l-1}), \; l \in \mathbb{N},$$

where $\beta_{\boldsymbol{X}}^{\lambda,l}$ is the $l$-th iteration of the approximation of $r^*(\boldsymbol{x})$. The methodology leverages two sample operators:

$$S_{X_p} f = \{f(\boldsymbol{x}_1^p), \ldots, f(\boldsymbol{x}_{n_p}^p)\}, \qquad S_{X_q} f = \{f(\boldsymbol{x}_1^q), \ldots, f(\boldsymbol{x}_{n_q}^q)\},$$

and two adjoint operators defined through kernel embeddings:

$$S_{X_p}^* u(\cdot) = \frac{1}{n_p} \sum_{j=1}^{n_p} K(\cdot, \boldsymbol{x}_j^p) u_j, \; u \in \mathbb{R}^{n_p}, \qquad S_{X_q}^* v(\cdot) = \frac{1}{n_q} \sum_{j=1}^{n_q} K(\cdot, \boldsymbol{x}_j^q) v_j, \; v \in \mathbb{R}^{n_q}.$$

For complete theoretical analysis and convergence properties, readers are directed to the original work by Nguyen et al. (2024).

**nnDRE**  We present a neural network-based density ratio estimation (nnDRE) method. It differs from our proposed ppDRE method in that it utilizes a feedforward neural network to model the density ratio, i.e.

$$r^*(\boldsymbol{x}) \approx r^{\mathrm{nn}}(\boldsymbol{x};\boldsymbol{\theta}) = \exp\{F(\boldsymbol{x}|\boldsymbol{\theta})\},$$

where $F(\cdot|\boldsymbol{\theta})$ represents the neural network with parameter $\boldsymbol{\theta}$. The estimation is proceeded by minimizing the squared loss based on the neural networks:

$$\mathrm{SQ}^*(\boldsymbol{\theta}) = \mathbb{E}_q\{[r^{\mathrm{nn}}(\boldsymbol{x};\boldsymbol{\theta})]^2\} - 2\mathbb{E}_p\{r^{\mathrm{nn}}(\boldsymbol{x};\boldsymbol{\theta})\}.$$

## B. Implementation Details and Hyperparameter Selection Process

To ensure rigorous and fair comparisons, we implemented the following protocols.

For our proposed ppDRE method, as mentioned in Remark 4.3, the optimal hyperparameters are identified by the minimal validation loss in CV. For clarity, we detail our approach as follows: across all experiments, we utilized 5-fold CV with random sampling. A grid search was conducted over a predefined set of parameter ranges, which are outlined in Table 4.

For open-source baseline methods (e.g., KLIEP, uLSIF, $D^3$-LHSS), we utilized their official implementations (e.g., Python densratio package for uLSIF) and default hyperparameter search grids as recommended in their original papers or standard toolkits. For instance, uLSIF's hyperparameters ($\sigma, \lambda$) were selected from the grid 1e-3:10:1e9, consistent with its standard implementation. These methods inherently incorporate CV-based hyperparameter selection in their standard workflows. We preserved these built-in CV mechanisms without modification.

For probabilistic classification approach and nnDRE, we implemented 5-fold CV (aligned with our ppDRE method) to identify the optimal hyperparameters within the search spaces outlined in Table 5.

*Table 4.* Search Grid for ppDRE

| Parameter | Description | Search Space |
|-----------|-------------|--------------|
| $K$ | Number of PP iterations | {5, 10, 15} |
| $J_k$ | Number of basis functions | {20, 50, 70, 100, 150} |
| $\lambda$ | $\ell_2$-regularization strength | {0.5, 1, 5, 10} |
| $\delta$ | Gradient descent learning rate | {0.001, 0.01, 0.1} |

*Table 5.* Search Grid for Probabilistic Classification approach and nnDRE

| Method | Parameter | Search Space |
|--------|-----------|--------------|
| Classification | n_estimators | {100, 300} |
| | learning_rate | {0.0001, 0.001, 0.01} |
| | num_leaves | {20, 30} |
| nnDRE | depth | {2, 3} |
| | width | {8, 32, 64} |
| | learning_rate | {0.0001, 0.001, 0.01} |

For fDRE, we have implemented a version that employs KLIEP as the second-stage DRE method. Adhering to the guidelines provided by the official open-source code[3] and the original research paper (Choi et al., 2021), we trained the masked autoregressive flow (MAF) models with the configurations presented in Table 6. For the KLIEP method in the second stage, we have retained the original hyperparameter settings as per the open-source implementation.

*Table 6.* Hyperparameter setting for the normalizing flow model in fDRE

| Dataset | n_blocks | n_hidden | hidden_size | n_epochs |
|---------|----------|----------|-------------|----------|
| IHDP | 5 | 1 | 100 | 100 |
| Regression Benchmarks | 5 | 1 | 100 | 100 |
| MI Gaussians | 5 | 1 | 100 | 200 |

For RRND, in the absence of open-source code, we implemented the algorithm by adhering to the implementation details outlined in the Numerical Illustrations section in Nguyen et al. (2024). The kernel function is assigned as $K(x, x') = 1 + \exp\{-(x-x')^2/2\}$. Utilizing a 5-fold CV, the hyperparameter $\lambda$ is chosen based on the quasi-optimality criterion, and the optimal iteration step $k$ is determined by minimizing squared distance loss function in the validation set. Following the configurations in Nguyen et al. (2024), the search grids are $k \in \{1, 2, \ldots, 10\}$ and $\lambda \in \{\lambda_\ell = \lambda_0 \rho^\ell, \ell = 1, \ldots, 9\}$ with $\lambda_0 = 0.9$. The decay factor $\rho = (\lambda_w / \lambda_0)^{1/l}$ is derived from the lower bound $\lambda_w = (n^{-1/2} + m^{-1/2})$, where $n, m$ are the sample sizes of the numerator and denominator distributions respectively.

## C. Additional Numerical Results

### C.1. Stabilized Weights Estimation

**Metrics** We use the mean squared error (RMSE) and the root mean squared logarithmic error (RMSLE) to evaluate the performance, which are defined as follows:

$$\text{RMSE} = \sqrt{\frac{1}{n} \sum_{i=1}^{n} (\hat{r}(\boldsymbol{x}_i) - r^*(\boldsymbol{x}_i))^2}, \qquad \text{RMSLE} = \sqrt{\frac{1}{n} \sum_{i=1}^{n} (\log \hat{r}(\boldsymbol{x}_i) - \log r^*(\boldsymbol{x}_i))^2}.$$

---

[3]https://github.com/ermongroup/f-dre

**Results** We present additional numerical results, including the box plot of RMSLE in Figure 4 and the RMSE values in Table 7. The box plot of RMSLE in Figure 4 shows the variation of all methods. In Table 7, due to the extremely large RMSE in some data replications, we report the median of RMSE values in 50 replications, which offers evidence supporting the superior performance of the ppDRE method.

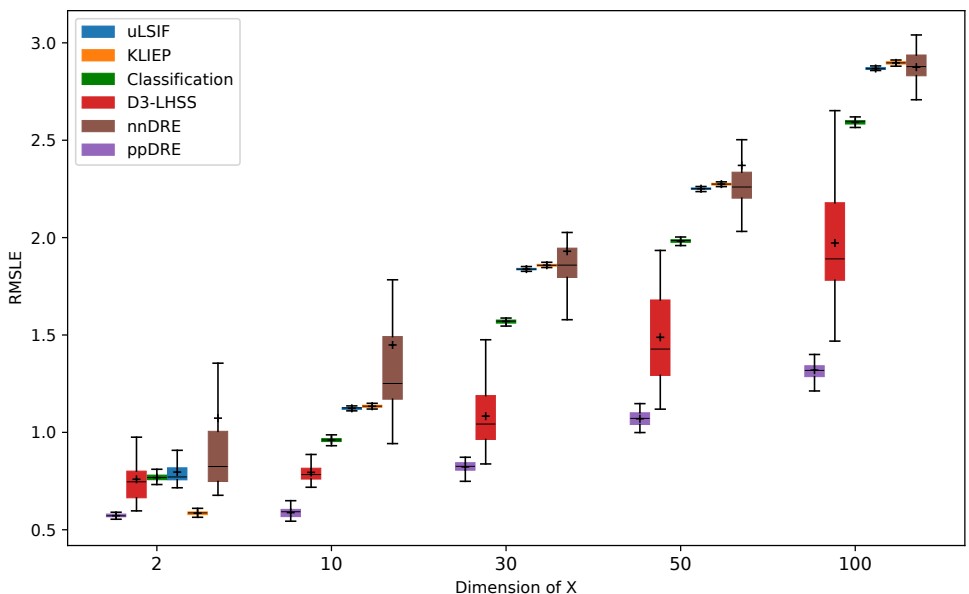

*Figure 4.* Boxplot of RMSLE in stabilized weights estimation for different covariate dimension settings $d_X \in \{2, 10, 30, 50, 100\}$ estimation across 50 data replications

*Table 7.* Median RMSE in stabilized weights estimation for different covariate dimension settings $d_X \in \{2, 10, 30, 50, 100\}$.

| (Median) RMSE | Dimension of Covariates | | | | |
|---|---|---|---|---|---|
| Method | 2 | 10 | 30 | 50 | 100 |
| uLSIF | 1.864 | 4.216 | 2.452 | 1.875 | 1.321 |
| KLIEP | 1.827 | 4.222 | 2.464 | 1.898 | 1.39 |
| Classification | 1.93 | 4.217 | 2.413 | 1.809 | 1.208 |
| $D^3$-LHSS | 2.604 | 8.793 | 2.637 | 2.454 | 2.054 |
| nnDRE | 2.181 | 4.739 | 2.772 | 2.227 | 1.549 |
| ppDRE | **1.742** | **3.835** | **1.777** | **1.12** | **0.9** |

## C.2. Computational Cost Evaluation

Through controlled numerical experimentation, we systematically evaluate the computational costs of all competing methodologies. Maintaining the experimental protocol established in Section 5.2.1, we conduct dimensional scalability analyses over $d_X \in \{2, 10, 30, 50, 100\}$ and measure absolute wall-clock execution times. The experiments were carried out on a computing node utilizing dual AMD EPYC 7713 processors, providing a total of 128 CPU cores. Table 8 presents the mean computational duration across three statistically independent trials, with temporal measurements recorded in minutes.

We obtain the following key observations: (i) when the dimension of the data is low or moderately large ($d_X = 2, 10$), the computation time of our PPDRE method is comparable to that of the uLSIF, KLIEP, and classification methods, which are suitable for low-dimensional data. Our computation time is much less than that of the fDRE, RRND, nnDRE and $D^3$-LHSS methods, some of which are also designed for high-dimensional data. (ii) when the dimension of the data is large ($d_X = 30, 50, 100$), the methods for low-dimensional data fail to work due to the curse of dimensionality. Our PPDRE

*Table 8.* Computation Time Comparison

| Method | $d = 2$ | $d = 10$ | $d = 30$ | $d = 50$ | $d = 100$ |
|---|---|---|---|---|---|
| uLSIF | 0.38 | 0.35 | 0.38 | 0.43 | 0.46 |
| KLIEP | 0.36 | 0.39 | 0.49 | 0.57 | 0.72 |
| Classification | 0.21 | 0.16 | 0.19 | 0.23 | 0.40 |
| fDRE | 3.78 | 3.88 | 3.92 | 4.28 | 7.25 |
| RRND | 3.78 | 3.87 | 3.77 | 3.82 | 3.81 |
| nnDRE | 5.91 | 3.84 | 8.76 | 6.02 | 8.63 |
| $D^3$-LHSS | 5.95 | 11.25 | 10.60 | 10.79 | 10.97 |
| ppDRE | 0.32 | 0.91 | 5.64 | 10.78 | 15.22 |

method consistently outperforms all baseline methods in estimation accuracy, and the computation time is comparable to that of existing methods for high-dimensional data. The observed increase in computation time is expected, as higher dimensions naturally require more iterations to reach convergence.

## D. Datasets

### D.1. Dataset in Dose Response Function Estimation

**IHDP-continuous Dataset**  The original IHDP dataset contains 25 covariates with binary treatments and continuous outcomes. We follow similar procedure as employed by Nie et al. (2020) and Gao et al. (2023), disregard the treatments and outcomes, and use the covariates to generate continuous dosages and treatments. Let $X_1, \ldots, X_{d_X}$ denote the first to the last element of $X$, where $d_X = 25$. The generating procedure for the semi-synthetic IHDP-continuous dataset is as follows:

$$\textbf{Assigned treatment: } T = (1 + \exp(\tilde{T}))^{-1}, \tag{12}$$

$$\text{where } \tilde{T} = \frac{X_1}{1 + X_2} + \frac{\max\{X_3, X_4, X_5\}}{0.2 + \min\{X_3, X_4, X_5\}} + \tanh\left(5\frac{\sum_{i \in I_1} X_i}{|I_1|}\right) - 2 + 0.5\epsilon,$$

$$\textbf{Potential outcome: } Y^*(t) = h(t, X) + 0.5\epsilon,$$

$$\text{where } h(t, X) = (1.2 - t^2)\sin(2\pi t - 2)\left\{0.5\tanh\left(5\frac{\sum_{i \in I_2} X_i}{|I_2|}\right) + 1.5\exp\left(\frac{0.2(X_1 - X_5)}{0.1 + \min\{X_2, X_3, X_4\}}\right)\right\}, \tag{13}$$

where $\epsilon \sim \mathcal{N}(0, 1)$, $I_1 = \{3, 6, 7, 8, 9, 10, 11, 12, 13, 14\}$ and $I_2 = \{15, 16, 17, 18, 19, 20, 21, 22, 23, 24\}$.

### D.2. Datasets in Covariate Shift Adaptation

In this section, we introduce the basic information and the regression tasks of the benchmark datasets that are used in the covariate shift adaption experiments. These datasets are readily accessible via the Ready Tensor platform. The specific numbers of observations and features for each dataset are detailed in Table 9.

*Table 9.* Observation and Feature Numbers of Regression Benchmark Datasets

| Dataset | Abalone | Billboard Spotify | Cancer Mortality | Computer Activity | Diamond Prices |
|---|---|---|---|---|---|
| Observation | 4177 | 8930 | 3047 | 8192 | 6000 |
| Feature | 8 | 18 | 31 | 21 | 7 |

**Abalone**  The Abalone dataset, accessible online[4], is a popular dataset used in machine learning and statistics to predict the age of abalone from physical measurements. Abalone age is determined by cutting the shell, staining it, and counting the number of growth rings. However, this is a destructive process, so a non-destructive method based on physical measurements is desirable.

---

[4]https://www.dcc.fc.up.pt/~ltorgo/Regression/DataSets.html

**Billboard Spotify** The Billboard Spotify dataset represents a comprehensive collection of audio features for songs that made their mark on various Billboard charts that span the years 1961 through 2022. For each song, the corresponding audio features were sourced from the Spotify API. The regression task for the Billboard Spotify dataset is to predict the danceability of a song, which describes how suitable a track is for dancing based on a combination of musical elements including tempo, rhythm stability, beat strength, and overall regularity.

**Cancer Mortality** The Cancer Mortality dataset, aggregated from authoritative sources including the American Community Survey, clinicaltrials.gov, and cancer.gov, is designed for regression tasks to predict cancer mortality rates in US counties from 2010 to 2016. It incorporates 2013 census data, including county-level features such as population, income, households, and other demographic attributes.

**Computer Activity** The Computer Activity databases consist of a collection of metrics related to the activity of computer systems. The data was collected from a Sun Sparcstation 20/712 with 128 Mbytes of memory running in a multi-user university department. This dataset, available online[4], is utilized to estimate the proportion of time during which central processing units (CPUs) are engaged in user mode operations, based on the recorded system activity measures.

**Diamond Prices** The Diamond Prices dataset is accessible via the PyCaret library within the Python programming environment. The objective of this dataset is to predict diamond prices based on a set of attributes, including carat weight, cut, color, clarity, polish, symmetry, and the report issued by the grading agency that assessed the diamond's qualities.

## E. $L^2$-distance Minimization

In this section, we show that minimizing the $L^2$-distance $\mathbb{E}_q\{[r^*(\boldsymbol{x}) - r_{k-1}(\boldsymbol{x})f(\boldsymbol{a}^\top\boldsymbol{x})]^2\}$ w.r.t. $f$ or $\boldsymbol{a}$ is equivalent to minimizing $H(f, \boldsymbol{a}) := \mathbb{E}_q\{r_{k-1}^2(\boldsymbol{x})f^2(\boldsymbol{a}^\top\boldsymbol{x})\} - 2\mathbb{E}_p\{r_{k-1}(\boldsymbol{x})f(\boldsymbol{a}^\top\boldsymbol{x})\}$.

Note that, by definition, $r^*(\boldsymbol{x}) = p(\boldsymbol{x})/q(\boldsymbol{x})$. Then,

$$\mathbb{E}_q\{[r^*(\boldsymbol{x}) - r_{k-1}(\boldsymbol{x})f(\boldsymbol{a}^\top\boldsymbol{x})]^2\}$$

$$= \mathbb{E}_q\{[r^*(\boldsymbol{x})]^2\} + \mathbb{E}_q\{[r_{k-1}(\boldsymbol{x})f(\boldsymbol{a}^\top\boldsymbol{x})]^2\} - 2\int_{\mathcal{X}} r_{k-1}(\boldsymbol{x})f(\boldsymbol{a}^\top\boldsymbol{x})r^*(\boldsymbol{x})q(\boldsymbol{x})\,d\boldsymbol{x}$$

$$= \mathbb{E}_q\{[r^*(\boldsymbol{x})]^2\} + \mathbb{E}_q\{[r_{k-1}(\boldsymbol{x})f(\boldsymbol{a}^\top\boldsymbol{x})]^2\} - 2\int_{\mathcal{X}} r_{k-1}(\boldsymbol{x})f(\boldsymbol{a}^\top\boldsymbol{x})p(\boldsymbol{x})\,d\boldsymbol{x}$$

$$= \mathbb{E}_q\{[r^*(\boldsymbol{x})]^2\} + \mathbb{E}_q\{[r_{k-1}(\boldsymbol{x})f(\boldsymbol{a}^\top\boldsymbol{x})]^2\} - 2\mathbb{E}_p\{r_{k-1}(\boldsymbol{x})f(\boldsymbol{a}^\top\boldsymbol{x})\}\,.$$

Since the first term $\mathbb{E}_q\{[r^*(\boldsymbol{x})]^2\}$ is independent of either $f$ or $\boldsymbol{a}$, we have that minimizing $\mathbb{E}_q\{[r^*(\boldsymbol{x}) - r_{k-1}(\boldsymbol{x})f(\boldsymbol{a}^\top\boldsymbol{x})]^2\}$ w.r.t. $f$ or $\boldsymbol{a}$ is equivalent to minimizing $H(f, \boldsymbol{a})$.

## F. Proof of Theorem 4.2

### F.1. Notations and Assumptions

To prove the theorem, we require the following notations and assumptions.

**Notations of derivative.** For any univariate function $f$, we let $f^{(j)}$ denote its $j$th derivative. For any multivariate function $g(\boldsymbol{\beta}, \boldsymbol{a})$, we denote $\partial_1 g(\boldsymbol{\beta}, \boldsymbol{a})$ to be its partial derivative with respect to w.r.t. the first argument $\boldsymbol{\beta}$, and $\partial_2 g(\boldsymbol{\beta}, \boldsymbol{a})$ to be its partial derivative (w.r.t.) the second argument $\boldsymbol{a}$.

**Notations regarding matrices.** Consider $j = 0, 1, 2$. We let $\Omega_{J_k}^{(j)}(\boldsymbol{a}) := \mathbb{E}[\boldsymbol{\Phi}_k^{(j)}(\boldsymbol{a}^\top\boldsymbol{x})\boldsymbol{\Phi}_k^{(j)}(\boldsymbol{a}^\top\boldsymbol{x})^\top]$, for $j = 0, 1, 2$, where the expectation can be taken w.r.t. the probability density $p(\cdot)$ or $q(\cdot)$.

For the corresponding empirical version, let $\{\boldsymbol{x}_1, \ldots, \boldsymbol{x}_n\}$ be i.i.d. from either the probability density $p(\cdot)$ (resp. $n = n_p$) or $q(\cdot)$ (resp. $n = n_q$), we define $\boldsymbol{P}^{(j)}(\boldsymbol{a}) = \{\boldsymbol{\Phi}_k^{(j)}(\boldsymbol{a}^\top\boldsymbol{x}_1), \ldots, \boldsymbol{\Phi}_k^{(j)}(\boldsymbol{a}^\top\boldsymbol{x}_n)\} \in \mathbb{R}^{n \times J_k}$, and $\hat{\Omega}_{J_k}^{(j)}(\boldsymbol{a}) = n^{-1}\boldsymbol{P}^{(j)}(\boldsymbol{a})^\top\boldsymbol{P}^{(j)}(\boldsymbol{a})$.

Similarly, we define $\Sigma_{J_k}(\boldsymbol{a}) = \mathbb{E}_q\left[r_{k-1}^2(\boldsymbol{x})\boldsymbol{\Phi}_k(\boldsymbol{a}^\top\boldsymbol{x})\boldsymbol{\Phi}_k(\boldsymbol{a}^\top\boldsymbol{x})^\top\right]$, $\hat{\Sigma}_{J_k}(\boldsymbol{a}) = n_q^{-1}\boldsymbol{Z}_k(\boldsymbol{a})^\top\boldsymbol{Z}_k(\boldsymbol{a})$, $\hat{\Sigma}_{J_k,\lambda}(\boldsymbol{a}) = \hat{\Sigma}_{J_k}(\boldsymbol{a}) + \lambda\boldsymbol{I}_{J_k}$, where $\boldsymbol{Z}_k(\boldsymbol{a})$ is defined in Proposition 4.1.

Moreover, we let $\eta_{\min}(\Omega)$ and $\eta_{\max}(\Omega)$ denote the minimum and maximum eigenvalues of any matrix $\Omega$, respectively.

**Notations regarding minimization criteria.** Recall the definitions of $H(f, \boldsymbol{a})$ and $\hat{\mathcal{L}}_k(\boldsymbol{a}, \boldsymbol{\beta}; \lambda)$ from (2) and (4), respectively. By definition, for any $\boldsymbol{a} \in \mathcal{A}$,

$$\hat{f}_{\boldsymbol{a},k}(\boldsymbol{a}^\top \boldsymbol{x}) = \hat{\boldsymbol{\beta}}_k(\boldsymbol{a})^\top \boldsymbol{\Phi}_k(\boldsymbol{a}^\top \boldsymbol{x}), \quad \text{and} \quad \hat{\boldsymbol{\beta}}_k(\boldsymbol{a}) = \{\hat{\Sigma}_{J_k,\lambda}(\boldsymbol{a})\}^{-1} \frac{1}{n_p} \sum_{i=1}^{n_p} \hat{r}_{k-1}(\boldsymbol{x}_i^p) \boldsymbol{\Phi}_k(\boldsymbol{a}^\top \boldsymbol{x}_i^p). \tag{14}$$

We define $H^*(\boldsymbol{\beta}, \boldsymbol{a}) := \mathbb{E}_q[r_{k-1}^2(\boldsymbol{x}) \cdot \{\boldsymbol{\beta}^\top \boldsymbol{\Phi}_k(\boldsymbol{a}^\top \boldsymbol{x})\}^2] - 2\mathbb{E}_p[r_{k-1}(\boldsymbol{x}) \cdot \boldsymbol{\beta}^\top \boldsymbol{\Phi}_k(\boldsymbol{a}^\top \boldsymbol{x})]$ to be the approximation of $H(f, \boldsymbol{a})$ and the theoretical counterpart of $\hat{\mathcal{L}}_k$. We then define $\boldsymbol{\beta}_k^*(\boldsymbol{a}) := \underset{\boldsymbol{\beta}}{\operatorname{argmin}} H^*(\boldsymbol{\beta}, \boldsymbol{a})$ for any $\boldsymbol{a} \in \mathcal{A}$, $\boldsymbol{a}_k^* = \underset{\boldsymbol{a} \in \mathcal{A}}{\operatorname{argmin}} H^*(\boldsymbol{\beta}_k^*(\boldsymbol{a}), \boldsymbol{a})$, and $f_{\boldsymbol{a},k}^*(\boldsymbol{a}^\top \boldsymbol{x}) = \boldsymbol{\beta}_k^*(\boldsymbol{a})^\top \boldsymbol{\Phi}_k(\boldsymbol{a}^\top \boldsymbol{x})$ for any $\boldsymbol{x} \in \mathcal{X}$.

**Assumption F.1.** (a) The support $\mathcal{X}$ of $\boldsymbol{x}$ is a compact subset of $\mathbb{R}^d$. (b) The parameters $\{\boldsymbol{a}_k\}_{k=1}^K$ defined in (7) are in the interior of a compact set $\mathcal{A} \subset \mathbb{S}_d^+$.

**Assumption F.2.** (a) For every $\boldsymbol{a} \in \mathcal{A}$, there exists a $\boldsymbol{\beta}_{k,j}(\boldsymbol{a}) \in \mathbb{R}^{J_k}$ such that

$$\sup_{\boldsymbol{a} \in \mathcal{A}} \sup_{z \in \mathcal{Z}} \left| f_{\boldsymbol{a},k}^{(j)}(z) - \{\boldsymbol{\beta}_{k,j}(\boldsymbol{a})\}^\top \boldsymbol{\Phi}_k^{(j)}(z) \right| < C_{0,j} J_k^{-(s-j)}, \tag{15}$$

for some constants $C_{0,j}, s > 0$, and $0 \le j < s$, where $\mathcal{Z} := \{\boldsymbol{a}^\top \boldsymbol{x} : \boldsymbol{a} \in \mathcal{A} \text{ and } \boldsymbol{x} \in \mathcal{X}\}$; (b) For every $z \in \mathcal{Z}$, $f_{\boldsymbol{a},k}(z)$ is continuously differentiable w.r.t. $\boldsymbol{a}$, and $\sup_{z \in \mathcal{Z}} \{\|\partial_{\boldsymbol{a}} f_{\boldsymbol{a},k}(z)\|_{\boldsymbol{a}=\boldsymbol{a}_k}\} < \infty$; (c) $f_{\boldsymbol{a},k}(z)$ are uniformly bounded and bounded away from 0.

**Assumption F.3.** (a) The eigenvalues of $\Omega_{J_k}^{(0)}(\boldsymbol{a})$ are bounded and bounded away from zero uniformly in $J_k$ and $\boldsymbol{a} \in \mathcal{A}$. (b) There exist sequences of constants $\tilde{\zeta}_j(J_k)$, $j = 1, 2$, such that the eigenvalues of $\Omega_{J_k}^{(j)}(\boldsymbol{a})$ are bounded by $\tilde{\zeta}_j(J_k)$ uniformly in $J_k$ and $\boldsymbol{a} \in \mathcal{A}$ (c) There exist sequences of constants $\zeta_j(J_k)$, $j = 0, 1, 2$, such that $\sup_{j \le m} \sup_{z \in \mathcal{Z}} \|\boldsymbol{\Phi}_k^{(j)}(z)\| \le \zeta_m(J_k)$ for $m = 0, 1, 2$ and all $k$. (d) As $n_p, n_q \to \infty$, $J_k \to \infty$, $\zeta_0(J_k)^2 \sqrt{J_k/(n_p \wedge n_q)} \to 0$, $J_k^{-(s-1)} \max\{\zeta_2(J_k), \zeta_0(J_k)^2\} \to 0$, $\zeta_2(J_k)\sqrt{J_k/(n_p \wedge n_q)} \to 0$ and $\lambda = O(\sqrt{J_k/n_q})$.

**Assumption F.4.** The minimum eigenvalue of $\partial_{\boldsymbol{a}}^2 H^*(\boldsymbol{\beta}_k^*(\boldsymbol{a}), \boldsymbol{a})|_{\boldsymbol{a}=\boldsymbol{a}_k^*}$ is bounded away from 0 uniformly in $J_k$.

Assumption F.1 imposes compactness conditions on the densities and parameter spaces. This condition is convenient for deriving uniform convergence rates. Assumption F.2 includes regularity conditions. Assumption F.2 (a) can be satisfied if $f_{\boldsymbol{a},k}(z)$ is $s$-times continuously differentiable w.r.t. $z \in \mathcal{Z}$ for any $\boldsymbol{a} \in \mathcal{A}$ (Lorentz, 1986). Assumption F.3 rules out near multicollinearity in the approximating basis functions. This condition is familiar in the sieve regression literature (Chen, 2007a). Assumption F.4 requires the Hessian matrix of the approximation of the theoretical minimization criteria to be positive definite at its minimum, $\boldsymbol{a}_k^*$, which is satisfied if $\boldsymbol{a}_k^*$ is in the interior of $\mathcal{A}$.

## F.2. Outline of the proof

We prove the results in an inductive way. Note that the initial estimate $f_0(\boldsymbol{x}; \boldsymbol{\beta}_0) \equiv 1$, $r_k(\boldsymbol{x}) = 1 \cdot \Pi_{m=1}^k f_{\boldsymbol{a}_m,m}(\boldsymbol{a}_m^\top \boldsymbol{x})$ and $\hat{r}_k(\boldsymbol{x}) = f_0(\boldsymbol{x}, \boldsymbol{\beta}_0) \Pi_{m=1}^k \hat{f}_{\hat{\boldsymbol{a}}_m,m}(\hat{\boldsymbol{a}}_m^\top \boldsymbol{x})$, and $\hat{f}_{\boldsymbol{a},k}$ is estimated based on $\hat{r}_{k-1}$. Let $r_0(\boldsymbol{x}) = 1$ and $\hat{r}_0(\boldsymbol{x}) = f_0(\boldsymbol{x}; \boldsymbol{\beta}_0) = 1$. The following results hold for $k = 1$.

$$\frac{1}{n} \sum_{i=1}^n |\hat{r}_{k-1}(\boldsymbol{x}_i) - r_{k-1}(\boldsymbol{x}_i)|^2 = O_p(\xi_{n,k-1}). \tag{16}$$

We shall show that, for any $k \in \{1, 2, \ldots, K\}$, given (16) holds, we have

$$\sup_{\boldsymbol{x} \in \mathcal{X}} |\hat{f}_{\hat{\boldsymbol{a}}_k,k}(\hat{\boldsymbol{a}}_k^\top \boldsymbol{x}) - f_{\boldsymbol{a}_k,k}(\boldsymbol{a}_k^\top \boldsymbol{x})| = O_P\left(\{J_k^{-(s-1)} + \sqrt{\xi_{n,k-1}} + \sqrt{J_k/(n_p \wedge n_q)}\} \cdot \{\sqrt{\tilde{\zeta}_1(J_k)} \vee \zeta_0^2(J_k)\}\right), \tag{17}$$

where $\tilde{\zeta}_1(J_k)$ is the rate of the maximum eigenvalue of $\Omega_{J_k}^{(1)}(\boldsymbol{a})$ and $\sqrt{\tilde{\zeta}_1(J_k)} \vee \zeta_0(J_k) = \max\{\sqrt{\tilde{\zeta}_1(J_k)}, \zeta_0(J_k)\}$. Then, using the fact that $\xi_{n,0} = 0$, we can inductively derive that

$$\sup_{\boldsymbol{x} \in \mathcal{X}} |\hat{r}_K(\boldsymbol{x}) - r_K(\boldsymbol{x})| = O_P\left(\sum_{\ell=1}^K \left[\left\{J_\ell^{-(s-1)} + \sqrt{\frac{J_\ell}{n_q \wedge n_p}}\right\} \cdot \prod_{i=\ell}^K \left\{\sqrt{\tilde{\zeta}_1(J_i)} \vee \zeta_0^2(J_i)\right\}\right]\right),$$

which establishes the last statement of Theorem 4.2.

**Proof of** (17). To prove (17), we will need to establish the results in (8) and (9), which are relegated to Sections F.3 and F.4, respectively. Specifically, we decompose

$$\hat{f}_{\hat{a}_k,k}(\hat{a}_k^\top x) - f_{a_k,k}(a_k^\top x) = \hat{f}_{\hat{a}_k,k}(\hat{a}_k^\top x) - f_{\hat{a}_k,k}(\hat{a}_k^\top x) \tag{18}$$

$$+ f_{\hat{a}_k,k}(a_k^\top x) - f_{a_k,k}(a_k^\top x) \tag{19}$$

$$+ f_{\hat{a}_k,k}(\hat{a}_k^\top x) - f_{\hat{a}_k,k}(a_k^\top x). \tag{20}$$

For (18), we have $\sup_{x\in\mathcal{X}}|\hat{f}_{\hat{a}_k,k}(\hat{a}_k^\top x) - f_{\hat{a}_k,k}(\hat{a}_k^\top x)| \le \sup_{a\in\mathcal{A}}\sup_{x\in\mathcal{X}}|\hat{f}_{a,k}(a^\top x) - f_{a,k}(a^\top x)|.$

For (19), under Assumption F.2 (b), we can use Taylor's expansion to expand $f_{\hat{a}_k,k}$ around $f_{a_k,k}$ and obtain $\sup_{x\in\mathcal{X}}|f_{\hat{a}_k,k}(a_k^\top x) - f_{a_k,k}(a_k^\top x)| = O_p(\|\hat{a}_k - a_k\|).$

For (20), we can first decompose it as

$$(20) = \left\{ f_{a_k,k}(\hat{a}_k^\top x) - f_{a_k,k}(a_k^\top x) \right\} + \left\{ f_{\hat{a}_k,k}(\hat{a}_k^\top x) - f_{a_k,k}(\hat{a}_k^\top x) \right\} + \left\{ f_{a_k,k}(a_k^\top x) - f_{\hat{a}_k,k}(a_k^\top x) \right\}.$$

The supremum norm of the first term can be bounded by $O_p(\|\hat{a}_k - a_k\|)$, using Taylor's expansion expanding $f_{a_k,k}(\hat{a}_k^\top x)$ around $f_{a_k,k}(a_k^\top x)$. The second and the third term can also be bounded by $O_p(\|\hat{a}_k - a_k\|)$ using the same argument for (19). Thus, we have

$$\sup_{x\in\mathcal{X}}|\hat{f}_{\hat{a}_k,k}(\hat{a}_k^\top x) - f_{a_k,k}(a_k^\top x)| = O_p\left( \sup_{a\in\mathcal{A}}\sup_{x\in\mathcal{X}}|\hat{f}_{a_k,k}(a_k^\top x) - f_{a_k,k}(a_k^\top x)| + \|\hat{a}_k - a_k\| \right).$$

Then, using (8) and (9), and noting that $\zeta_0(J_k) \le J_k$, we then have (17) holds. The outline of the proof is completed.

### F.3. Proof of (8)

Recalling the definitions of $f_{a,k}$ in (6) and $f_{a,k}^*$ in Section F.1, we decompose

$$\sup_{a\in\mathcal{A}}\sup_{z\in\mathcal{Z}}\left|\hat{f}_{a,k}(z) - f_{a,k}(z)\right| \le \sup_{a\in\mathcal{A}}\sup_{z\in\mathcal{Z}}\left|f_{a,k}^*(z) - f_{a,k}(z)\right| \tag{21}$$

$$+ \sup_{a\in\mathcal{A}}\sup_{z\in\mathcal{Z}}\left|\hat{f}_{a,k}(z) - f_{a,k}^*(z)\right|. \tag{22}$$

We derive the convergence rates for (21) and (22).

**Rate of** (21).

We obtain the rate

$$(21) = O(J_k^{-s}\zeta_0(J_k))$$

from Lemma F.9.

**Rate for** (22). By definition, we have

$$\beta_k^*(a) = \{\Sigma_{J_k}(a)\}^{-1}\mathbb{E}_p[r_{k-1}(x)\Phi_k(a^\top x)].$$

Recalling the representation of $\hat{f}_{a,k}$ in (14), we decompose

$$\sup_{a\in\mathcal{A}}\sup_{x\in\mathcal{X}}|\hat{f}_{a,k}(a^\top x) - f_{a,k}^*(a^\top x)|$$

$$= \sup_{a\in\mathcal{A}}\sup_{x\in\mathcal{X}}\left|\Phi_k(a^\top x)^\top\{\hat{\Sigma}_{J_k,\lambda}(a)\}^{-1}\frac{1}{n_p}\sum_{i=1}^{n_p}\hat{r}_{k-1}(x_i^p)\Phi_k(a^\top x_i^p) - \Phi_k(a^\top x)^\top\{\Sigma_{J_k}(a)\}^{-1}\mathbb{E}_p[r_{k-1}(x)\Phi_k(a^\top x)]\right|$$

$$\le \sup_{a\in\mathcal{A}}\sup_{x\in\mathcal{X}}\left|\Phi_k(a^\top x)^\top\{\hat{\Sigma}_{J_k,\lambda}(a)\}^{-1}\frac{1}{n_p}\sum_{i=1}^{n_p}\{\hat{r}_{k-1}(x_i^p) - r_{k-1}(x_i^p)\}\Phi_k(a^\top x_i^p)\right| \tag{23}$$

$$+ \sup_{a\in\mathcal{A}}\sup_{x\in\mathcal{X}}\left|\Phi_k(a^\top x)^\top\{\hat{\Sigma}_{J_k,\lambda}(a)\}^{-1}\frac{1}{n_p}\sum_{i=1}^{n_p}r_{k-1}(x_i^p)\Phi_k(a^\top x_i^p) - \Phi_k(a^\top x)^\top\{\hat{\Sigma}_{J_k,\lambda}(a)\}^{-1}\mathbb{E}_p[r_{k-1}(x)\Phi_k(a^\top x)]\right| \tag{24}$$

$$+ \sup_{a\in\mathcal{A}}\sup_{x\in\mathcal{X}}\left|\Phi_k(a^\top x)^\top\left\{\{\hat{\Sigma}_{J_k,\lambda}(a)\}^{-1} - \{\Sigma_{J_k}(a)\}^{-1}\right\}\mathbb{E}_p[r_{k-1}(x)\Phi_k(a^\top x)]\right|. \tag{25}$$

We derive the rate for (23) to (25) one by one.

**Rate for** (23). Note that the term in the absolute value sign in (23) is the $L_2$-projection of $\{\hat{r}_{k-1}(x^P) - r_{k-1}(x^P)\}$ to the linear space spanned by the sieve basis $\Phi_k(a^\top x)$. Given that $\mathcal{A}$ and $\mathcal{X}$ are compact sets and the sieve basis $\Phi_k$ is a local basis, we have $(23) \leq \sup_{a \in \mathcal{A}} \sup_{x \in \mathcal{X}} \|\|\Phi_k(a^\top x)\|\| \sqrt{\sum_{i=1}^{n_p} \{\hat{r}_{k-1}(x_i^p) - r_{k-1}(x_i^P)\}^2/n_P} = O_P(\zeta_0(J_k)\sqrt{\xi_{n,k-1}})$ .

**Rate for** (24). Define $b_a(z) = \Phi_k(z)^\top \{\hat{\Sigma}_{J_k,\lambda}(a)\}^{-1}/\|\Phi_k(z)^\top \{\hat{\Sigma}_{J_k,\lambda}(a)\}^{-1}\|$ and $\mathbb{S}_{J_k}$ to be the $J_k$ dimensional unit ball $\mathbb{S}_{J_k} := \{b \in \mathbb{R}^{J_k} : \|b\| = 1\}$. Since, under Assumption F.3,

$$\sup_{a \in \mathcal{A}} \sup_{x \in \mathcal{X}} \|\Phi_k(z)^\top \{\hat{\Sigma}_{J_k,\lambda}(a)\}^{-1}\| = O_P(\zeta_0(J_k)),$$

the term (24) can be bounded as follows:

$$(24) = O_P\{\zeta_0(J_k)\} \cdot \sup_{a \in \mathcal{A}} \sup_{x \in \mathcal{X}} \left| \frac{1}{n_P} \sum_{i=1}^{n_p} b_a(a^\top x)^\top r_{k-1}(x_i^P)\Phi_k(a^\top x_i^P) - \mathbb{E}_p[b_a(a^\top x)^\top r_{k-1}(x)\Phi_k(a^\top x)] \right|$$

$$\leq O_P\{\zeta_0(J_k)\} \cdot \sup_{(b,a) \in \mathbb{S}_{J_k} \times \mathcal{A}} \left| \frac{1}{n_P} \sum_{i=1}^{n_p} b^\top r_{k-1}(x_i^P)\Phi_k(a^\top x_i^P) - \mathbb{E}_p[b^\top r_{k-1}(x)\Phi_k(a^\top x)] \right|. \tag{26}$$

Consider the measurable function class $\mathcal{H}_k := \{x \mapsto b^\top \Phi_k(a^\top x)\} : (b,a) \in \mathbb{S}_{J_k} \times \mathcal{A}\}$ with its envelope denoted by $H_k(x) := \sup_{(b,a) \in \mathbb{S}_{J_k} \times \mathcal{A}} \|b^\top \Phi_k(a^\top x)\| = \sup_{a \in \mathcal{A}} \|\Phi_k(a^\top x)\|$. We denote $\|H\|_f := \{\int H(x)^2 f(x) dx\}^{1/2}$ for any function $H$ and density $f$. Then $\|H_k\|_p = O(\zeta_0(J_k))$. Note that by the maximal inequality (e.g., Vaart & Wellner, 1996, Theorem 2.14.2 ),

$$\mathbb{E}\left[ \sup_{(b,a) \in \mathbb{S}_{J_k} \times \mathcal{A}} \left| \frac{1}{n_P} \sum_{i=1}^{n_p} b^\top r_{k-1}(x_i^P)\Phi_k(a^\top x_i^P) - \mathbb{E}_p[b^\top r_{k-1}(x)\Phi_k(a^\top x)] \right| \right]$$

$$\leq \frac{C}{\sqrt{n_P}} \cdot \int_0^1 \sqrt{1 + \log N_{[]}(\epsilon \cdot \|H_k\|_p, \mathcal{H}_k, \|\cdot\|_p)} d\epsilon \cdot \|H_k\|_p,$$

where $N_{[]}(\epsilon\|H_k\|_p, \mathcal{H}_k, \|\cdot\|_p)$ is the minimum number of $\epsilon\|H_k\|_p$-brackets needed to cover $\mathcal{H}_k$ (Vaart & Wellner, 1996, Section 2.1). Since $\|H_k\|_p \leq \|H_k\|_\infty$, we have

$$N_{[]}(\epsilon, \mathcal{H}_k, \|\cdot\|_p) \leq N_{[]}(\epsilon, \mathcal{H}_k, \|\cdot\|_\infty). \tag{27}$$

Then, by Lemma F.5 we deduce that

$$\int_0^1 \sqrt{1 + \log N_{[]}(\epsilon \cdot \|H_k\|_p, \mathcal{H}_k, \|\cdot\|_p)} d\epsilon$$

$$\leq C \int_0^1 \sqrt{1 - J_k \log\left(\frac{\epsilon \cdot \|H_k\|_p}{\zeta_0(J_k)}\right)} d\epsilon$$

$$\leq C\sqrt{J_k} \int_0^1 \sqrt{\frac{1}{J_k} - \log(\epsilon)} d\epsilon = O(\sqrt{J_k}).$$

Hence,

$$\mathbb{E}\left[ \sup_{(b,a) \in \mathbb{S}_{J_k} \times \mathcal{A}} \left| \frac{1}{n_P} \sum_{i=1}^{n_p} b^\top r_{k-1}(x_i^P)\Phi_k(a^\top x_i^P) - \mathbb{E}_p[b^\top r_{k-1}(x)\Phi_k(a^\top x)] \right| \right] = O_P\left(\frac{\sqrt{J_k}\zeta_0(J_k)}{n_P}\right).$$

Consequently, by (26), we have

$$(24) = O_P\left(\frac{\sqrt{J_k}\zeta_0(J_k)^2}{\sqrt{n_P}}\right). \tag{28}$$

**Rate for** (25).

Note that, under Assumption F.3 and recalling the definition of $\eta_{\max}(\cdot)$ and the results in Lemma F.6,

$$
\begin{aligned}
(25) &\leq \zeta_0(J_k) \sup_{\boldsymbol{a} \in \mathcal{A}} \eta_{\max}\left(\{\hat{\Sigma}_{J,\lambda}(\boldsymbol{a})\}^{-1}\{\Sigma_{J_k}(\boldsymbol{a})\}^{-1}\right) \left\|\{\Sigma_J(\boldsymbol{a}) - \hat{\Sigma}_{J,\lambda}(\boldsymbol{a})\}\mathbb{E}_p\left[r_{k-1}(\boldsymbol{x})\boldsymbol{\Phi}_k(\boldsymbol{a}^\top \boldsymbol{x})\right]\right\| \\
&= O_p(\zeta_0(J_k)) \sup_{\boldsymbol{a} \in \mathcal{A}} \left[\sqrt{\eta_{\max}\left(\{\Sigma_J(\boldsymbol{a}) - \hat{\Sigma}_{J_k}(\boldsymbol{a})\}^2\right)} + \lambda\right] \left\|\mathbb{E}_p\left[r_{k-1}(\boldsymbol{x})\boldsymbol{\Phi}_k(\boldsymbol{a}^\top \boldsymbol{x})\right]\right\| \\
&\leq O_p(\zeta_0(J_k)) \sup_{\boldsymbol{a} \in \mathcal{A}} \left\{\lambda + \left\|\Sigma_J(\boldsymbol{a}) - \hat{\Sigma}_{J_k}(\boldsymbol{a})\right\|\right\} \left\|\mathbb{E}_p\left[r_{k-1}(\boldsymbol{x})\boldsymbol{\Phi}_k(\boldsymbol{a}^\top \boldsymbol{x})\right]\right\| \\
&= O_p\left(\sqrt{\xi_{n,k-1}}\zeta_0(J_k)^2 + \frac{\sqrt{J_k}\zeta_0(J_k)^2}{\sqrt{n_q}}\right) \cdot \sup_{\boldsymbol{a} \in \mathcal{A}} \left\|\mathbb{E}_p\left[r_{k-1}(\boldsymbol{x})\boldsymbol{\Phi}_k(\boldsymbol{a}^\top \boldsymbol{x})\right]\right\|,
\end{aligned} \tag{29}
$$

where the last equality follows from Lemma F.6 and the rate of $\lambda$ in Assumption F.3.

Let $V(\boldsymbol{a}) := \mathbb{E}_p\left[\mathbb{E}_p\{r_{k-1}(\boldsymbol{x})|\boldsymbol{a}^\top \boldsymbol{x}\}\boldsymbol{\Phi}_k(\boldsymbol{a}^\top \boldsymbol{x})\right]$. Under Assumption F.3, we have the eigenvalues of $\Omega_{J,p} := \mathbb{E}_p\{\boldsymbol{\Phi}_k(\boldsymbol{a}^\top \boldsymbol{x})\boldsymbol{\Phi}_k(\boldsymbol{a}^\top \boldsymbol{x})^\top\}$ are bounded and bounded away from 0 uniformly in $\boldsymbol{a} \in \mathcal{A}$. Then

$$
\begin{aligned}
\sup_{\boldsymbol{a} \in \mathcal{A}} \|V(\boldsymbol{a})\|^2 &= \sup_{\boldsymbol{a} \in \mathcal{A}} V(\boldsymbol{a})^\top V(\boldsymbol{a}) \\
&= O(1) \cdot \sup_{\boldsymbol{a} \in \mathcal{A}} V(\boldsymbol{a})^\top \Omega_{J,p}(\boldsymbol{a})^{-1}\Omega_{J,p}(\boldsymbol{a})\Omega_{J,p}(\boldsymbol{a})^{-1}V(\boldsymbol{a}) \\
&= O(1) \cdot \sup_{\boldsymbol{a} \in \mathcal{A}} \mathbb{E}_p\left[\{V(\boldsymbol{a})^\top \Omega_{J,p}(\boldsymbol{a})^{-1}\boldsymbol{\Phi}_k(\boldsymbol{a}^\top \boldsymbol{x})\}^2\right],
\end{aligned}
$$

which is the $L_2$-projection of $\mathbb{E}_p\{r_{k-1}(\boldsymbol{x})|\boldsymbol{a}^\top \boldsymbol{x}\}$ to the space linearly spanned by $\boldsymbol{\Phi}_k(\boldsymbol{a}^\top \boldsymbol{x})$. Therefore,

$$
\sup_{\boldsymbol{a} \in \mathcal{A}} \|V(\boldsymbol{a})\|^2 \leq \mathbb{E}_p\left[\mathbb{E}_p\{r_{k-1}(\boldsymbol{x})|\boldsymbol{a}^\top \boldsymbol{x}\}^2\right] = O(1).
$$

Combining the results of (23) to (25), we have

$$
(22) = O_p\left(\sqrt{\xi_{n,k-1}}\zeta_0^2(J_k) + \frac{\zeta_0^2(J_k)\sqrt{J_k}}{\sqrt{n_p \wedge n_q}}\right). 
$$

Consequently, we have (8).

## F.4. Proof of (9)

**Consistency.** We first show $\hat{\boldsymbol{a}}_k \overset{P}{\to} \boldsymbol{a}_k$ as $n_p, n_q \to \infty$. By definition, $\hat{\boldsymbol{a}}_k$ (resp. $\boldsymbol{a}_k$) is the unique minimizer of $\hat{\mathcal{L}}_k(\boldsymbol{a}, \hat{\boldsymbol{\beta}}(\boldsymbol{a}); \lambda)$ (resp. $H(f_{\boldsymbol{a},k}, \boldsymbol{a})$). From the theory of $M$-estimation (Van der Vaart, 2000, Theorem 5.7), if the following condition holds:

$$
\sup_{\boldsymbol{a} \in \mathcal{A}} \left|\hat{\mathcal{L}}_k(\boldsymbol{a}, \hat{\boldsymbol{\beta}}(\boldsymbol{a}); \lambda) - H(f_{\boldsymbol{a},k}, \boldsymbol{a})\right| \overset{P}{\to} 0
$$

as $n_p, n_q \to \infty$, then we have $\hat{\boldsymbol{a}}_k \overset{P}{\to} \boldsymbol{a}_k$. Since

$$
\hat{\mathcal{L}}_k(\boldsymbol{a}, \hat{\boldsymbol{\beta}}_k(\boldsymbol{a}), \lambda) = \frac{1}{n_q}\sum_{i=1}^{n_q} \hat{r}_{k-1}^2(\boldsymbol{x}_i^q)\hat{f}_{\boldsymbol{a},k}^2(\boldsymbol{a}^\top \boldsymbol{x}_i^q) - \frac{2}{n_p}\sum_{i=1}^{n_p} \hat{r}_{k-1}(\boldsymbol{x}_i^p)\hat{f}_{\boldsymbol{a},k}(\boldsymbol{a}^\top \boldsymbol{x}_i^p) + \lambda\hat{\boldsymbol{\beta}}_k(\boldsymbol{a})^\top \hat{\boldsymbol{\beta}}_k(\boldsymbol{a}),
$$

we can decompose

$$\sup_{\boldsymbol{a}\in\mathcal{A}}\left|\hat{\mathcal{L}}_k(\boldsymbol{a},\hat{\boldsymbol{\beta}}(\boldsymbol{a});\lambda)-H(f_{\boldsymbol{a},k},\boldsymbol{a})\right|$$

$$\leq \sup_{\boldsymbol{a}\in\mathcal{A}}\left|\frac{1}{n_q}\sum_{i=1}^{n_q}r_{k-1}^2(\boldsymbol{x}_i^q)f_{\boldsymbol{a},k}^2(\boldsymbol{a}^\top\boldsymbol{x}_i^q)-\mathbb{E}_q[r_{k-1}^2(\boldsymbol{x})f_{\boldsymbol{a},k}^2(\boldsymbol{a}^\top\boldsymbol{x})]\right|$$

$$+2\sup_{\boldsymbol{a}\in\mathcal{A}}\left|\frac{1}{n_p}\sum_{i=1}^{n_p}r_{k-1}(\boldsymbol{x}_i^p)f_{\boldsymbol{a},k}(\boldsymbol{a}^\top\boldsymbol{x}_i^p)-\mathbb{E}_p[r_{k-1}(\boldsymbol{x})f_{\boldsymbol{a},k}(\boldsymbol{a}^\top\boldsymbol{x})]\right| \tag{30}$$

$$+\sup_{\boldsymbol{a}\in\mathcal{A}}\left|\frac{1}{n_q}\sum_{i=1}^{n_q}\left[\hat{r}_{k-1}^2(\boldsymbol{x}_i^q)\hat{f}_{\boldsymbol{a},k}^2(\boldsymbol{a}^\top\boldsymbol{x}_i^q)-r_{k-1}^2(\boldsymbol{x}_i^q)f_{\boldsymbol{a},k}^2(\boldsymbol{a}^\top\boldsymbol{x}_i^q)\right]\right|$$

$$+2\sup_{\boldsymbol{a}\in\mathcal{A}}\left|\frac{1}{n_p}\sum_{i=1}^{n_p}\left[\hat{r}_{k-1}(\boldsymbol{x}_i^p)\hat{f}_{\boldsymbol{a},k}(\boldsymbol{a}^\top\boldsymbol{x}_i^p)-r_{k-1}(\boldsymbol{x}_i^p)f_{\boldsymbol{a},k}(\boldsymbol{a}^\top\boldsymbol{x}_i^p)\right]\right| \tag{31}$$

$$+\lambda\sup_{\boldsymbol{a}\in\mathcal{A}}\hat{\boldsymbol{\beta}}_k(\boldsymbol{a})^\top\hat{\boldsymbol{\beta}}_k(\boldsymbol{a})\,. \tag{32}$$

**Rate for** (30). As imposed in Assumptions F.1 and F.2, $\mathcal{A}$ is compact and $f_{\boldsymbol{a},k}(\boldsymbol{a}^\top\boldsymbol{x})$ is continuous in $\boldsymbol{a}$, applying the uniform law of large numbers (Newey & McFadden, 1994, Lemma 2.4), we have

$$(30) = O_p(1/\sqrt{n_q}+1/\sqrt{n_p}) = o_p(1)\,.$$

**Rate for** (31). Note that

$$\left|\frac{1}{n_q}\sum_{i=1}^{n_q}\left[\hat{r}_{k-1}^2(\boldsymbol{x}_i^q)\hat{f}_{\boldsymbol{a},k}^2(\boldsymbol{a}^\top\boldsymbol{x}_i^q)-r_{k-1}^2(\boldsymbol{x}_i^q)f_{\boldsymbol{a},k}^2(\boldsymbol{a}^\top\boldsymbol{x}_i^q)\right]\right| \leq \left|\frac{1}{n_q}\sum_{i=1}^{n_q}\left[\{\hat{r}_{k-1}^2(\boldsymbol{x}_i^q)-r_{k-1}^2(\boldsymbol{x}_i^q)\}\hat{f}_{\boldsymbol{a},k}^2(\boldsymbol{a}^\top\boldsymbol{x}_i^q)\right]\right|$$

$$+\left|\frac{1}{n_q}\sum_{i=1}^{n_q}\left[r_{k-1}^2(\boldsymbol{x}_i^q)\{\hat{f}_{\boldsymbol{a},k}^2(\boldsymbol{a}^\top\boldsymbol{x}_i^q)-f_{\boldsymbol{a},k}^2(\boldsymbol{a}^\top\boldsymbol{x}_i^q)\}\right]\right|\,.$$

We can apply the same decomposition to the second term of (31).

For any $k\in\{1,2,\ldots,K\}$, given (16) that $n_q^{-1}\sum_{i=1}^{n_q}|\hat{r}_{k-1}(\boldsymbol{x}_i^q)-r_{k-1}(\boldsymbol{x}_i^q)|^2 = O_p(\xi_{n,k-1})$, using (8) and Cauchy-Schwarz inequality, we can conclude that

$$(31) = O_p\left(J_k^{-s}\zeta_0(J_k)+\sqrt{\xi_{n,k-1}}\zeta_0^2(J_k)+\zeta_0^2(J_k)\sqrt{J_k/(n_q\wedge n_p)}\right) = o_p(1)\,.$$

**Rate for** (32). Using the rate of $\lambda$ in Assumption F.3 and Lemma F.11, we have $(32) = o_p(1)$.

Consequently, we have $\hat{\boldsymbol{a}}_k-\boldsymbol{a}_k\xrightarrow{p}0$.

**Convergence Rate.** Let $\hat{\ell}_k(\boldsymbol{a}) = \hat{\mathcal{L}}_k(\boldsymbol{a},\hat{\boldsymbol{\beta}}_k(\boldsymbol{a});\lambda)$. Since $\hat{\boldsymbol{a}}_k$ is the unique global minimizer of the context function $\hat{\ell}_k(\boldsymbol{a})$, by the first order condition, we have $\partial_{\boldsymbol{a}}\hat{\ell}_k(\hat{\boldsymbol{a}}_k)=0$. By applying the mean value theorem, we obtain

$$0 = \partial_{\boldsymbol{a}}\hat{\ell}_k(\hat{\boldsymbol{a}}_k) = \partial_{\boldsymbol{a}}\hat{\ell}_k(\boldsymbol{a}_k)+\partial_{\boldsymbol{a}}^2\hat{\ell}_k(\bar{\boldsymbol{a}}_k)(\hat{\boldsymbol{a}}_k-\boldsymbol{a}_k),$$

where $\bar{\boldsymbol{a}}$ lies between $\hat{\boldsymbol{a}}_k$ and $\boldsymbol{a}_k$. Hence,

$$\hat{\boldsymbol{a}}_k-\boldsymbol{a}_k = -\partial_{\boldsymbol{a}}^2\hat{\ell}_k(\bar{\boldsymbol{a}}_k)^{-1}\cdot\partial_{\boldsymbol{a}}\hat{\ell}_k(\boldsymbol{a}_k). \tag{33}$$

Equation (33) implies that the convergence rate of $\hat{\boldsymbol{a}}_k-\boldsymbol{a}_k$ is determined by the Hessian matrix $\partial_{\boldsymbol{a}}^2\hat{\ell}_k(\bar{\boldsymbol{a}}_k)$ and the score function $\partial_{\boldsymbol{a}}\hat{\ell}_k(\boldsymbol{a}_k)$. We derive the rate of these terms one by one. Recalling the definition of $\hat{\ell}_k(\boldsymbol{a})$ at the beginning of the subsection and applying the chain rule, we obtain

$$\partial_{\boldsymbol{a}}\hat{\ell}_k(\boldsymbol{a}) = \partial_1\hat{\mathcal{L}}_k(\boldsymbol{a},\hat{\boldsymbol{\beta}}_k(\boldsymbol{a});\lambda)+\partial_2\hat{\mathcal{L}}_k(\boldsymbol{a},\hat{\boldsymbol{\beta}}_k(\boldsymbol{a});\lambda)\partial_{\boldsymbol{a}}\hat{\boldsymbol{\beta}}_k(\boldsymbol{a})$$
$$= \partial_1\hat{\mathcal{L}}_k(\boldsymbol{a},\hat{\boldsymbol{\beta}}_k(\boldsymbol{a});\lambda)\,, \tag{34}$$

where for any bivariate function $g(x, y)$, $\partial_1 g(x, y)$ and $\partial_2 g(x, y)$ denotes the partial derivative of $g$ w.r.t. to the first and the second arguments, $x$ and $y$, respectively. The second equality follows from the fact that $\hat{\beta}_k(a)$ is the unique global minimizer of the convex function $\hat{\mathcal{L}}_k(a, \hat{\beta}_k(a); \lambda)$ for any $a \in \mathcal{A}$, so $\partial_2 \hat{\mathcal{L}}_k(a, \hat{\beta}_k(a); \lambda) = 0$ for any $a \in \mathcal{A}$. Then, we have

$$\partial_a^2 \hat{\ell}_k(a) = \partial_1^2 \hat{\mathcal{L}}_k(a, \hat{\beta}_k(a); \lambda) + \partial_2 \partial_1 \hat{\mathcal{L}}_k(a, \hat{\beta}_k(a); \lambda) \cdot \partial_a \hat{\beta}_k(a). \tag{35}$$

**Asymptotics of the Hessian matrix $\partial_a^2 \hat{\ell}_k(\bar{a}_k)$.** Taking derivative on both sides of $\partial_2 \hat{\mathcal{L}}_k(a, \hat{\beta}_k(a); \lambda) = 0$ w.r.t. $a$ yields

$$\partial_1 \partial_2 \hat{\mathcal{L}}_k(a, \hat{\beta}_k(a); \lambda) + \partial_2^2 \hat{\mathcal{L}}_k(a, \hat{\beta}_k(a); \lambda) \partial_a \hat{\beta}_k(a) = 0.$$

Since $\partial_2^2 \hat{\mathcal{L}}_k(a, \hat{\beta}_k(a); \lambda) = \hat{\Sigma}_{J_k, \lambda}(a)$, substituting this expression into the above equation, we obtain

$$\partial_a \hat{\beta}_k(a) = -\hat{\Sigma}_{J_k, \lambda}(a)^{-1} \partial_1 \partial_2 \hat{\mathcal{L}}_k(a, \hat{\beta}_k(a); \lambda). \tag{36}$$

Thus

$$\partial_a^2 \hat{\ell}_k(a) = \partial_1^2 \hat{\mathcal{L}}_k(a, \hat{\beta}_k(a); \lambda) - \partial_2 \partial_1 \hat{\mathcal{L}}_k(a, \hat{\beta}_k(a); \lambda) \hat{\Sigma}_{J_k, \lambda}(a)^{-1} \partial_1 \partial_2 \hat{\mathcal{L}}_k(a, \hat{\beta}_k(a); \lambda).$$

Note that

$$\partial_1^2 \hat{\mathcal{L}}_k(a, \hat{\beta}_k(a); \lambda) = \frac{2}{n_q} \sum_{i=1}^{n_q} \hat{r}_{k-1}^2(x_i^q) \{\hat{f}_{a,k}^{(1)}(a^\top x_i^q)\}^2 x_i^q (x_i^q)^\top$$

$$+ \frac{2}{n_q} \sum_{i=1}^{n_q} \hat{r}_{k-1}^2(x_i^q) \hat{f}_{a,k}(a^\top x_i^q) \hat{f}_{a,k}^{(2)}(a^\top x_i^q) x_i^q (x_i^q)^\top - \frac{2}{n_p} \sum_{i=1}^{n_p} \hat{r}_{k-1}(x_i^p) \hat{f}_{a,k}^{(2)}(a^\top x_i^p) x_i^p (x_i^p)^\top,$$

and

$$\partial_2 \partial_1 \hat{\mathcal{L}}_k(a, \hat{\beta}_k(a); \lambda)^\top = \partial_1 \partial_2 \hat{\mathcal{L}}_k(a, \hat{\beta}_k(a); \lambda) = \frac{2}{n_q} \sum_{i=1}^{n_q} \hat{r}_{k-1}^2(x_i^q) \hat{f}_{a,k}^{(1)}(a^\top x_i^q) \Phi_k(a^\top x_i^q)(x_i^q)^\top$$

$$+ \frac{2}{n_q} \sum_{i=1}^{n_q} \hat{r}_{k-1}^2(x_i^q) \hat{f}_{a,k}(a^\top x_i^q) \Phi_k^{(1)}(a^\top x_i^q)(x_i^q)^\top - \frac{2}{n_p} \sum_{i=1}^{n_p} \hat{r}_{k-1}(x_i^p) \Phi_k^{(1)}(a^\top x_i^p)(x_i^p)^\top.$$

Note that $\partial_2 \partial_1 \hat{\mathcal{L}}_k(a, \hat{\beta}_k(a); \lambda)^\top = \partial_1 \partial_2 \hat{\mathcal{L}}_k(a, \hat{\beta}_k(a); \lambda)$. Using calculations similar to those in the derivation for (31), the results of Lemmas F.6 and F.10, and the rate of $\lambda$ in Assumption F.3, we have

$$\left\| \partial_1^2 \hat{\mathcal{L}}_k(a, \hat{\beta}_k(a); \lambda) - \left( 2\mathbb{E}_q \left[ r_{k-1}^2(x) \{f_{a,k}^{*(1)}(a^\top x)\}^2 x(x)^\top \right] \right. \right.$$

$$\left. \left. + 2\mathbb{E}_q \left[ r_{k-1}^2(x) f_{a,k}^*(a^\top x) f_{a,k}^{*(2)}(a^\top x) x(x)^\top \right] - 2\mathbb{E}_p \left[ r_{k-1}(x) f_{a,k}^{*(2)}(a^\top x) x(x)^\top \right] \right) \right\|$$

$$= o_p(1),$$

$$\left\| \partial_1 \partial_2 \hat{\mathcal{L}}_k(a, \hat{\beta}_k(a); \lambda) - \left( 2\mathbb{E}_q \left[ r_{k-1}^2(x) f_{a,k}^{*(1)}(a^\top x) \Phi_k(a^\top x)(x)^\top \right] \right. \right. \tag{37}$$

$$\left. \left. + 2\mathbb{E}_q \left[ r_{k-1}^2(x) f_{a,k}^*(a^\top x) \Phi_k^{(1)}(a^\top x)(x)^\top \right] - 2\mathbb{E}_p \left[ r_{k-1}(x) \Phi_k^{(1)}(a^\top x)(x)^\top \right] \right) \right\|$$

$$= o_p(1),$$

and

$$\|\hat{\Sigma}_{J_k, \lambda}(a) - \Sigma_{J_k}(a)\| = o_p(1).$$

Consequently, we have,

$$\partial_a^2 \hat{\ell}_k(\bar{a}) = \partial_a^2 H^*(\boldsymbol{\beta}_k^*(\bar{a}), \bar{a}) + o_P(1) = \partial_a^2 H^*(\boldsymbol{\beta}_k^*(\boldsymbol{a}_k), \boldsymbol{a}_k) + o_P(1),$$

where the last equality follows from the fact that $\bar{a}$ lies between $\hat{a}_k$ and $\boldsymbol{a}_k$, and $\hat{a}_k - \boldsymbol{a}_k \xrightarrow{p} 0$ and the continuous mapping theory.

Using arguments similar to the proof of the consistency of $\hat{a}_k$ to $\boldsymbol{a}_k$, we have $\boldsymbol{a}_k^* - \boldsymbol{a}_k \to 0$. Thus,

$$\partial_a^2 \hat{\ell}_k(\bar{a}) = \partial_a^2 H^*(\boldsymbol{\beta}_k^*(\boldsymbol{a}_k^*), \boldsymbol{a}_k^*) + o_P(1).$$

Thus, under Assumption F.4, $\partial_a^2 \hat{\ell}_k(\bar{a})$ is asymptotically invertible and $\|\partial_a^2 \hat{\ell}_k(\bar{a})^{-1}\| = O_P(1)$.

**Asymptotics of the score function $\partial_a \hat{\ell}_k(\boldsymbol{a}_k)$.** Note that

$$\partial_a \hat{\ell}_k(\boldsymbol{a}_k) = \partial_1 \hat{\mathcal{L}}_k(\boldsymbol{a}_k, \hat{\boldsymbol{\beta}}_k(\boldsymbol{a}_k); \lambda)$$
$$= \frac{2}{n_q} \sum_{i=1}^{n_q} \hat{r}_{k-1}^2(\boldsymbol{x}_i^q) \hat{f}_{\boldsymbol{a}_k, k}(\boldsymbol{a}_k^\top \boldsymbol{x}_i^q) \hat{f}_{\boldsymbol{a}_k, k}^{(1)}(\boldsymbol{a}_k^\top \boldsymbol{x}_i^q)(\boldsymbol{x}_i^q)^\top - \frac{2}{n_p} \sum_{i=1}^{n_p} \hat{r}_{k-1}(\boldsymbol{x}_i^p) \hat{f}_{\boldsymbol{a}_k, k}^{(1)}(\boldsymbol{a}_k^\top \boldsymbol{x}_i^p)(\boldsymbol{x}_i^p)^\top.$$

Using (16), Lemmas F.9 and F.10, and the arguments similar to those in the proof of Lemma F.8, we have

$$\partial_a \hat{\ell}_k(\boldsymbol{a}_k) = 2\mathbb{E}_q \left[ r_{k-1}^2(\boldsymbol{x}) f_{\boldsymbol{a}_k, k}(\boldsymbol{a}_k^\top \boldsymbol{x}) f_{\boldsymbol{a}_k, k}^{(1)}(\boldsymbol{a}_k^\top \boldsymbol{x})(\boldsymbol{x})^\top \right] - 2\mathbb{E}_P \left[ r_{k-1}(\boldsymbol{x}) f_{\boldsymbol{a}_k, k}^{(1)}(\boldsymbol{a}_k^\top \boldsymbol{x})(\boldsymbol{x})^\top \right] \qquad (38)$$
$$+ O_P \left( \{ J_k^{-(s-1)} + \sqrt{\xi_{n,k-1}} \} \cdot \sqrt{\tilde{\zeta}_1(J_k)} + \sqrt{\tilde{\zeta}_1(J_k) \cdot J_k / (n_p \wedge n_q)} \right).$$

By the definition in (6), we have, for any $\boldsymbol{a} \in \mathcal{A}$,

$$2\mathbb{E}_q \left[ r_{k-1}^2(\boldsymbol{x}) f_{\boldsymbol{a}}(\boldsymbol{a}^\top \boldsymbol{x}) \varphi(\boldsymbol{a}^\top \boldsymbol{x}) \right] - 2\mathbb{E}_P \left[ r_{k-1}(\boldsymbol{x}) \varphi(\boldsymbol{a}^\top \boldsymbol{x}) \right] = 0 \qquad (39)$$

for all integrable $\varphi : \mathbb{R} \mapsto \mathbb{R}$. Moreover, by the definition in (7) and chain rule, we have

$$\partial_1 H(f_{\boldsymbol{a}}, \boldsymbol{a}_k) \cdot \partial_{\boldsymbol{a}} f_{\boldsymbol{a}} \Big|_{\boldsymbol{a} = \boldsymbol{a}_k} + \partial_2 H(f_{\boldsymbol{a}_k}, \boldsymbol{a}) \Big|_{\boldsymbol{a} = \boldsymbol{a}_k} = 0. \qquad (40)$$

Note that, for any $\boldsymbol{a} \in \mathcal{A}$,

$$\partial_1 H(f_{\boldsymbol{a}}, \boldsymbol{a}_k) \cdot \partial_{\boldsymbol{a}} f_{\boldsymbol{a}} = 2\mathbb{E}_q \left[ r_{k-1}^2(\boldsymbol{x}) f_{\boldsymbol{a}}(\boldsymbol{a}_k^\top \boldsymbol{x}) \partial_{\boldsymbol{a}} f_{\boldsymbol{a}}(\boldsymbol{a}_k^\top \boldsymbol{x}) \right] - 2\mathbb{E}_P \left[ r_{k-1}(\boldsymbol{x}) \partial_{\boldsymbol{a}} f_{\boldsymbol{a}}(\boldsymbol{a}_k^\top \boldsymbol{x}) \right].$$

Using (39), we have

$$\partial_1 H(f_{\boldsymbol{a}}, \boldsymbol{a}_k) \cdot \partial_{\boldsymbol{a}} f_{\boldsymbol{a}} \Big|_{\boldsymbol{a} = \boldsymbol{a}_k} = 0.$$

Then (40) implies that

$$\partial_2 H(f_{\boldsymbol{a}_k}, \boldsymbol{a}) \Big|_{\boldsymbol{a} = \boldsymbol{a}_k} = 2\mathbb{E}_q \left[ r_{k-1}^2(\boldsymbol{x}) f_{\boldsymbol{a}_k, k}(\boldsymbol{a}_k^\top \boldsymbol{x}) f_{\boldsymbol{a}_k, k}^{(1)}(\boldsymbol{a}_k^\top \boldsymbol{x})(\boldsymbol{x})^\top \right] - 2\mathbb{E}_P \left[ r_{k-1}(\boldsymbol{x}) f_{\boldsymbol{a}_k, k}^{(1)}(\boldsymbol{a}_k^\top \boldsymbol{x})(\boldsymbol{x})^\top \right] = 0,$$

which, combined with (38), implies that the score function

$$\partial_a \hat{\ell}_k(\boldsymbol{a}_k) = O_P \left( \{ J_k^{-(s-1)} + \sqrt{\xi_{n,k-1}} \} \cdot \sqrt{\tilde{\zeta}_1(J_k)} + \sqrt{\tilde{\zeta}_1(J_k) \cdot J_k / (n_p \wedge n_q)} \right).$$

Then we have

$$\|\hat{a}_k - \boldsymbol{a}_k\| = O_P \left( \{ J_k^{-(s-1)} + \sqrt{\xi_{n,k-1}} \} \cdot \sqrt{\tilde{\zeta}_1(J_k)} + \sqrt{\tilde{\zeta}_1(J_k) \cdot J_k / (n_p \wedge n_q)} \right).$$

## F.5. Technical Lemmas

**Lemma F.5.** *For any $k \in \{1, 2, \ldots, K\}$, $\log N_{[\,]}(\epsilon, \mathcal{H}_k, \|\cdot\|_\infty) = O\left[J_k \log\left\{\epsilon^{-1}/\zeta_0(J_k)\right\}\right]$.*

*Proof.* For a fixed $\epsilon > 0$, we choose $\epsilon/\{2\zeta_0(J_k)\}$-balls centering at $b_1, \ldots, b_{M_1} \in \mathbb{S}_{J_k}$ such that $\mathbb{S}_{J_k}$ can be covered, and $\epsilon/\{2\sup_{x \in \mathcal{X}} \|x\|\zeta_1(J_k)\}$-balls centering at $a_1, \ldots, a_{M_2}$ such that $\mathcal{A}$ can be covered. Using the triangle inequality and the mean value theorem, we deduce that, for any $(b, a) \in \mathbb{S}_{J_k} \times \mathcal{A}$, there exist $j \in \{1, \cdots, M_1\}$ and $\ell \in \{1, \cdots, M_2\}$, such that

$$\left|b^\top \Phi_k(a^\top x) - b_j^\top \Phi_k(a_\ell^\top x)\right|$$
$$\leq \|b - b_j\| \cdot \left\|\Phi_k(a^\top x)\right\| + \zeta_1(J_k) \cdot \|x\| \cdot \|a - a_\ell\|$$
$$\leq \|b - b_j\| \cdot \zeta_0(J_k) + \zeta_1(J_k) \cdot \|x\| \cdot \|a - a_\ell\| \leq \epsilon.$$

Hence, $\{[b_j^\top \Phi_k(a_\ell^\top x) - \epsilon, b_j^\top \Phi_k(a_\ell^\top x) + \epsilon]\}_{j,\ell}$ form a set of $\epsilon$-brackets that cover the space $\mathcal{H}_k$. Because $j$ ranges from 1 to $M_1$, and $\ell$ ranges from 1 to $M_2$, we have a total of $M_1 \times M_2$ brackets. Therefore,

$$N_{[\,]}(\epsilon, \mathcal{H}_k, \|\cdot\|_\infty) \leq M_1 \times M_2$$
$$= N(\epsilon/\{2\zeta_0(J_k)\}, \mathbb{S}_{J_k}, \|\cdot\|) \times N(\epsilon/\{2\sup_{x \in \mathcal{X}} \|x\|\zeta_1(J_k)\}, \mathcal{A}, \|\cdot\|).$$

Since $\mathbb{S}_{J_k}$ is a compact set in $\mathbb{R}^{J_k}$ and $\mathcal{A}$ is a compact set in $\mathbb{R}^d$, $N(\epsilon, \mathbb{S}_{J_k}, \|\cdot\|) \leq C\epsilon^{-CJ_k}$ and $N(\epsilon, \mathcal{A}, \|\cdot\|) \leq C\epsilon^{-Cd}$. Hence

$$\log N_{[\,]}(\epsilon, \mathcal{H}_k, \|\cdot\|_\infty) = O\left[-d\log\{\epsilon/\zeta_1(J_k)\} - J_k \log\{\epsilon/\zeta_0(J_k)\}\right]$$
$$= O\left[-J_k \log\{\epsilon/\zeta_0(J_k)\}\right].$$

$\square$

**Lemma F.6.** *For any $k \in \{1, 2, \ldots, K\}$, suppose* (16) *holds. Under Assumption F.3, for $j = 0, 1, 2$, we have*

$$\sup_{a \in \mathcal{A}} \|\hat{\Omega}_{J_k}^{(j)}(a) - \Omega_{J_k, p}^{(j)}(a)\| = O_P(\zeta_j(J_k)\sqrt{J_k/n}).$$

*Similarly, we have*

$$\sup_{a \in \mathcal{A}} \|\hat{\Sigma}_{J_k}(a) - \Sigma_{J_k}(a)\| = O_P\left(\sqrt{\xi_{n,k-1}}\zeta_0(J_k) + \frac{\zeta_0(J_k)\sqrt{J_k}}{\sqrt{n_p \wedge n_q}}\right).$$

*Furthermore, we have*

$$\eta_{\min}\{\hat{\Omega}_{J_k}^{(j)}(a)\} \geq \eta_{\min}\{\Omega_{J_k}^{(j)}(a)\} - O_P(\zeta_j(J_k)\sqrt{J_k/n}),$$

$$\eta_{\min}\{\hat{\Sigma}_{J_k}(a)\} \geq \eta_{\min}\{\Sigma_{J_k}(a)\} - O_P\left(\sqrt{\xi_{n,k-1}}\zeta_0(J_k) + \frac{\zeta_0(J_k)\sqrt{J_k}}{\sqrt{n_p \wedge n_q}}\right),$$

$$\eta_{\min}\{\hat{\Sigma}_{J_k, \lambda}(a)\} \geq \eta_{\min}\{\Sigma_{J_k}(a)\} - O_P\left(\sqrt{\xi_{n,k-1}}\zeta_0(J_k) + \frac{\zeta_0(J_k)\sqrt{J_k}}{\sqrt{n_p \wedge n_q}}\right),$$

*and*

$$\eta_{\max}\{\hat{\Omega}_{J_k}^{(j)}(a)\} \leq \eta_{\max}\{\Omega_{J_k}^{(j)}(a)\} + O_P(\zeta_j(J_k)\sqrt{J_k/n}),$$

$$\eta_{\max}\{\hat{\Sigma}_{J_k}(a)\} \leq \eta_{\max}\{\Sigma_{J_k}(a)\} + O_P\left(\sqrt{\xi_{n,k-1}}\zeta_0(J_k) + \frac{\zeta_0(J_k)\sqrt{J_k}}{\sqrt{n_p \wedge n_q}}\right),$$

$$\eta_{\max}\{\hat{\Sigma}_{J_k, \lambda}(a)\} \leq \eta_{\max}\{\Sigma_{J_k}(a)\} + O_P\left(\sqrt{\xi_{n,k-1}}\zeta_0(J_k) + \frac{\zeta_0(J_k)\sqrt{J_k}}{\sqrt{n_p \wedge n_q}}\right),$$

*uniformly in $a \in \mathcal{A}$.*

*Proof.* Under Assumption F.3, we have the eigenvalues of $\Omega_{J_k}^{(j)}$ are bounded for $j = 0, 1, 2$ uniformly in $\boldsymbol{a} \in \mathcal{A}$. We can then derive that

$$
\begin{aligned}
\sup_{\boldsymbol{a} \in \mathcal{A}} \mathbb{E} \left[ \|\hat{\Omega}_{J_k}^{(j)}(\boldsymbol{a}) - \Omega_{J_k}^{(j)}(\boldsymbol{a})\|^2 \right] &= \sup_{\boldsymbol{a} \in \mathcal{A}} \sum_{j=1}^{J_k} \sum_{\ell=1}^{J_k} \mathbb{E} \left[ \left\{ \frac{1}{n} \sum_{i=1}^{n} \phi_j^{(j)}(\boldsymbol{a}^\top \boldsymbol{x}_i) \phi_\ell^{(j)}(\boldsymbol{a}^\top \boldsymbol{x}_i) - \mathbb{E}[\phi_j^{(j)}(\boldsymbol{a}^\top \boldsymbol{x}) \phi_\ell^{(j)}(\boldsymbol{a}^\top \boldsymbol{x})] \right\}^2 \right] \\
&\leq \sup_{\boldsymbol{a} \in \mathcal{A}} \sum_{j=1}^{J_k} \sum_{\ell=1}^{J_k} \frac{1}{n} \mathbb{E}[\{\phi_j^{(j)}(\boldsymbol{a}^\top \boldsymbol{x})\}^2 \{\phi_\ell^{(j)}(\boldsymbol{a}^\top \boldsymbol{x})\}^2] \\
&= \sup_{\boldsymbol{a} \in \mathcal{A}} \frac{1}{n} \mathbb{E} \left[ \sum_{j=1}^{J_k} \{\phi_j^{(j)}(\boldsymbol{a}^\top \boldsymbol{x})\}^2 \sum_{\ell=1}^{J_k} \{\phi_\ell^{(j)}(\boldsymbol{a}^\top \boldsymbol{x})\}^2 \right] \\
&= \sup_{\boldsymbol{a} \in \mathcal{A}} \frac{1}{n} \mathbb{E} \left[ \boldsymbol{\Phi}_k^{(j)}(\boldsymbol{a}^\top \boldsymbol{x})^\top \boldsymbol{\Phi}_k^{(j)}(\boldsymbol{a}^\top \boldsymbol{x}) \boldsymbol{\Phi}_k^{(j)}(\boldsymbol{a}^\top \boldsymbol{x})^\top \boldsymbol{\Phi}_k^{(j)}(\boldsymbol{a}^\top \boldsymbol{x}) \right] \\
&\leq \frac{\zeta_j(J_k)^2}{n} \sup_{\boldsymbol{a} \in \mathcal{A}} \mathrm{tr} \left( \mathbb{E} \left[ \boldsymbol{\Phi}_k^{(j)}(\boldsymbol{a}^\top \boldsymbol{x}) \boldsymbol{\Phi}_k^{(j)}(\boldsymbol{a}^\top \boldsymbol{x})^\top \right] \right) = O\left( \frac{\zeta_j(J_k)^2 J_k}{n} \right).
\end{aligned}
$$

Thus,

$$
\sup_{\boldsymbol{a} \in \mathcal{A}} \|\hat{\Omega}_{J_k}^{(j)}(\boldsymbol{a}) - \Omega_{J_k}^{(j)}(\boldsymbol{a})\| = O_P(\zeta_j(J_k) \sqrt{J_k/n}).
$$

It then follows from the definition of the maximum and minimum eigenvalues that

$$
\eta_{\min}\{\hat{\Omega}_{J_k}^{(j)}(\boldsymbol{a})\} = \min_{\|\boldsymbol{\beta}^\top \boldsymbol{\beta}\| = 1} \{\boldsymbol{\beta}^\top \Omega_{J_k}^{(j)}(\boldsymbol{a})\boldsymbol{\beta} + \boldsymbol{\beta}^\top [\hat{\Omega}_{J_k}^{(j)}(\boldsymbol{a}) - \Omega_{J_k}^{(j)}(\boldsymbol{a})]\boldsymbol{\beta}\} \geq \eta_{\min}\{\Omega_{J_k}^{(j)}(\boldsymbol{a})\} - O_P(\zeta_j(J_k) \sqrt{J_k/n}),
$$

and

$$
\eta_{\max}\{\hat{\Omega}_{J_k}^{(j)}(\boldsymbol{a})\} \leq \eta_{\max}\{\Omega_{J_k}^{(j)}(\boldsymbol{a})\} + O_P(\zeta_j(J_k) \sqrt{J_k/n}),
$$

uniformly in $\boldsymbol{a} \in \mathcal{A}$.

For $\hat{\Sigma}_{J_k}(\boldsymbol{a})$, we first define

$$
\tilde{\Sigma}_{J_k}(\boldsymbol{a}) := \frac{1}{n_q} \sum_{i=1}^{n_q} r_{k-1}(\boldsymbol{x}_i^q) \boldsymbol{\Phi}_k(\boldsymbol{a}^\top \boldsymbol{x}_i^q) \boldsymbol{\Phi}_k(\boldsymbol{a}^\top \boldsymbol{x}_i^q)^\top.
$$

Then, we have

$$
\begin{aligned}
\hat{\Sigma}_{J_k}(\boldsymbol{a}) - \Sigma_{J_k}(\boldsymbol{a}) =& \hat{\Sigma}_{J_k}(\boldsymbol{a}) - \tilde{\Sigma}_{J_k}(\boldsymbol{a}) + \tilde{\Sigma}_{J_k}(\boldsymbol{a}) - \Sigma_{J_k}(\boldsymbol{a}) \\
=& \frac{1}{n_q} \sum_{i=1}^{n_q} \{\hat{r}_{k-1}(\boldsymbol{x}_i^q) - r_{k-1}(\boldsymbol{x}_i^q)\} \boldsymbol{\Phi}_k(\boldsymbol{a}^\top \boldsymbol{x}_i^q) \boldsymbol{\Phi}_k(\boldsymbol{a}^\top \boldsymbol{x}_i^q)^\top \quad (41) \\
&+ \frac{1}{n_q} \sum_{i=1}^{n_q} r_{k-1}(\boldsymbol{x}_i^q) \boldsymbol{\Phi}_k(\boldsymbol{a}^\top \boldsymbol{x}_i^q) \boldsymbol{\Phi}_k(\boldsymbol{a}^\top \boldsymbol{x}_i^q)^\top - \mathbb{E}_q[r_{k-1}(\boldsymbol{x}) \boldsymbol{\Phi}_k(\boldsymbol{a}^\top \boldsymbol{x}) \boldsymbol{\Phi}_k(\boldsymbol{a}^\top \boldsymbol{x})^\top]. \quad (42)
\end{aligned}
$$

For (41), using (16) and Cauchy-Schwarz inequality, we can derive

$$
\begin{aligned}
\sup_{\boldsymbol{a} \in \mathcal{A}} \|(41)\| &\leq \sqrt{\frac{1}{n_q} \sum_{i=1}^{n_q} \{\hat{r}_{k-1}(\boldsymbol{x}_i^q) - r_{k-1}(\boldsymbol{x}_i^q)\}^2} \cdot \sup_{\boldsymbol{a} \in \mathcal{A}} \sqrt{\eta_{\max}(\hat{\Omega}_{J_k}(\boldsymbol{a}))} \cdot \sup_{\boldsymbol{a} \in \mathcal{A}} \sup_{\boldsymbol{x} \in \mathcal{X}} \|\boldsymbol{\Phi}_k(\boldsymbol{a}^\top \boldsymbol{x})\| \\
&= O_P\left( \sqrt{\xi_{n,k-1}} \cdot \zeta_0(J_k) \right).
\end{aligned}
$$

For (42), using the same arguments for $\hat{\Omega}_{J_k}(\boldsymbol{a})$, we have

$$
\sup_{\boldsymbol{a} \in \mathcal{A}} \|(42)\| = O_P(\zeta_0(J_k) \sqrt{J_k/n_q}).
$$

The result then follows.

The results for $\hat{\Sigma}_{J_k,\lambda}(\boldsymbol{a})$ follows from the fact that $\hat{\Sigma}_{J_k,\lambda}(\boldsymbol{a}) = \hat{\Sigma}_{J_k}(\boldsymbol{a}) + \lambda \boldsymbol{I}_{J_k}$ and the rate of $\lambda$ in Assumption F.3.

$\square$

**Lemma F.7.** *For any $k \in \{1, 2, \ldots, K\}$, suppose Assumptions F.2 – F.4 and (16) hold. Then,*

$$\sup_{\boldsymbol{a} \in \mathcal{A}} \|\boldsymbol{\beta}_k^*(\boldsymbol{a}) - \boldsymbol{\beta}_{k,0}(\boldsymbol{a})\| = O(J_k^{-s}).$$

*Proof.* Note that $H^*(\boldsymbol{\beta}, \boldsymbol{a})$ is convex in $\boldsymbol{\beta}$ under Assumption F.3, and has a unique global minimizer $\boldsymbol{\beta}_k^*(\boldsymbol{a})$. Given a constant $d > C \cdot J_k^{-s}$ for some constant $C > 0$ (to be chosen later), for any $\boldsymbol{\beta}$ satisfying $\|\boldsymbol{\beta} - \boldsymbol{\beta}_{k,0}(\boldsymbol{a})\| = d$, we have

$$\left(1 - \frac{CJ_k^{-s}}{d}\right) H^*(\boldsymbol{\beta}_{k,0}(\boldsymbol{a}), \boldsymbol{a}) + \frac{CJ_k^{-s}}{d} H^*(\boldsymbol{\beta}, \boldsymbol{a}) \geq H^*\left(\boldsymbol{\beta}_{k,0}(\boldsymbol{a}) - \frac{CJ_k^{-s}}{d}\{\boldsymbol{\beta}_{k,0}(\boldsymbol{a}) - \boldsymbol{\beta}\}, \boldsymbol{a}\right).$$

Then, we have

$$\frac{CJ_k^{-s}}{d}\left[H^*(\boldsymbol{\beta}, \boldsymbol{a}) - H^*(\boldsymbol{\beta}_{k,0}(\boldsymbol{a}), \boldsymbol{a})\right] \geq H^*\left(\boldsymbol{\beta}_{k,0}(\boldsymbol{a}) - \frac{CJ_k^{-s}}{d}\{\boldsymbol{\beta}_{k,0}(\boldsymbol{a}) - \boldsymbol{\beta}\}, \boldsymbol{a}\right) - H^*(\boldsymbol{\beta}_{k,0}(\boldsymbol{a}), \boldsymbol{a})$$

$$= H\left(f_{\boldsymbol{a},k} - \frac{CJ_k^{-s}}{d}\{\boldsymbol{\beta}_{k,0}(\boldsymbol{a}) - \boldsymbol{\beta}\}^\top \boldsymbol{\Phi}_k, \boldsymbol{a}\right) - H(f_{\boldsymbol{a},k}, \boldsymbol{a})$$

$$- \xi\left(\frac{CJ_k^{-s}}{d}\{\boldsymbol{\beta}_{k,0}(\boldsymbol{a}) - \boldsymbol{\beta}\}\right) + \xi(0),$$

where $\xi(\boldsymbol{\theta}) := H\left(f_{\boldsymbol{a},k} - \boldsymbol{\theta}^\top \boldsymbol{\Phi}_k, \boldsymbol{a}\right) - H^*(\boldsymbol{\beta}_{k,0}(\boldsymbol{a}) - \boldsymbol{\theta}, \boldsymbol{a})$ for any $\boldsymbol{\theta} \in \mathbb{R}^{J_k}$. Note that $f_{\boldsymbol{a},k}$ is the minimizer of $H(f, \boldsymbol{a})$. Under Assumption F.3, $H(f_{\boldsymbol{a},k} - \boldsymbol{\theta}^\top \boldsymbol{\Phi}_k, \boldsymbol{a})$ is globally convex in $\boldsymbol{\theta}$ and attains the minimum at $\boldsymbol{\theta} = 0$. Applying the Taylor's expansion of $H(f_{\boldsymbol{a},k} - \boldsymbol{\theta}^\top \boldsymbol{\Phi}_k, \boldsymbol{a})$ around $\boldsymbol{\theta} = 0$, we have

$$\inf_{\{\boldsymbol{\beta} : \|\boldsymbol{\beta} - \boldsymbol{\beta}_{k,0}(\boldsymbol{a})\| = d\}} \left[H\left(f_{\boldsymbol{a},k} - \frac{CJ_k^{-s}}{d}\{\boldsymbol{\beta}_{k,0}(\boldsymbol{a}) - \boldsymbol{\beta}\}^\top \boldsymbol{\Phi}_k, \boldsymbol{a}\right) - H(f_{\boldsymbol{a},k}, \boldsymbol{a})\right]$$

$$= \inf_{\{\boldsymbol{\beta} : \|\boldsymbol{\beta} - \boldsymbol{\beta}_{k,0}(\boldsymbol{a})\| = d\}} \frac{C^2 J_k^{-2s}}{d^2}\{\boldsymbol{\beta}_{k,0}(\boldsymbol{a}) - \boldsymbol{\beta}\}^\top \mathbb{E}[r_{k-1}^2(\boldsymbol{x}) \boldsymbol{\Phi}_k(\boldsymbol{a}^\top \boldsymbol{x}) \boldsymbol{\Phi}_k(\boldsymbol{a}^\top \boldsymbol{x})^\top]\{\boldsymbol{\beta}_{k,0}(\boldsymbol{a}) - \boldsymbol{\beta}\}$$

$$\geq \eta_1 C^2 J_k^{-2s},$$

where $\eta_1 > 0$ is the minimum eigenvalue of $\mathbb{E}[r_{k-1}^2(\boldsymbol{x}) \boldsymbol{\Phi}_k(\boldsymbol{a}^\top \boldsymbol{x}) \boldsymbol{\Phi}_k(\boldsymbol{a}^\top \boldsymbol{x})^\top]$ under Assumptions F.2 and F.3.

By the definition of $\xi(\boldsymbol{\theta})$, we can derive that, for $\|\boldsymbol{\beta} - \boldsymbol{\beta}_{k,0}(\boldsymbol{a})\| = d$,

$$\left|\xi\left(\frac{CJ_k^{-s}}{d}\{\boldsymbol{\beta}_{k,0}(\boldsymbol{a}) - \boldsymbol{\beta}\}\right) - \xi(0)\right| = \left|\frac{2CJ_k^{-s}}{d}\{\boldsymbol{\beta}_{k,0}(\boldsymbol{a}) - \boldsymbol{\beta}\}^\top \mathbb{E}_q\left[r_{k-1}^2(\boldsymbol{x}) \boldsymbol{\Phi}_k(\boldsymbol{a}^\top \boldsymbol{x})\{f_{\boldsymbol{a},k}(\boldsymbol{a}^\top \boldsymbol{x}) - \boldsymbol{\beta}_{k,0}(\boldsymbol{a})^\top \boldsymbol{\Phi}_k(\boldsymbol{a}^\top \boldsymbol{x})\}\right]\right|$$

$$\leq 2\eta_2 C C_0 J_k^{-2s},$$

uniformly in $\boldsymbol{a} \in \mathcal{A}$ and $\boldsymbol{x} \in \mathcal{X}$, where $\eta_2$ is the maximum eigenvalue of $\mathbb{E}[r_{k-1}^2(\boldsymbol{x}) \boldsymbol{\Phi}_k(\boldsymbol{a}^\top \boldsymbol{x}) \boldsymbol{\Phi}_k(\boldsymbol{a}^\top \boldsymbol{x})^\top]$ under Assumptions F.2 and F.3. By choosing $C \geq 2\eta_2 C_0 / \eta_1$, we have $H^*(\boldsymbol{\beta}, \boldsymbol{a}) - H^*(\boldsymbol{\beta}_{k,0}(\boldsymbol{a}), \boldsymbol{a}) \geq 0$ for all $\|\boldsymbol{\beta} - \boldsymbol{\beta}_{k,0}(\boldsymbol{a})\| > C \cdot J_k^{-s}$ uniformly in $\boldsymbol{a} \in \mathcal{A}$ and $\boldsymbol{x} \in \mathcal{X}$, which implies that $H^*(\boldsymbol{\beta}, \boldsymbol{a})$ has a local minimum for $\|\boldsymbol{\beta} - \boldsymbol{\beta}_{k,0}(\boldsymbol{a})\| \leq C \cdot J_k^{-s}$. Since $\boldsymbol{\beta}_k^*(\boldsymbol{a})$ is the unique global minimizer of $H^*(\boldsymbol{\beta}, \boldsymbol{a})$, we have

$$\sup_{\boldsymbol{a} \in \mathcal{A}} \|\boldsymbol{\beta}_k^*(\boldsymbol{a}) - \boldsymbol{\beta}_{k,0}(\boldsymbol{a})\| \leq C \cdot J_k^{-s}. \tag{43}$$

$\square$

**Lemma F.8.** *For any $k \in \{1, 2, \ldots, K\}$, suppose Assumptions F.2 – F.4 and (16) hold. For any $\boldsymbol{a} \in \mathcal{A}$, we have*

$$\|\hat{\boldsymbol{\beta}}_k(\boldsymbol{a}) - \boldsymbol{\beta}_k^*(\boldsymbol{a})\| = O_P\left(J_k^{-(s-1)} + \sqrt{\xi_{n,k-1}} + \frac{\sqrt{J_k}}{\sqrt{n_q \wedge n_p}}\right).$$

*Proof.* Recall that

$$\boldsymbol{\beta}_k^*(\boldsymbol{a}) = \{\Sigma_{J_k}(\boldsymbol{a})\}^{-1}\mathbb{E}_p[r_{k-1}(\boldsymbol{x})\boldsymbol{\Phi}_k(\boldsymbol{a}^\top\boldsymbol{x})] \quad \text{and} \quad \hat{\boldsymbol{\beta}}_k(\boldsymbol{a}) = \{\hat{\Sigma}_{J_k,\lambda}(\boldsymbol{a})\}^{-1}\frac{1}{n_P}\sum_{i=1}^{n_P}\hat{r}_{k-1}(\boldsymbol{x}_i^p)\boldsymbol{\Phi}_k(\boldsymbol{a}^\top\boldsymbol{x}_i^p).$$

We decompose

$$\left\|\hat{\boldsymbol{\beta}}_k(\boldsymbol{a}) - \boldsymbol{\beta}^*(\boldsymbol{a})\right\| \tag{44}$$

$$= \left\|\{\hat{\Sigma}_{J_k,\lambda}(\boldsymbol{a})\}^{-1}\frac{1}{n_P}\sum_{i=1}^{n_P}\hat{r}_{k-1}(\boldsymbol{x}_i^p)\boldsymbol{\Phi}_k(\boldsymbol{a}^\top\boldsymbol{x}_i^p) - \boldsymbol{\beta}_k^*(\boldsymbol{a})\right\|$$

$$\leq \left\|\{\hat{\Sigma}_{J_k,\lambda}(\boldsymbol{a})\}^{-1}\frac{1}{n_P}\sum_{i=1}^{n_P}\{\hat{r}_{k-1}(\boldsymbol{x}_i^p) - r_{k-1}(\boldsymbol{x}_i^p)\}\boldsymbol{\Phi}_k(\boldsymbol{a}^\top\boldsymbol{x}_i^p)\right\| \tag{45}$$

$$+ \left\|\{\hat{\Sigma}_{J_k,\lambda}(\boldsymbol{a})\}^{-1}\frac{1}{n_q}\sum_{i=1}^{n_q}\{\hat{r}_{k-1}^2(\boldsymbol{x}_i^q) - r_{k-1}^2(\boldsymbol{x}_i^q)\}\boldsymbol{\Phi}_k(\boldsymbol{a}^\top\boldsymbol{x}_i^q)f_{\boldsymbol{a}}^*(\boldsymbol{a}^\top\boldsymbol{x}_i^q)\right\| \tag{46}$$

$$+ \left\|\{\hat{\Sigma}_{J_k,\lambda}(\boldsymbol{a})\}^{-1}\frac{1}{n_P}\sum_{i=1}^{n_P}r_{k-1}(\boldsymbol{x}_i^p)\boldsymbol{\Phi}_k(\boldsymbol{a}^\top\boldsymbol{x}_i^p) - \{\hat{\Sigma}_{J_k,\lambda}(\boldsymbol{a})\}^{-1}\frac{1}{n_q}\sum_{i=1}^{n_q}r_{k-1}^2(\boldsymbol{x}_i^q)\boldsymbol{\Phi}_k(\boldsymbol{a}^\top\boldsymbol{x}_i^q)\boldsymbol{\Phi}_k(\boldsymbol{a}^\top\boldsymbol{x}_i^q)^\top\boldsymbol{\beta}_k^*(\boldsymbol{a})\right\| \tag{47}$$

$$+ \lambda\left\|\boldsymbol{\beta}_k^*(\boldsymbol{a})\right\|. \tag{48}$$

**For** (45). Defining $\hat{\boldsymbol{r}} = \{\hat{r}_{k-1}(\boldsymbol{x}_1^p), \ldots, \hat{r}_{k-1}(\boldsymbol{x}_{n_P}^p)\} \in \mathbb{R}^{n_P}$ and $\boldsymbol{r} = \{r_{k-1}(\boldsymbol{x}_1^p), \ldots, r_{k-1}(\boldsymbol{x}_{n_P}^p)\} \in \mathbb{R}^{n_P}$, by Lemma F.6, we have $\eta_{\max}(\{\hat{\Sigma}_{J_k,\lambda}(\boldsymbol{a})\}^{-1}) = O(1)$. Thus,

$$\begin{aligned}\|(45)\|^2 &= n_P^{-2}\mathrm{tr}\left(\{\hat{\Sigma}_{J_k,\lambda}(\boldsymbol{a})\}^{-2}\boldsymbol{P}(\boldsymbol{a})^\top(\hat{\boldsymbol{r}}-\boldsymbol{r})(\hat{\boldsymbol{r}}-\boldsymbol{r})^\top\boldsymbol{P}(\boldsymbol{a})\right)\\ &\leq O_p(n_P^{-2})\mathrm{tr}\left(\boldsymbol{P}(\boldsymbol{a})^\top(\hat{\boldsymbol{r}}-\boldsymbol{r})(\hat{\boldsymbol{r}}-\boldsymbol{r})^\top\boldsymbol{P}(\boldsymbol{a})\right)\\ &= O_p(n_P^{-1})\mathrm{tr}\left(\boldsymbol{P}(\boldsymbol{a})^\top(\hat{\boldsymbol{r}}-\boldsymbol{r})(\hat{\boldsymbol{r}}-\boldsymbol{r})^\top\boldsymbol{P}(\boldsymbol{a})(\boldsymbol{P}(\boldsymbol{a})^\top\boldsymbol{P}(\boldsymbol{a}))^{-1}\right)\\ &= O_p(n_P^{-1})\mathrm{tr}\left((\hat{\boldsymbol{r}}-\boldsymbol{r})(\hat{\boldsymbol{r}}-\boldsymbol{r})^\top\boldsymbol{P}(\boldsymbol{a})(\boldsymbol{P}(\boldsymbol{a})^\top\boldsymbol{P}(\boldsymbol{a}))^{-1}\boldsymbol{P}(\boldsymbol{a})^\top\right)\\ &\leq O_p(n_P^{-1})\cdot\|\hat{\boldsymbol{r}}-\boldsymbol{r}\|^2 = O_P\left(J_k^{-2(s-1)} + \xi_{n,k-1} + J_k/(n_P \wedge n_q)\right),\end{aligned}$$

where the last inequality follows from the fact that $\boldsymbol{P}(\boldsymbol{a})(\boldsymbol{P}(\boldsymbol{a})^\top\boldsymbol{P}(\boldsymbol{a}))^{-1}\boldsymbol{P}(\boldsymbol{a})^\top$ is a projection matrix with maximum eigenvalue 1, and last equality follows from (16). Using exactly the same arguments and the fact that $\sup_{\boldsymbol{a},\boldsymbol{x}}|f_{\boldsymbol{a}}^*(\boldsymbol{a}^\top\boldsymbol{x})| = O(1)$, we have $(46) = O_p((45))$.

**For** (47)**,** we have

$$\{\hat{\Sigma}_{J_k,\lambda}(\boldsymbol{a})\}^{-1}\left\{\frac{1}{n_P}\sum_{i=1}^{n_P}r_{k-1}(\boldsymbol{x}_i^p)\boldsymbol{\Phi}_k(\boldsymbol{a}^\top\boldsymbol{x}_i^p) - \frac{1}{n_q}\sum_{i=1}^{n_q}r_{k-1}^2(\boldsymbol{x}_i^q)\boldsymbol{\Phi}_k(\boldsymbol{a}^\top\boldsymbol{x}_i^q)\boldsymbol{\Phi}_k(\boldsymbol{a}^\top\boldsymbol{x}_i^q)^\top\boldsymbol{\beta}_k^*(\boldsymbol{a})\right\}$$

$$= \{\hat{\Sigma}_{J_k,\lambda}(\boldsymbol{a})\}^{-1}\left\{\frac{1}{n_P}\sum_{i=1}^{n_P}r_{k-1}(\boldsymbol{x}_i^p)\boldsymbol{\Phi}_k(\boldsymbol{a}^\top\boldsymbol{x}_i^p) - \mathbb{E}_p\{r_{k-1}(\boldsymbol{x})\boldsymbol{\Phi}_k(\boldsymbol{a}^\top\boldsymbol{x})\}\right\} \tag{49}$$

$$+ \{\hat{\Sigma}_{J_k,\lambda}(\boldsymbol{a})\}^{-1}\left\{\mathbb{E}_q\{r_{k-1}^2(\boldsymbol{x})\boldsymbol{\Phi}_k(\boldsymbol{a}^\top\boldsymbol{x})\boldsymbol{\Phi}_k(\boldsymbol{a}^\top\boldsymbol{x})^\top\boldsymbol{\beta}_k^*(\boldsymbol{a})\} - \frac{1}{n_q}\sum_{i=1}^{n_q}r_{k-1}^2(\boldsymbol{x}_i^q)\boldsymbol{\Phi}_k(\boldsymbol{a}^\top\boldsymbol{x}_i^q)\boldsymbol{\Phi}_k(\boldsymbol{a}^\top\boldsymbol{x}_i^q)^\top\boldsymbol{\beta}_k^*(\boldsymbol{a})\right\} \tag{50}$$

$$+ \{\hat{\Sigma}_{J_k,\lambda}(\boldsymbol{a})\}^{-1}\left[\mathbb{E}_p\{r_{k-1}(\boldsymbol{x})\boldsymbol{\Phi}_k(\boldsymbol{a}^\top\boldsymbol{x})\} - \mathbb{E}_q\{r_{k-1}^2(\boldsymbol{x})\boldsymbol{\Phi}_k(\boldsymbol{a}^\top\boldsymbol{x})\boldsymbol{\Phi}_k(\boldsymbol{a}^\top\boldsymbol{x})^\top\boldsymbol{\beta}_k^*(\boldsymbol{a})\}\right]. \tag{51}$$

Note that

$$\mathbb{E}\left[\left\|\frac{1}{n_p}\sum_{i=1}^{n_p}r_{k-1}(\boldsymbol{x}_i^p)\boldsymbol{\Phi}_k(\boldsymbol{a}^\top\boldsymbol{x}_i^p)-\mathbb{E}_p\{r_{k-1}(\boldsymbol{x})\boldsymbol{\Phi}_k(\boldsymbol{a}^\top\boldsymbol{x})\}\right\|^2\right]$$

$$=\frac{1}{n_p}\mathbb{E}_p\left[\left\|r_{k-1}(\boldsymbol{x})\boldsymbol{\Phi}_k(\boldsymbol{a}^\top\boldsymbol{x})-\mathbb{E}_p\{r_{k-1}(\boldsymbol{x})\boldsymbol{\Phi}_k(\boldsymbol{a}^\top\boldsymbol{x})\}\right\|^2\right]$$

$$\leq\frac{1}{n_p}\mathbb{E}_p\left[\left\|r_{k-1}(\boldsymbol{x})\boldsymbol{\Phi}_k(\boldsymbol{a}^\top\boldsymbol{x})\right\|^2\right]$$

$$=\frac{1}{n_p}\text{tr}\left(\mathbb{E}_p[r_{k-1}^2(\boldsymbol{x})\boldsymbol{\Phi}_k(\boldsymbol{a}^\top)\boldsymbol{\Phi}_k(\boldsymbol{a}^\top)^\top]\right)=O\left(\frac{J_k}{n_p}\right),$$

where the last equality follows from Assumptions F.2 and F.3. Thus, by Chebyshev's inequality, we have

$$\|(49)\|=O_P\left(\frac{\sqrt{J_k}}{\sqrt{n_p}}\right).$$

Similarly, we have

$$\|(50)\|=O_P\left(\frac{\sqrt{J_k}}{\sqrt{n_q}}\right).$$

For (51), noting that $\boldsymbol{\beta}^*(\boldsymbol{a})$ is the globally unique minimizer of $H^*(\boldsymbol{a},\boldsymbol{\beta})$ for any $\boldsymbol{a}\in\mathcal{A}$, by the first order condition, we have

$$0=\partial_1 H^*(\boldsymbol{\beta}_k^*(\boldsymbol{a}),\boldsymbol{a})=2\left[\mathbb{E}_q\{r_{k-1}^2(\boldsymbol{x})\boldsymbol{\Phi}_k(\boldsymbol{a}^\top\boldsymbol{x})\boldsymbol{\Phi}_k(\boldsymbol{a}^\top\boldsymbol{x})^\top\boldsymbol{\beta}_k^*(\boldsymbol{a})\}-\mathbb{E}_p\{r_{k-1}(\boldsymbol{x})\boldsymbol{\Phi}_k(\boldsymbol{a}^\top\boldsymbol{x})\}\right].$$

Thus, we have (51) = 0. Consequently, we have

$$\|(47)\|=O_P\left(\frac{\sqrt{J_k}}{\sqrt{n_p\wedge n_q}}\right).$$

Finally, under Assumption F.3, we have (48) $=O_P(\sqrt{J_k/n_q})$. Thus, we obtain the result. $\qquad\square$

**Lemma F.9.** *For any $k\in\{1,2,\ldots,K\}$, suppose Assumptions F.1 – F.4 and (16) hold. For any $\boldsymbol{a}\in\mathcal{A}$ and $j=0,1$,*

$$\sup_{\boldsymbol{a}\in\mathcal{A}}\sup_{\boldsymbol{x}\in\mathcal{X}}\left|f_{\boldsymbol{a},k}^{(j)}(\boldsymbol{a}^\top\boldsymbol{x})-f_{\boldsymbol{a},k}^{*(j)}(\boldsymbol{a}^\top\boldsymbol{x})\right|=O\left(J_k^{-(s-j)}\zeta_j(J_k)\right),$$

$$\mathbb{E}\left[|f_{\boldsymbol{a},k}^{(j)}(\boldsymbol{a}^\top\boldsymbol{x})-f_{\boldsymbol{a},k}^{*(j)}(\boldsymbol{a}^\top\boldsymbol{x})|^2\right]=O\left(J_k^{-2(s-j)}\tilde{\zeta}_j(J_k)\right),$$

$$\frac{1}{n}\sum_{i=1}^{n}|f_{\boldsymbol{a},k}^{(j)}(\boldsymbol{a}^\top\boldsymbol{x}_i)-f_{\boldsymbol{a},k}^{*(j)}(\boldsymbol{a}^\top\boldsymbol{x}_i)|^2=O_P\left(J_k^{-2(s-j)}\tilde{\zeta}_j(J_k)\right).$$

*Proof.* Note that by Lorentz (1986, Theorem 8) and Assumption F.2, for $j<s_1$, there exists a $\boldsymbol{\beta}_{k,j}(\boldsymbol{a})\in\mathbb{R}^{J_k}$ such that

$$\sup_{\boldsymbol{a}\in\mathcal{A}}\sup_{z\in\mathcal{Z}}\left|f_{\boldsymbol{a},k}^{(j)}(z)-\{\boldsymbol{\beta}_{k,j}(\boldsymbol{a})\}^\top\boldsymbol{\Phi}_k^{(j)}(z)\right|<C_{0,j}J_k^{-(s-j)},\tag{52}$$

for some constant $C_{0,j}>0$. Then, we have

$$\sup_{\boldsymbol{a}\in\mathcal{A}}\sup_{z\in\mathcal{Z}}\left|f_{\boldsymbol{a},k}^{*(j)}(z)-f_{\boldsymbol{a},k}^{(j)}(z)\right|$$

$$=\sup_{\boldsymbol{a}\in\mathcal{A}}\sup_{z\in\mathcal{Z}}\left|\boldsymbol{\beta}_k^*(a)^\top\boldsymbol{\Phi}_k^{(j)}(z)-f_{\boldsymbol{a}}^{(j)}(z)\right|$$

$$\leq\sup_{\boldsymbol{a}\in\mathcal{A}}\sup_{z\in\mathcal{Z}}\left|(\boldsymbol{\beta}_k^*(\boldsymbol{a})-\boldsymbol{\beta}_{k,j}(\boldsymbol{a}))^\top\boldsymbol{\Phi}_k^{(j)}(z)\right|$$

$$+\sup_{\boldsymbol{a}\in\mathcal{A}}\sup_{z\in\mathcal{Z}}\left|\boldsymbol{\beta}_{k,j}(\boldsymbol{a})^\top\boldsymbol{\Phi}_k^{(j)}(z)-f_{\boldsymbol{a},k}^{(j)}(z)\right|$$

$$\leq\sup_{\boldsymbol{a}\in\mathcal{A}}\|\boldsymbol{\beta}_k^*(\boldsymbol{a})-\boldsymbol{\beta}_{k,j}(\boldsymbol{a})\|\cdot\zeta_j(J_k)+O(J_k^{-(s-j)}).\tag{53}$$

For $j = 0$, we have $(53) = O(J_k^{-s} \zeta_0(J_k))$ from Lemma F.7.

For $j = 1$, since $\mathcal{Z}$ is a compact set, using the fundamental theorem of calculus, (52) implies

$$\sup_{\boldsymbol{a} \in \mathcal{A}} \sup_{z \in \mathcal{Z}} \left| f_{\boldsymbol{a},k}(z) - \boldsymbol{\beta}_{k,j}(\boldsymbol{a})^\top \boldsymbol{\Phi}_k(z) \right| = O\left( J_k^{-(s-j)} \right). \tag{54}$$

Then, using a similar argument in Lemma F.7, we have the following result

$$\sup_{\boldsymbol{a} \in \mathcal{A}} \left\| \boldsymbol{\beta}_k^*(\boldsymbol{a}) - \boldsymbol{\beta}_{k,j}(\boldsymbol{a}) \right\| = O\left( J_k^{-(s-j)} \right). \tag{55}$$

Then we have

$$(53) = O(J_k^{-2(s-j)} \zeta_j(J_k) \tilde{\zeta}_j(J_k)).$$

We next prove $\mathbb{E}[|f_{\boldsymbol{a},k}^{(j)}(\boldsymbol{a}^\top \boldsymbol{x}) - f_{\boldsymbol{a},k}^{*(j)}(\boldsymbol{a}^\top \boldsymbol{x})|^2] = O(J_k^{-2(s-j)} \tilde{\zeta}_j(J_k))$.

$$\mathbb{E}\left[ \left| (\boldsymbol{\beta}_{k,j}(\boldsymbol{a}) - \boldsymbol{\beta}_k^*(\boldsymbol{a}))^\top \boldsymbol{\Phi}_k^{(j)}(\boldsymbol{a}^\top \boldsymbol{x}) \right|^2 \right]$$

$$= (\boldsymbol{\beta}_{k,j}(\boldsymbol{a}) - \boldsymbol{\beta}_k^*(\boldsymbol{a}))^\top \mathbb{E}\left[ \boldsymbol{\Phi}_k^{(j)}(\boldsymbol{a}^\top \boldsymbol{x}) \boldsymbol{\Phi}_k^{(j)}(\boldsymbol{a}^\top \boldsymbol{x})^\top \right] (\boldsymbol{\beta}_{k,j}(\boldsymbol{a}) - \boldsymbol{\beta}_k^*(\boldsymbol{a}))$$

$$\leq \left\| (\boldsymbol{\beta}_{k,j}(\boldsymbol{a}) - \boldsymbol{\beta}_k^*(\boldsymbol{a}))^\top \right\|^2 \cdot O(\tilde{\zeta}_j(J_k))$$

$$= O\left( \| \boldsymbol{\beta}_{k,j}(\boldsymbol{a}) - \boldsymbol{\beta}_k^*(\boldsymbol{a}) \|^2 \tilde{\zeta}_j(J_k) \right) = O\left( J_k^{-2(s-j)} \tilde{\zeta}_j(J_k) \right). \tag{56}$$

Thus, using similar decomposition as (53), we obtain

$$\mathbb{E}[|f_{\boldsymbol{a},k}^{(j)}(\boldsymbol{a}^\top \boldsymbol{x}) - f_{\boldsymbol{a},k}^{*(j)}(\boldsymbol{a}^\top \boldsymbol{x})|^2]$$

$$\leq 2\mathbb{E}\left[ \left| f_{\boldsymbol{a},k}^{(j)}(\boldsymbol{a}^\top \boldsymbol{x}) - \boldsymbol{\beta}_{k,j}(\boldsymbol{a})^\top \boldsymbol{\Phi}_k^{(j)}(\boldsymbol{a}^\top \boldsymbol{x}) \right|^2 \right]$$

$$+ \mathbb{E}\left[ \left| (\boldsymbol{\beta}_{k,j}(\boldsymbol{a}) - \boldsymbol{\beta}_k^*(\boldsymbol{a}))^\top \boldsymbol{\Phi}_k^{(j)}(\boldsymbol{a}^\top \boldsymbol{x})^\top \right|^2 \right]$$

$$= O\left( J_k^{-2(s-j)} \zeta_j(J_k) \right) + O\left( J_k^{-2(s-j)} \tilde{\zeta}_j(J_k) \right) = O\left( J_k^{-2(s-j)} \tilde{\zeta}_j(J_k) \right). \tag{57}$$

We finally prove $n^{-1} \sum_{i=1}^n |f_{\boldsymbol{a},k}^{(j)}(\boldsymbol{a}^\top \boldsymbol{x}_i) - f_{\boldsymbol{a},k}^{*(j)}(\boldsymbol{a}^\top \boldsymbol{x}_i)|^2 = O_P\left( J_k^{-2(s-j)} \tilde{\zeta}_j(J_k) \right)$. Note that, for large enough $J_k$, $\mathbb{E}\left[ |f_{\boldsymbol{a},k}^{(j)}(\boldsymbol{a}^\top \boldsymbol{x}) - f_{\boldsymbol{a},k}^{*(j)}(\boldsymbol{a}^\top \boldsymbol{x})|^2 \right] \leq C \cdot J_k^{-2(s-j)} \tilde{\zeta}_j(J_k)$, for some constant $C$. Choosing $C(\epsilon) = 2C/\epsilon$ and using the Markov's inequality, we have

$$\mathbb{P}\left( \frac{1}{n} \sum_{i=1}^n |f_{\boldsymbol{a},k}^{(j)}(\boldsymbol{a}^\top \boldsymbol{x}_i) - f_{\boldsymbol{a},k}^{*(j)}(\boldsymbol{a}^\top \boldsymbol{x}_i)|^2 > C \cdot J_k^{-2(s-j)} \tilde{\zeta}_j(J_k) \right)$$

$$\leq \frac{\mathbb{E}\left[ |f_{\boldsymbol{a},k}^{(j)}(\boldsymbol{a}^\top \boldsymbol{x}) - f_{\boldsymbol{a},k}^{*(j)}(\boldsymbol{a}^\top \boldsymbol{x})|^2 \right]}{C(\epsilon) \cdot J_k^{-2(s-j)} \tilde{\zeta}_j(J_k)} \leq \frac{\epsilon}{2},$$

which implies the third result. $\qquad \square$

**Lemma F.10.** *For any $k \in \{1, 2, \ldots, K\}$, suppose Assumptions F.1 – F.4 and (16) hold. For any $\boldsymbol{a} \in \mathcal{A}$ and $j = 0, 1, 2$,*

$$\sup_{\boldsymbol{x} \in \mathcal{X}} \left| \hat{f}_{\boldsymbol{a},k}^{(j)}(\boldsymbol{a}^\top \boldsymbol{x}) - f_{\boldsymbol{a},k}^{*(j)}(\boldsymbol{a}^\top \boldsymbol{x}) \right| = O_P\left( \{ J_k^{-(s-1)} + \sqrt{\xi_{n,k-1}} \} \zeta_j(J_k) + \frac{\zeta_j(J_k) \sqrt{J_k}}{\sqrt{n_q \wedge n_p}} \right),$$

$$\mathbb{E}\left[ |\hat{f}_{\boldsymbol{a},k}^{(j)}(\boldsymbol{a}^\top \boldsymbol{x}) - f_{\boldsymbol{a},k}^{*(j)}(\boldsymbol{a}^\top \boldsymbol{x})|^2 \right] = O_P\left( \{ J_k^{-2(s-1)} + \xi_{n,k-1} \} \cdot \tilde{\zeta}_j(J_k) + \frac{\tilde{\zeta}_j(J_k) J_k}{n_q \wedge n_p} \right),$$

$$\frac{1}{n} \sum_{i=1}^n |\hat{f}_{\boldsymbol{a},k}^{(j)}(\boldsymbol{a}^\top \boldsymbol{x}_i) - f_{\boldsymbol{a},k}^{*(j)}(\boldsymbol{a}^\top \boldsymbol{x}_i)|^2 = O_P\left( \{ J_k^{-2(s-1)} + \xi_{n,k-1} \} \cdot \tilde{\zeta}_j(J_k) + \frac{\tilde{\zeta}_j(J_k) J_k}{n_q \wedge n_p} \right).$$

*Proof.* Under Assumption F.3 and by Lemma F.8, we have

$$
\sup_{x \in \mathcal{X}} |\hat{f}_{a,k}^{(j)}(a^\top x) - f_{a,k}^{*(j)}(a^\top x)|
$$

$$
\leq \sup_{x \in \mathcal{X}} \left| \{\hat{\beta}_k(a) - \beta_k^*(a)\}^\top \Phi_k^{(j)}(a^\top x) \right|
$$

$$
\leq \|\hat{\beta}_k(a) - \beta_k^*(a)\| \cdot \sup_{z \in \mathcal{Z}} \|\Phi_k^{(j)}(t,z)\|
$$

$$
= O_P \left( \zeta_j(J_k)\{J_k^{-(s-1)} + \sqrt{\xi_{n,k-1}}\} + \zeta_j(J_k)\sqrt{\frac{J_k}{n_q \wedge n_p}} \right).
$$

Similar to (56), by Lemma F.8, we deduce that

$$
\int_{\mathcal{X}} |\hat{f}_{a,k}^{(j)}(a^\top x) - f_{a,k}^{*(j)}(a^\top x)|^2 dF_X(x)
$$

$$
= \int_{\mathcal{X}} \left| \hat{\beta}_k(a)^\top \Phi_k^{(j)}(a^\top x) - \beta_k^*(a)^\top \Phi_k(x^\top a) \right|^2 dF_X(x)
$$

$$
= (\hat{\beta}_k(a) - \beta_k^*(a))^\top \cdot \mathbb{E}\left[ \Phi_k^{(j)}(a^\top x)\Phi_k^{(j)}(a^\top x)^\top \right] \cdot (\tilde{\beta}_k(a) - \beta_k^{*(j)}(a))
$$

$$
= O_P \left( \|\hat{\beta}_k(a) - \beta_k^*(a)\|^2 \cdot \tilde{\zeta}_j(J_k) \right) = O_P \left( \{J_k^{-2(s-1)} + \xi_{n,k-1}\} \cdot \tilde{\zeta}_j(J_k) + \frac{\tilde{\zeta}_j(J_k)J_k}{n_q \wedge n_p} \right).
$$

Similar to the end of the proof of Lemma F.9, using Markov's inequality, we also have

$$
\frac{1}{n} \sum_{i=1}^{n} \left| \hat{\beta}_k(a)^\top \Phi_k^{(j)}(a^\top x_i) - \beta_k^*(a)^\top \Phi_k^{(j)}(x_i^\top a) \right|^2
$$

$$
= O_P \left( \int_{\mathcal{X}} \left| \hat{\beta}_k(a)^\top \Phi_k^{(j)}(a^\top x) - \beta_k^*(a)^\top \Phi_k^{(j)}(x^\top a) \right|^2 dF_X(x) \right)
$$

$$
= O_P \left( \{J_k^{-2(s-1)} + \xi_{n,k-1}\} \cdot \tilde{\zeta}_j(J_k) + \frac{\tilde{\zeta}_j(J_k)J_k}{n_q \wedge n_p} \right).
$$

$\square$

**Lemma F.11.** *For any $k \in \{1,2,\ldots,K\}$, suppose Assumptions F.1 – F.4 and (16) hold. We have*

$$
\sup_{a \in \mathcal{A}} \|\hat{\beta}_k(a)\| = O_P(1) \quad and \quad \|\partial_a \hat{\beta}_k(a)\| = O_P(1), \text{ for any } a \in \mathcal{A}.
$$

*Proof.* First, from (8), we have

$$
\sup_{a \in \mathcal{A}} \left| \int_{\mathcal{X}} \hat{f}_{a,k}^2(a^\top x) - f_{a,k}^2(a^\top x)p(x)\,dx \right| = O_P \left( J_k^{-(s-1)}\zeta_0(J_k) + \zeta_0(J_k)^2\sqrt{J_k}/\sqrt{n_q \wedge n_p} \right) = o_P(1).
$$

Then, by Lemma F.6,

$$
\sup_{a \in \mathcal{A}} \|\hat{\beta}_k(a)\|^2 = \sup_{a \in \mathcal{A}} \hat{\beta}_k(a)^\top \hat{\beta}_k(a) \leq \sup_{a \in \mathcal{A}} \eta_{\min}(\Omega_{J_k}^{(0)}(a))^{-1} \cdot \sup_{a \in \mathcal{A}} \hat{\beta}_k(a)^\top \Omega_{J_k}^{(0)}(a)\hat{\beta}_k(a)
$$

$$
= O(1) \cdot \sup_{a \in \mathcal{A}} \int_{\mathcal{X}} \hat{f}_{a,k}^2(a^\top x)dF_X(x)\,dx
$$

$$
\leq O(1) \sup_{a \in \mathcal{A}} \int_{\mathcal{X}} f_{a,k}^2(a^\top x)dF_X(x)\,dx + o_P(1)
$$

$$
= O_P(1).
$$

From (36) and (37), using Lemma F.6, we have

$$\|\partial_{\boldsymbol{a}}\hat{\boldsymbol{\beta}}_k(\boldsymbol{a}) - \Sigma_{J_k}(\boldsymbol{a})^{-1}M\| = o_p(1),$$

where

$$
\begin{aligned}
M =& 2\mathbb{E}_q\left[r_{k-1}^2(\boldsymbol{x})f^{(1)}(\boldsymbol{a}^\top\boldsymbol{x})\boldsymbol{\Phi}_k(\boldsymbol{a}^\top\boldsymbol{x})(\boldsymbol{x})^\top\right]\\
&+2\mathbb{E}_q\left[r_{k-1}^2(\boldsymbol{x})f(\boldsymbol{a}^\top\boldsymbol{x})\boldsymbol{\Phi}_k^{(1)}(\boldsymbol{a}^\top\boldsymbol{x})(\boldsymbol{x})^\top\right] - 2\mathbb{E}_p\left[r_{k-1}(\boldsymbol{x})\boldsymbol{\Phi}_k^{(1)}(\boldsymbol{a}^\top\boldsymbol{x})(\boldsymbol{x})^\top\right]\\
=&:M_1 + M_2 + M_3\,,
\end{aligned}
$$

where the definition of $M_1, M_2$ and $M_3$ are clear.

By triangle inequality,

$$\|\Sigma_{J_k}(\boldsymbol{a})^{-1}M\| \le \|\Sigma_{J_k}(\boldsymbol{a})^{-1}M_1\| + \|\Sigma_{J_k}(\boldsymbol{a})^{-1}M_2\| + \|\Sigma_{J_k}(\boldsymbol{a})^{-1}M_3\|\,.$$

Consider $\|\Sigma_{J_k}(\boldsymbol{a})^{-1}M_1\|$. Define

$$V(\boldsymbol{a}^\top\boldsymbol{x}) = \mathbb{E}_q\left[r_{k-1}^2(\boldsymbol{x})f^{(1)}(\boldsymbol{a}^\top\boldsymbol{x})\boldsymbol{\Phi}_k(\boldsymbol{a}^\top\boldsymbol{x})\mathbb{E}_q(\boldsymbol{x}|\boldsymbol{a}^\top\boldsymbol{x})^\top\right]\Sigma_{J_k}(\boldsymbol{a})^{-1}\boldsymbol{\Phi}_k(\boldsymbol{a}^\top\boldsymbol{x}),$$

which is the $L_2$ projection of $r_{k-1}(\boldsymbol{x})f^{(1)}(\boldsymbol{a}^\top\boldsymbol{x})\mathbb{E}_q(\boldsymbol{x}|\boldsymbol{a}^\top\boldsymbol{x})^\top$ onto the space linearly spanned by $\boldsymbol{\Phi}_k(\boldsymbol{a}^\top)$. Then we have

$$
\begin{aligned}
\|\Sigma_{J_k}(\boldsymbol{a})^{-1}M_1\|^2 =& \operatorname{tr}\left[M_1^\top\Sigma_{J_k}(\boldsymbol{a})^{-2}M_1\right] \le \eta_{\max}(\Sigma_{J_k}(\boldsymbol{a})^{-1})\operatorname{tr}\left[M_1^\top\Sigma_{J_k}(\boldsymbol{a})^{-1}\Sigma_{J_k}(\boldsymbol{a})\Sigma_{J_k}(\boldsymbol{a})^{-1}M_1\right]\\
\le& O\left(\|\mathbb{E}_q\{V(\boldsymbol{a}^\top\boldsymbol{x})V(\boldsymbol{a}^\top\boldsymbol{x})^\top\}\|\right) \le O\left(\|\mathbb{E}_q\{r_{k-1}^2(\boldsymbol{x})\{f^{(1)}(\boldsymbol{a}^\top\boldsymbol{x})\}^2\mathbb{E}_q(\boldsymbol{x}|\boldsymbol{a}^\top\boldsymbol{x})\mathbb{E}_q(\boldsymbol{x}|\boldsymbol{a}^\top\boldsymbol{x})^\top\}\|\right)\\
=& O(1)\,.
\end{aligned}
$$

Similarly, under Assumption F.3, we have $\|\Sigma_{J_k}(\boldsymbol{a})M_2\| = O(1)$ and $\|\Sigma_{J_k}(\boldsymbol{a})M_3\| = O(1)$. The result follows.

$\square$

