# OpenReview forum: "Projection Pursuit Density Ratio Estimation"
_ICML.cc/2025/Conference — ICML 2025 poster_

### Official Review · Reviewer_WQAb · 2025-03-05

**Overall Recommendation:** 3

**Summary:**

This paper considers the problem of density ratio estimation (DRE), where parametric methods may leads to bias, while non-parametric methods struggle with the curse of dimensionality in high-dimensional settings. The authors present an approach that leverages projection pursuit approximation to estimate the density ratio. Theoretical analysis establishes the estimator’s consistency and convergence rate. Additionally, experimental results demonstrate the effectiveness of the proposed method.

**Claims And Evidence:**

The consistency and the convergence rate of the proposed algorithms is established in Theorem 3.2.

The effectiveness of the algorithm is presented in Table 1,2 and 3, where the proposed density ratio estimation method is applied in several different cases and achieves good permormance

**Essential References Not Discussed:**

Not finding any missing essential related works.

**Experimental Designs Or Analyses:**

The experimental designs are in general sound. Only one question: how do you choose the hyper parameters of other algorithms? How can readers be convinced that the superiority performance of ppDRE is not an artifact of hyper parameter selection?

**Methods And Evaluation Criteria:**

The proposed method makes sense, since it has a clear motivation and mathematical derivation, and also has experimental support. For benchmark datasets, the authors use the dataset that has been adopted by previous works in the corresponding application, which is also reasonable.

**Other Comments Or Suggestions:**

No further comments.

**Other Strengths And Weaknesses:**

Strength: novel idea of leveraging projection pursuit in density ratio estimation. Clear motivation and explanation of the algorithm. Theoretical results are provided, justifying the consistency of the estimator. Experimental results are provided, showing the effectiveness of this algorithm, compared with prior works.

**Questions For Authors:**

I am wondering if you can see from your theory (Thm 3.2) of why your method achieves superior performance than other methods. Is your theory only about convergence rate and consistency, not any non-asymptotic result that can show your superiority?

**Relation To Broader Scientific Literature:**

The ppDRE method, as already shown in the paper, can be applied to dose response function estimation and other fields of science.

**Theoretical Claims:**

I did not check every proof in detail.

---

> ### Author Rebuttal · Authors · 2025-03-31
>
> Thank you for the positive comments.
>
> *Q1: Theoretical support for the superiority of the proposed method*
>
> We develop asymptotic convergence rates (Theorem 3.2) for our proposed projection pursuit density ratio estimator:
> $$ \sup_{x\in\mathcal{X}}| \hat{r}\_{K}(x) - r_{K}(x)| =O_{p}(\sum_{\ell=1}^K[\\{J_{\ell}^{-(s-1)}+ \sqrt{\frac{J_{\ell}}{n_q \wedge n_p}}\\}\cdot \Pi_{i=\ell}^K\\{\sqrt{\tilde{\zeta}\_1(J_i)}\vee \zeta^2_0(J_i)\\}])$$
> Please kindly refer to the details in our response to Q1 of Reviewer mSdz.  This rate does not depend on the dimension of the data, which highlights the strengths of projection pursuit modeling in mitigating the curse of dimensionality and may explain the superior performance compared to other methods based on direct nonparametric approximation. We agree with you that the non-asymptotic rates are more proper evidence for evaluating the performance of the estimator, but the proof presents significant challenges and is beyond the scope of this paper. We will pursue it in future work.
>
> *Q2: Hyperparameter selection process*
>
> To ensure rigorous and fair comparisons, we implemented the following protocol.
> - For our proposed ppDRE method, as mentioned in Remark 3.3, the optimal hyperparameters are identified by the minimal validation loss in CV. For clarity, we detail our approach as follows: across all experiments, we utilized 5-fold CV with random sampling. A grid search was conducted over a predefined set of parameter ranges, which are outlined in the table below:
>
> Table1. Hyperparameters for ppDRE method
>
> |parameter|description|searchspace|
> |:-:|:-:|:-:|
> |$K$|Number of PP iterations|\{5,10,15\}|
> |$J_k$|Number of basis functions|\{20,50,70,100,150\}|
> |$\lambda$|L2-penalty strength|\{0.5,1,5,10\}|
> |$\delta$|Gradient descent learning rate|\{0.001,0.01,0.1\}|
>
> - For open-source baseline methods (e.g., KLIEP, uLSIF, $\text{D}^3$-LHSS), we utilized their official implementations (e.g., Python densratio package for uLSIF) and default hyperparameter search grids as recommended in their original papers or standard toolkits. For instance, uLSIF's hyperparameters ($\sigma$, $\lambda$) were selected from the grid 1e-3:10:1e9, consistent with its standard implementation. These methods inherently incorporate CV-based hyperparameter selection in their standard workflows. We preserved these built-in CV mechanisms without modification.
> - For Classification and nnDRE, we implemented 5-fold CV (aligned with our ppDRE method) to identify the optimal hyperparameters within the search spaces outlined in the table below:
>
> Table2. Hyperparameters for Classification and nnDRE methods
>
> |     method     |   parameter   |      search space       |
> | :-: | :-: | :-: |
> | Classification | n_estimators  |       \{100, 300}       |
> |                | learning_rate | \{0.0001, 0.001, 0.01\} |
> |                |  num_leaves   |        \{20, 30}        |
> |     nnDRE      |     depth     |         \{2, 3}         |
> |                |     width     |      \{8, 32, 64\}      |
> |                |   learning    | \{0.0001, 0.001, 0.01\} |
>
> - For fDRE, we have implemented a version that employs KLIEP as the second-stage DRE method. Adhering to the guidelines provided by the official open-source code and the original research paper [1], we trained the masked autoregressive flow (MAF) models with the configurations presented in the table below. For the KLIEP method in the second stage, we have retained the original hyperparameter settings as per the open-source implementation.
>
> Table3. Hyperparameters for the flow model in fDRE method
>
> |Dataset|n_blocks|n_hidden|hidden_size|n_epochs|
> |:-:|:-:|:-:|:-:|:-:|
> |IHDP-Continuous|5|1|100|100|
> |Regression Benchmarks|5|1|100|100|
> |MI Gaussians|5|1|100|200|
>
> - For RRND,  in the absence of open-source code, we implemented the algorithm by adhering to the implementation details outlined in the Numerical Illustrations section of [2]. The kernel function is assigned as $K(x,x^\prime)=1+\exp\\{-(x-x^\prime)^2/2\\}$. Utilizing a 5-fold CV, the hyperparameter $\lambda$ is chosen based on the quasi-optimality criterion, and the optimal iteration step $k$ is determined by minimizing squared distance loss function in the validation set. Following the configurations in [2], the search grids are $k\in\\{1,2,\ldots,10\\}$ and $\lambda\in\\{\lambda_\ell=\lambda_0\rho^\ell,\ell=1,\ldots,w\\}$ with $\lambda_0=0.9$ and $w=9$. The decay factor $\rho=(\lambda_w/\lambda_0)^{1/w}$ is derived from the lower bound $\lambda_w=(n^{-1/2}+m^{-1/2})$, where $n,m$ are the sample sizes of the numerator and denominator distributions respectively.
>
> We will incorporate these implementation details into the Appendix of the revised manuscript to improve reproducibility.
>
> [1] Choi et al. Featurized Density Ratio Estimation, 2021
>
> [2] Duc Hoan Nguyen, Werner Zellinger, and Sergei Pereverzyev. On regularized Radon- Nikodym differentiation. Journal of Machine Learning Research, 25(266):1–24, 2024.

---

### Official Review · Reviewer_qyuE · 2025-03-09

**Overall Recommendation:** 3

**Summary:**

This paper proposes a non-parametric method for the density ratio estimation (DRE) task using a projection pursuit (PP) approximation. The method is computationally convenient and also helps alleviate the curse of dimensionality. The authors conduct extensive experiments demonstrating that the proposed method consistently outperforms other approaches.

**Claims And Evidence:**

Yes, the claims are well supported by strong theoretical proofs and a large number of experiments.Yes, the claims are well supported by strong theoretical proofs and a large number of experiments.

**Essential References Not Discussed:**

No, I believe all relevant references are adequately cited in this work.

**Experimental Designs Or Analyses:**

The datasets used in this paper are appropriate. The authors employ simulated data for a 2-D DRE experiment on stabilized weights
estimation. They also use the real IHDP dataset for dose-response function estimation. In the section on covariate shift adaptation, they use 5 different benchmark datasets.

**Methods And Evaluation Criteria:**

Most of the evaluation metrics make sense, such as MSE or similar measures. However, there is no metric provided to demonstrate the computational cost of this method.

**Other Comments Or Suggestions:**

No

**Other Strengths And Weaknesses:**

Strengths: The paper is well written and presents the ideas clearly, with solid proofs and thorough experimental validation.

Weaknesses: The main idea is not entirely new, but it is well integrated into this particular problem. There is no demonstration of computational cost in high-dimensional cases within the experiments

**Questions For Authors:**

No

**Relation To Broader Scientific Literature:**

This paper is a good application of the projection pursuit (PP) idea to the density ratio estimation problem. As stated in the paper, the approach is similar to nnDRE, except that the feedforward neural network (NN) is replaced by projection pursuit (PP).

**Theoretical Claims:**

Yes. I checked the proofs for Proposition3.1 and AppendixC (L2-distance Minimization); they all appear to be correct.

---

> ### Author Rebuttal · Authors · 2025-03-31
>
> Thanks for your suggestions.
>
> *Q1: Computational cost evaluation in high-dimensional cases*
>
> Compared to the conventional projection pursuit method (kindly refer to our response to Q1 of Reviewer fbc2) that requires additional Monte Carlo sampling to estimate the pursuit function, our estimators of $\\{f_{k}(·)\\}_{k=1}^K$ in equation (5) take closed-form expressions, enhancing computational efficiency. Moreover, we use Cholesky decomposition in computing the closed-form solution, improving numerical stability and reducing overall computation time.
>
> We evaluated computational costs via numerical experiments, which will be included in Appendix. Specifically, we will report the runtime of ppDRE method and all baseline methods (including newly added baselines, fDRE and RRND, suggested by Reviewers fbc2 and mSdz), in the same setup as Figure 3 in Section 4.2.1 with dimensions $d\in\\{2,10,30,50,100\\}$. The experiments were conducted on a node with dual AMD EPYC 7713 CPUs, totaling 128 cores. The computation time (minutes) averaged over 3 runs is reported below:
>
> Table4. Computation Time Comparison
>
> |Method|$d=2$|$d=10$|$d=30$|$d=50$|$d=100$|
> |:-:|:-:|:-:|:-:|:-:|:-:|
> |uLSIF|0.38|0.35|0.38|0.43|0.46|
> |KLIEP|0.36|0.39|0.49|0.57|0.72|
> |Class|0.21|0.16|0.19|0.23|0.40|
> |fDRE|3.78|3.88|3.92|4.28|7.25|
> |RRND|3.78|3.87|3.77|3.82|3.81|
> |nnDRE|5.91|3.84|8.76|6.02|8.63|
> |$\text{D}^3$-LHSS|5.95|11.25|10.60|10.79|10.97|
> |ppDRE|0.32|0.91|5.64|10.78|15.22|
>
> We observe the following results: (i) when the dimension is low or moderately large ($d=2,10$),  the computation time of our ppDRE method is comparable to that of the uLSIF, KLIEP, and classification methods, which are suitable for low-dimensional data.  Our computation time is much less than that of the fDRE, RRND, nnDRE and $\text{D}^3$-LHSS methods, some of which are also designed for high-dimensional data. (ii) When the dimension is large ($d=30,50,100$), the methods for low-dimensional data fail due to the curse of dimensionality. Our ppDRE method consistently outperforms all baseline methods in estimation accuracy, and the computation time is comparable to that of existing methods for high-dimensional data. The computation time increases significantly, which can be expected as we need more iterations to achieve convergence.
>
> *Q2: Adding fDRE and RRND as baselines*
>
> We will make the following improvements to our experimental analysis in Section 4 of the paper.
> -  We will incorporate the two methods as baselines.
> 1. We will add a discussion on the featurized DRE method (fDRE) in [1] (kindly refer to our response to Q3 of Reviewer fbc2). The fDRE method maps the data to a shared feature space via a flow model to mitigate inaccuracies caused by distribution discrepancy.
> 2. We will add a discussion on the regularized Radon-Nikodym differentiation estimation method (RRND) in [2] (kindly refer to our response to Q5 of Reviewer mSdz). The RRND method employs a regularized kernel method in RKHS to estimate the density ratio.
> - We will add the following experimental results of both baselines across key applications to Tables 1, 2 and 3 in our paper. Implementation details are given in our response to Q2 of Reviewer WQAb.
>
> Table1. ASE in dose response function estimation
>
> |Method|ARDF|QRDF(0.1)|QRDF(0.25)|QRDF(0.5)|QRDF(0.75)|QRDF(0.9)|
> |:-:|:-:|:-:|:-:|:-:|:-:|:-:|
> |fDRE|0.106(0.026)|0.331(0.056)|0.127(0.029)|0.045(0.016)|0.088(0.021)|0.267(0.061)|
> |RRND|0.117(0.022)|**0.307**(0.053)|0.125(0.026)|0.043(0.015)|0.090(0.021)|0.408(0.082)|
>
> Table2. MAE in mutual information estimation for varying dimension $p$ and correlation coefficient $ρ$
>
> |Method|$d=2,ρ=0.2$|$d=2,ρ=0.8$|$d=10,ρ=0.2$|$d=10,ρ=0.8$|$d=20,ρ=0.2$|$d=20,ρ=0.8$|
> |:-:|:-:|:-:|:-:|:-:|:-:|:-:|
> |fDRE|0.043(0.000)|0.907(0.048)|0.199(0.007)|4.283(0.014)|0.487(0.036)|8.710(0.059)|
> |RRND|0.061(0.001)|0.926(0.006)|0.200(0.008)|5.092(0.031)|0.414(0.022)|10.241(0.089)|
>
> Table3. NMSE under covariate shift adaptation
>
> |Method|Abalone|Billboard Spotify|Cancer Mortality|Computer Activity|Diamond Prices|
> |:-:|:-:|:-:|:-:|:-:|:-:|
> |fDRE|0.586(0.136)|0.615(0.084)|0.664(0.064)|0.294(0.053)|0.233(0.078)|
> |RRND|0.574(0.144)|0.616(0.087)|**0.655**(0.069)|0.279(0.051)|0.232(0.082)|
>
> The results indicate that our ppDRE maintains superior performance in most cases. The fDRE method (mapping data with a flow model prior to KLIEP) shows its effectiveness compared to the naive KLIEP method with enhanced performance in most cases, but remains inferior to ppDRE. The RRND method proves its merit by outperforming the non-regularized kernel methods (uLSIF, KLIEP) in several instances and even achieves top performance in some cases (bolded); however, its overall performance is less stable, ultimately underperforms ppDRE.
>
> [1] Choi et al. Featurized Density Ratio Estimation, 2021
>
> [2] Duc Hoan Nguyen, Werner Zellinger, and Sergei Pereverzyev. On regularized Radon- Nikodym differentiation. Journal of Machine Learning Research, 25(266):1–24, 2024.

---

### Official Review · Reviewer_mSdz · 2025-03-13

**Overall Recommendation:** 3

**Summary:**

This paper introduces a novel projection pursuit (PP)-based method for density ratio estimation (DRE), a critical task in machine learning with applications in areas like causal inference and covariate shift adaptation. Addressing the limitations of parametric methods (potential bias) and non-parametric methods (curse of dimensionality), the proposed approach approximates the density ratio function as a product of functions, each representing a projection onto a low-dimensional space. These projections are iteratively estimated using semi-parametric single-index functions and a regularized empirical loss, leading to a consistent estimator with a demonstrable convergence rate. Experimental results showcase the method's superior performance compared to existing DRE techniques in various applications.

**Claims And Evidence:**

Yes, the claims are made and supported by clear and convincing evidence

**Essential References Not Discussed:**

To put the paper in the proper place, it would be reasonable to include in the discussion the recent publications such as

[1] Duc Hoan Nguyen, Werner Zellinger, and Sergei Pereverzyev. On regularized Radon-
Nikodym differentiation. Journal of Machine Learning Research, 25(266):1–24, 2024.

[2] Elke R. Gizewski, Lukas Mayer, Bernhard A. Moser, Duc Hoan Nguyen, Sergiy
Pereverzyev, Sergei V. Pereverzyev, Natalia Shepeleva, and Werner Zellinger. On a
regularization of unsupervised domain adaptation in RKHS. Applied and Computational
Harmonic Analysis, 57:201–227, 2022.

[3] Qichao Que, Mikhail Belkin. Inverse Density as an Inverse Problem: The Fredholm Equation Approach. Advances in Neural Information Processing System, 26 (2013).

**Experimental Designs Or Analyses:**

Yes, I have checked. All experiments are designed to support the theory. However, it would be better if the up-to-date methods are added, such as in a recent paper [1].

[1] Duc Hoan Nguyen, Werner Zellinger, and Sergei Pereverzyev. On regularized Radon-
Nikodym differentiation. Journal of Machine Learning Research, 25(266):1–24, 2024.

**Methods And Evaluation Criteria:**

Yes, the proposed methods make sense for the problem.

**Other Comments Or Suggestions:**

1. Line 163 in the right column page 3, $k$ in this fomula $\hat{f}_k (\hat{a}_k^Tx)$ should be $m$.
2. In proposition 3.1, $I_{J_k}$ is not defined.

**Other Strengths And Weaknesses:**

Strengths: The authors provide the analysis of algorithms.

**Questions For Authors:**

1. The main results are two estimations for $\hat{f}_{a,k}$ and $\hat{a}_k$, but there is no error bound for density ratio $\hat{r}$. Can the authors provide a bound for $\hat{r}$?
2. Which norm do the authors measure the difference between $\hat{a}_k - a_k$?
3. Can the authors indicate the norms in which space? For example, in line 128 in the right column on page 3, the authors use “$||a||_2$”, but $||.||_2$ is not defined. In the proof of Theorem 3.2 in Appendix D, the authors use $||.||$, the readers cannot know what space the norm is considered.
4. It will be beneficial if the authors add to the baseline in experiments the up-to-date method such as in [1]

[1] Duc Hoan Nguyen, Werner Zellinger, and Sergei Pereverzyev. On regularized Radon-
Nikodym differentiation. Journal of Machine Learning Research, 25(266):1–24, 2024.

**Relation To Broader Scientific Literature:**

The proposed approach approximates the density ratio function as a product of functions, each representing a projection onto a low-dimensional space, to address the limitations of parametric methods (potential bias) and non-parametric methods (curse of dimensionality).

**Theoretical Claims:**

Yes, I have checked the proof of Proposition 3.1.

---

> ### Author Rebuttal · Authors · 2025-03-30
>
> Thank you for the constructive comments.
>
> *Q1: Error bound for density ratio $\hat{r}$*
>
> The main result in the original manuscript gives the convergence rates of $\hat{f}\_{a,k}$ and $a_l$ given the estimate $\hat{r}\_{k-1}$. Since our estimator $\hat{r}\_k$ is computed iteratively, and $\hat{r}\_0=r_0=1$, we can establish the rate of $\hat{r}\_k$ in an inductive way. We first suppose for $k=1,\ldots, K$, $n_p^{-1}\sum^{n_p}\_{i=1}\\{\hat{r}\_{k-1}(x_i^p)-r_{k-1}(x_i^p)\\}^2=O_p(\xi_{n,k-1})$ for some $\xi_{n,k-1}>0$. Using the arguments for establishing Theorem 3.2 in the original manuscript, we can establish the convergence rates:$$\sup_{a\in\mathcal{A}}\sup_{z\in\mathcal{Z}}|\hat{f}\_{a,k}(z)-f_{a,k}(z)|=O_p(J_k^{-s}\zeta_0(J_k)+\sqrt{\xi_{n,k-1}}\zeta_0(J_k)^2+\frac{\sqrt{J_k}\zeta_0(J_k)^2}{\sqrt{n_q\wedge n_p}})\tag{1}$$and $\\|\hat{a}\_k-a_k\\|=O_p(\\{J_k^{-(s-1)}+\sqrt{\xi_{n,k-1}}+\frac{\sqrt{J_k}}{\sqrt{n_q\wedge n_p}}\\}·\sqrt{\tilde{\zeta}\_1(J_k)})$
>
> where $\tilde{\zeta}_1(J)$ is a sequence of constants such that the maximum eigenvalue of $\mathbb{E}[\Phi_k^{(1)}(a^\top x)\Phi_k^{(1)}(a^\top x)^\top]$ is bounded by $\tilde{\zeta}_1(J)$ uniformly in $a\in\mathcal{A}$.
>
> We can then decompose$$\hat{f}\_{\hat{a}\_k,k}(\hat{a}\_k^\top x)-f_{a_k,k}(a_k^\top x)=\hat{f}\_{\hat{a}\_k,k}(\hat{a}\_k^\top x)-f_{\hat{a}\_k,k}(\hat{a}\_k^\top x)\tag{2}$$
> $$\qquad+f_{\hat{a}\_k,k}(a_k^\top x)-f_{a_k,k}(a_k^\top x)\tag{3}$$
> $$\qquad+f_{\hat{a}\_k,k}(\hat{a}\_k^\top x)-f_{\hat{a}\_k,k}(a_k^\top x)\tag{4}$$
> For (2),  we have $\sup_{x\in\mathcal{X}}|\hat{f}\_{\hat{a}\_k,k}(\hat{a}\_k^\top x)-f_{\hat{a}\_k,k}(\hat{a}\_k^\top x)|\leq\sup_{a\in\mathcal{A}}\sup_{x\in\mathcal{X}}|\hat{f}\_{a,k}(a^\top x)-f_{a,k}(a^\top x)|$.
>
> For (3), under Assumption D.2. (b), we can use Taylor's expansion to expand $f_{\hat{a}\_k,k}$ around $f_{a_k,k}$ and obtain $\sup_{x\in\mathcal{X}}|f_{\hat{a}\_k,k}(a_k^\top x)-f_{a_k,k}(a_k^\top x)|=O_p(\\|\hat{a}\_k - a_k\\|)$.
>
> For (4), we can first decompose it as$$(4) = [f_{a_k,k}(\hat{a}\_k^\top x) - f_{a_k,k}(a_k^\top x)] +[f_{\hat{a}\_k,k}(\hat{a}\_k^\top x) - f_{a_k,k}(\hat{a}\_k^\top x)] + [f_{a_k,k}(a_k^\top x)-f_{\hat{a}\_k,k}(a_k^\top x)].$$
> The supremum norm of the first term can be bounded by $O_p(\\|\hat{a}\_k-a_k\\|)$, using Taylor's expansion expanding $f_{a_k,k}(\hat{a}\_k^\top x)$ around $f_{a_k,k}({a}\_k^\top x)$. The second and the third term can also be bounded by $O_p(\\|\hat{a}\_k-a_k\\|)$ using the same argument for (3). Thus, we have$$\sup_{\in\mathcal{X}}|\hat{f}\_{\hat{a}\_k,k}(\hat{a}\_k^\top x)-f_{a_k,k}(a_k^\top x)|=O_p(\sup_{a\in\mathcal{A}}\sup_{x\in\mathcal{X}}|\hat{f}\_{a_k,k}(a_k^\top x)-f_{a_k,k}(a_k^\top x)|+\\|\hat{a}\_k-a_k\\|).$$
> Noting that $\zeta_0(J_k)\leq J_k$, we then have$$\sup_{x\in\mathcal{X}}|\hat{f}\_{\hat{a}\_k,k}(\hat{a}\_k^\top x)-f_{a_k,k}(a_k^\top x)|=O_p([J_k^{-(s-1)}+\sqrt{\xi_{n,k-1}}+\sqrt{\frac{J_k}{n_q\wedge n_p}}]·[\sqrt{\tilde{\zeta}\_1(J_k)}\vee\zeta^2_0(J_k)]),$$
> where $\sqrt{\tilde{\zeta}_1(J_k)}\vee \zeta_0(J_k)=\max[\sqrt{\tilde{\zeta}_1(J_k)},\zeta_0(J_k)]$.
>
> Finally, using the fact that $\xi_{n,0}=0$, we can inductively derive that$$\sup_{x\in\mathcal{X}}|\hat{r}\_{K}(x)-r_{K}(x)|=O_{p}(\sum_{\ell=1}^K[\\{J_{\ell}^{-(s-1)}+\sqrt{\frac{J_{\ell}}{n_q\wedge n_p}}\\}·\Pi_{i=\ell}^K\\{\sqrt{\tilde{\zeta}\_1(J_i)}\vee\zeta^2_0(J_i)\\}]).
> $$We will add this result to the paper.
>
> *Q2\&3: Clarification of norms*
>
> Thank you for your careful comments. We use the Euclidean norm to measure the distance between $\hat{a}\_k$ and $a_k$. To make the notation consistent, we will clearly define the Euclidean norm $\\| v\\|:=\sqrt{v^{\top}v}$ (without the subscript "2" shown in $\\|·\\|_2$ in the previous manuscript) for a general vector $v$, and we will apply it to both $\hat{a}\_k-a_k$ and $a$, i.e. $\\|\hat{a}\_k-a_k\\|$ and  $\\| a\\|$.
>
> *Q4: Adding Baseline from [1]*
>
> We will incorporate the regularized kernel method proposed in [1] into our experimental comparisons. Please kindly refer to our response to Reviewer qyuE. Due to space limitation, we addressed this point there.
>
> *Q5: Discussion of related work*
>
> Thank you for your references. We will add the following discussion to the introduction:
> > **Regularized Kernel Learning Methods**. Recently, the regularization scheme within reproducing Kernel Hilbert space (RKHS) has been developed for estimating the DRE problem. [3] reformulated the DRE problem as an inverse problem in terms of an integral operator corresponding to a kernel, then proposed a regularized estimation method with an RKHS norm penalty [1] established the pointwise convergence rate of the regularized estimator taking into account both the smoothness of the density ratio and the capacity of the space in which it is estimated.  [2] applied the regularized kernel methods in the context of unsupervised domain adaptation under covariate shift and developed the convergence rates.

---

> > ### Comment · Reviewer_mSdz · 2025-04-07
> >
> > Thank the authors for the rebuttal. The authors cleared up all my concerns. I am satisfied with the author's rebuttal and have no further questions.

---

> > > ### Author Response · Authors · 2025-04-08
> > >
> > > Thank you for your time and constructive feedback on our work. Your insights are greatly appreciated and have significantly contributed to enhancing the rigor and clarity of our manuscript.

---

### Official Review · Reviewer_fbc2 · 2025-03-14

**Overall Recommendation:** 3

**Summary:**

Parametric methods for DRE are susceptible to bias when model assumptions are misspecified, whereas traditional non-parametric approaches often struggle with the curse of dimensionality in high-dimensional settings. To overcome these limitations, the authors suggest using a projection pursuit (PP) based approach to estimate density ratios. They prove consistency and convergence rates for their suggested estimator. Their method is experimentally evaluated on diverse tasks and compared to several baselines.

**Claims And Evidence:**

Overall, the claims made in the paper are supported by theoretical and experimental evidence. Theorem 3.2 presents the theoretical findings, while Section 4 provides a detailed account of the experiments conducted on various datasets.

**Essential References Not Discussed:**

To enhance the comprehensiveness of the study, it would be beneficial to discuss related approaches or incorporate them as baseline methods for comparison. For instance, the method proposed in [3] first maps the data to a low-dimensional latent space before performing DRE to mitigate the curse of dimensionality.
Additionally, it would be valuable to explore how the theoretical results presented in this manuscript compare to the convergence rates established in [4].

[3] Choi et al. Featurized Density Ratio Estimation, 2021

[4] Gruber et al. Overcoming Saturation in Density Ratio Estimation by Iterated Regularization, 2024

**Experimental Designs Or Analyses:**

The experimental setup appears to be well-structured for evaluating the stated claims. I could not find a detailed description of the hyperparameter selection process, including dataset splits, search methodology, and search grid specification, however.

**Methods And Evaluation Criteria:**

The proposed PP method is well-motivated for the DRE setting, with a clear intuitive rationale for its potential to enhance performance. The selection of benchmark datasets and evaluation metrics is appropriate and aligns with standard practices in assessing DRE methodologies.

**Other Comments Or Suggestions:**

None

**Other Strengths And Weaknesses:**

None

**Questions For Authors:**

At first glance, PP for DRE seems to be an application of the PP algorithm conceptually very similar to how it is used in [1,2]. Could the authors emphasize the major differences/novelties compared to these approaches?

**Relation To Broader Scientific Literature:**

The primary contributions of this paper pertain to Projection Pursuit Density Estimation [1,2] and established DRE methods, which are utilized as baselines for comparison.

[1] Friedman et al. A projection pursuit density estimation, 1984

[2] Welling et al. Efficient Parametric Projection Pursuit Density Estimation, 2012

**Theoretical Claims:**

The results in Proposition 3.1 seem to be valid and skimming over the proof of Theorem 3.2 I did not find any inconsistencies.

---

> ### Author Rebuttal · Authors · 2025-03-30
>
> Thank you for the insightful comments. Below, we address them point by point.
>
> *Q1: Comparison with projection pursuit density estimation*
>
> [1] and [2] focused on estimating a pdf $p(x)$. In contrast, we aim at estimating the ratio $r^*(x)=p(x)/q(x)$ between two probabilities densities, which is significantly different from [1] and [2] in problem formulation, estimation methods, and theoretical results. We will include these discussions in the revised manuscript.
> - Comparison with [1]:
>
> |   |  $\qquad\qquad\qquad\qquad$  [1]      |  $\qquad\qquad\qquad\qquad$  Ours       |
> | :-------: | :----------------------: |:----------------------: |
> |  Model  |  $p(x)\approx p_K(x)=f_0(x)\prod_{k=1}^Kf_k(\theta_k^{\top}x)$ |     $r^*(x)\approx r_K(x)=\prod_{k=1}^Kf_k(a_k^{\top}x)$      |
> |   Identification   | They iteratively identify $f_k(\theta_k^{\top}x)$ by minimizing the KL distance between $p_K$ and $p$ | We iteratively identify $f_k(a_k^{\top}x)$ by minimizing the $L^2$ distance between $r_K(x)$ and $r^*(x)$ |
> | Estimation | Monte Carlo Sampling |    For a fixed $a_k$, we derived a closed-form estimator of the pursuit function $f_k(·)$, which provides convenience compuation of the estimator.   |
> | Theory  | No any theoretical justification |   We developed convergence results for our proposed method  |
>
> - Comparison with [2]:
>
> [2] proposed a parametric projected probabilistic product model:
> $$p(x)=p(y,z)=\prod_{i=1}^{d-J}\mathcal{N}(y_i|0,1)\prod_{j=1}^J\mathcal{T}(z_j|\alpha_j),$$
> where $y_i=v_i^{\top}x$ and $z_j=w_j^{\top}x$ are projections of $x$ onto $v_i$ and $w_j$ respectively, $\mathcal{N}(y_i|0,1)$ is the standard Gaussian distribution, and $\mathcal{T}(·|\alpha_j)$ is a univariate parametric distribution indexed by $\alpha_j$ such as the Student-t distribution. They further developed a sequential learning algorithm for this model. Note that, [2] employs a fully parametric approach, distinct from both [1] and our work which require estimation of unknown pursuit functions. Again, no theoretical results are provided in [2].
>
> *Q2: Hyperparameter selection process*
>
> We will add a detailed description in the paper. Please kindly refer to our response to Q2 of reviewer WQAb. Due to space limitation, we addressed this point there.
>
> *Q3: Discussion of  featurized DRE*
>
>  We will incorporate the fDRE method as a baseline (kindly refer to our response to reviewer qyuE for experimental results) and discuss it in the paper. We would like to clarify that [3] neither map the data to a low-dimensional latent space nor address the curse of dimensionality as focused by our paper. In fact, they tackled a different DRE challenge posed by significant distributional divergence between $p(x)$ and $q(x)$. To address this issue, they proposed an invertible parametric transform $f_{\theta}:\mathbf{R}^{d}\to\mathbf{R}^{d}$ such that the transformed densities become closer. Then $r^*(x)$ is estimated based on:
> $$r^*(x)= \frac{p(x)}{q(x)}=\frac{p'(f_{\theta}(x))}{q'(f_{\theta}(x))},$$ where $p'$ and $q'$ are densities of transformed data $f_{\theta}(x_p)$ and $f_{\theta}(x_q)$ respectively. Note that, the invertible transformation preserves the original data's dimensionality.
>
> *Q4: Theoretical results comparison with DRE with iterated regularization*
>
> We will add a discussion on the theoretical results established in [4].  [4] proposed an estimation method for the density ratio $r^*(x)$ by minimizing the Bregman distance based on a RKHS $\mathcal{H}$ with iterated regularization:
> $$r^{\lambda,t+1}:=\arg\min_{r\in\mathcal{H}} B(r^*(x),r)+\frac{\lambda}{2}\|r-r^{\lambda,t}\|^2.$$
> with $r^{\lambda,0}=0$. Their key theoretical result is an improved error bounds faster than the non-iterated error bound $(n_p+n_q)^{-1/3}$, under the Bregman distance and certain regular conditions (e.g. source condition and capacity condition). In contrast, we establish the convergence rates (after $k$th iteration) for our proposed estimator under sup-norm:
> $$\sup_{x\in\mathcal{X}}|\hat{r}\_{k}(x)-r_{k}(x)|=O_{p}(\sum_{\ell=1}^k[\\{J_{\ell}^{-(s-1)}+\sqrt{\frac{J_{\ell}}{n_q\wedge n_p}}\\}·\Pi_{i=\ell}^k\\{\sqrt{\tilde{\zeta}\_1(J_i)}\vee\zeta^2_0(J_i)\\}]),$$where $s$ is the smoothness of the pursuit functions $\\{f_j\\}\_{j=1}^k$. Under sufficient conditions, our rate can also achieve $o_P(n_p+n_q)^{-1/3}$. However, as the function spaces, approximation space, and distance metrics considered in [4] and our paper are completely different, it is difficult/meaningless to explicitly compare the two theoretical results based on the same standard.

---

### Decision · Program_Chairs · 2025-05-01

**Decision:**

Accept (poster)

**Comment:**

The reviewers appreciated the theoretical contributions made by the paper, as well as the potential practical relevance of the methodology for problems requiring density ratio estimation in higher dimensional applications where non-parametric techniques typically struggle.

The reviewers raised some concerns over the computational complexity/running time of the method, as well as querying the existence of finite sample bounds on accuracy. There were also multiple comments relating to the presentation of the paper contents, in that the contribution's position within the existing body of literature was not clearly and/or insufficiently justified.

I believe the authors have thoroughly addressed the vast majority of these matters in their rebuttal, and with the proposed modifications I believe the work may be a valuable contribution.